# A mouse brain stereotaxic topographic atlas with isotropic 1-µm resolution

Zhao Feng[1,2,3], Xiangning Li[1,2,3], Yue Luo[3], Xin Liu[3], Ben Long[3], Tao Jiang[2], Xueyan Jia[2], Xiaowei Chen[2], Jie Luo[2], Xiaokang Chai[3], Zhen Wang[3], Miao Ren[1,2], Xin Lu[3], Gang Yao[3], Mengting Zhao[1,3], Yuxin Li[3], Zhixiang Liu[3], Hong Ni[3], Chuhao Dou[3], Shengda Bao[3], Shicheng Yang[1], Zoutao Zhang[3], Jiandong Zhou[3], Lingyi Cai[3], Qi Zhang[3], Ayizuohere Tudi[3], Chaozhen Tan[3], Zhengchao Xu[3], Siqi Chen[3], Wenxiang Ding[3], Wenjuan Shi[3], Anan Li[2,3], Hong-wei Dong[4 ✉], Hui Gong[2,3 ✉] & Qingming Luo[1,2 ✉]

Multi-omics studies, represented by connectomes and spatial transcriptomes, have entered the era of single-cell resolution, necessitating a reference brain atlas with spatial localization capability at the single-cell level[1–4]. However, such atlases are unavailable[5]. Here we present a whole mouse brain dataset of Nissl-based cytoarchitecture with isotropic 1-µm resolution, achieved through continuous micro-optical sectioning tomography. By integrating multi-modal images, we constructed a three-dimensional reference atlas of the mouse brain, providing the three-dimensional topographies of 916 structures and enabling arbitrary-angle slice image generation at 1-µm resolution. We developed an informatics-based platform for visualizing and sharing of the atlas images, offering services such as brain slice registration, neuronal circuit mapping and intelligent stereotaxic surgery planning. This atlas is interoperable with widely used stereotaxic atlases, supporting cross-atlas navigation of corresponding coronal planes in two dimensions and spatial mapping across atlas spaces in three dimensions. By facilitating the data analysis and visualization for large brain mapping projects, our atlas promises to be a versatile brainsmatics tool for studying the whole brain at single-cell level.

Brain stereotaxic atlases have long served as essential references for determining spatial locations and understanding the organizational principles of biological structures in the brain. In recent years, the spatial transcriptomics at mesoscopic scale and neural circuit tracing at single-neurite level have led to urgent demand for determining the spatial location of any given cell within the brain[1,2], posing a higher challenge for the accurate anatomical localization of brain nuclei[3,4].

Traditional rodent brain reference atlases, composed of more than 100 Nissl-stained coronal sections, are manually annotated by experienced neuroanatomists[5–8]. These sections are spaced with intervals of hundreds of micrometres due to technical limitations and labour-intensive processes, preventing the observation of continuous changes, especially the starting and ending points of any given brain structures along the axial direction. This also hinders accurate three-dimensional (3D) reconstruction and the precise determination of anatomical boundaries[9]. Although some atlases simultaneously provide a few supplementary sagittal and horizontal planes, these slices come from different samples, leading to inconsistencies in identifying brain structures from different orientations. Further discrepancies in identifying brain structures may arise when researchers cut brain slices at angles different from the reference atlas, further limiting their utility[10].

To overcome the limitations of two-dimensional (2D) reference atlases and facilitate 3D brain mapping of large-scale neural circuits and multi-omics datasets, Wang et al.[11] constructed a common coordinate framework (CCF) based on the autofluorescence of the mouse brain tissue. Anatomical delineations of structures in the CCF were based on computationally derived average template at relatively low resolution, rather than actual cytoarchitecture. Furthermore, the axial resolution of the datasets used to construct this template is only 100 µm, which is insufficient for recognizing cellular-level details. Consequently, delineations of many brain structures became subjects of controversy[11]. These limitations are not well-suited for mapping single-neuron resolution morphology and spatial transcriptome data.

To address these challenges, we leveraged a 3D Nissl-stained image dataset with isotropic 1-µm resolution to construct a 3D mouse brain stereotaxic atlas, representing the topography of all structures while achieving single-cell resolution. We defined a spatial coordinate system for the atlas based on both cranial and intracranial reference points, which we called datum marks. Furthermore, we have developed visualization and application services for the atlas to meet the diverse needs of the scientific community in atlas visualization, intelligent stereotaxic surgery planning and more. Brain atlases have continuously evolved over the past 100 years. We believe this newly reconstructed mouse

[1]State Key Laboratory of Digital Medical Engineering, Key Laboratory of Biomedical Engineering of Hainan Province, School of Biomedical Engineering, Hainan University, Haikou, China. [2]HUST-Suzhou Institute for Brainsmatics, JITRI, Suzhou, China. [3]Britton Chance Center for Biomedical Photonics, MOE Key Laboratory for Biomedical Photonics, Wuhan National Laboratory for Optoelectronics, Huazhong University of Science and Technology, Wuhan, China. [4]Department of Neurobiology, David Geffen School of Medicine at UCLA, University of California Los Angeles, Los Angeles, CA, USA. ✉e-mail: HongWeiD@mednet.ucla.edu; huigong@brainsmatics.org; qluo@hainanu.edu.cn

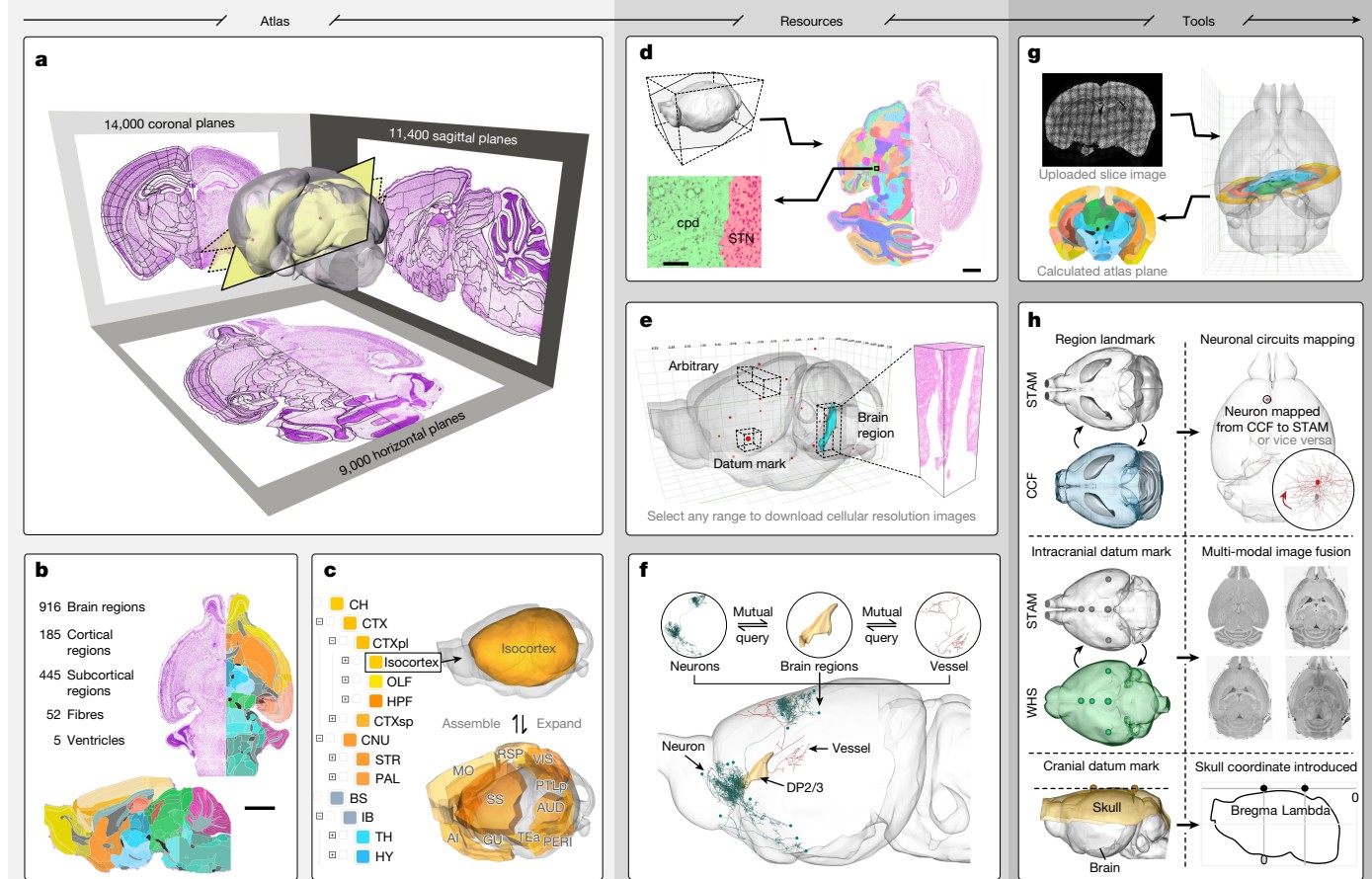

**Fig. 1 | Overview of STAM and its attached resources and tools. a**, Coronal, sagittal and horizontal planes resliced from the 3D cytoarchitectural image used to construct STAM, with black lines indicating delineated boundaries. Centre, reconstructed brain outline of STAM; yellow planes represent arbitrary-angle atlas planes. **b**, Examples of sagittal and horizontal atlas levels showing each region and nucleus in different colours. Top left, STAM's basic information. **c**, Left, hierarchically organized anatomical ontology. Top right, 3D topography of the isocortex. Bottom right, expanded subdivisions. **d**, Generating an arbitrary-angle plane. Top left, position of a resliced plane at an arbitrary angle. Right, corresponding generated atlas plane. Bottom left, magnified view of the boxed region. **e**, Interface for accessing the cytoarchitectural image used to construct STAM. Dashed and solid cubes show the region of interest selection for downloading. Centre, 3D topography of nucleus of the lateral lemniscus (NLL). Right, cropped 3D image within the bounding box of nucleus of the lateral lemniscus. **f**, Localizing neuroinformation in STAM. Insets at the top show different neuroinformation. Green dots indicate neuronal terminals. **g**, Brain slice registration tool on the STAM platform. Top left, slice to register. Bottom left, corresponding atlas level. Right, the location of corresponding atlas level in STAM. **h**, Inter-atlas mapping. Top row, bidirectional mapping between STAM and CCF enables integration of neuron morphologies (example on the right). Outline of CCF derived from the volume of Allen Mouse Brain Atlas (https://download.alleninstitute.org/informatics-archive/current-release/mouse_ccf/annotation/ccf_2017/annotation_10.nrrd). Middle row, mapping between STAM and WHS supports multi-modal image fusion. The second column from top to bottom shows horizontal sections of Nissl, T2*, T1 and T2-weighted (T2W) images, respectively. Bottom row, mapping from default to flat-skull position using cranial datum marks. Scale bars, 2 mm (**b**), 100 μm (**d** (left)), 1 mm (**d** (right)). AI, agranular insular area; AUD, auditory areas; cpd, cerebral peduncle; DP, dorsal peduncular area; GU, gustatory areas; MO, somatomotor areas; PERI, perirhinal area; PTLp, posterior parietal association areas; RSP, retrosplenial area; SS, somatosensory area; STN, subthalamic nucleus; TEa, temporal association areas; VIS, visual areas. Abbreviations for other brain structures correspond to the full names found in the Supplementary Information. In **h**, the outline and planes of WHS were derived from NITRC data (https://www.nitrc.org/projects/incfwhsmouse) and are adapted from ref. 22, PLoS, under a Creative Commons licence CC BY 4.0.

brain atlas, featuring a 1-μm isotropic resolution, will mark another milestone. It provides a versatile informatics tool for large-scale brain mapping projects and serves as a valuable 'traditional reference atlas' for numerous individual scientists.

## The mouse brain stereotaxic topographic atlas

We have constructed the stereotaxic topographic atlas of the mouse brain (STAM) with isotropic 1-μm resolution based on various types of dataset, including cytoarchitecture, immunohistochemistry and distribution of specific gene-type neurons. This atlas, available through a web portal (https://atlas.brainsmatics.cn/STAM/), comprises 14,000 coronal slices, 11,400 sagittal slices and 9,000 horizontal slices (Fig. 1a). Following the nomenclatures defined in the original Allen Reference Atlas (ARA) and Swanson's Brain maps v.4.0 (refs. 7,8), a total of 916 hierarchically organized brain structures are delineated and reconstructed in 3D, including 185 most-detailed cortical areas, and 445 most-detailed subcortical regions (Fig. 1b,c). We provide a list of discriminative criteria for each brain region in STAM, with most regions relying on two or more types of supporting evidence (Supplementary Table 1).

As the STAM is primarily based on isotropic 1-μm resolution image datasets of the whole brain, we also offer atlas levels generated at arbitrary angles, along with open access to this high-resolution dataset (Fig. 1d,e). The 3D STAM facilitates the localization of various types of neuroinformation, based on which we developed various web services to support neuroscience research (Fig. 1f). We also provide tools for conventional needs, such as brain slice registration, multi-modal image fusion and the use of a skull-based stereotaxic coordinate system (Fig. 1g,h).

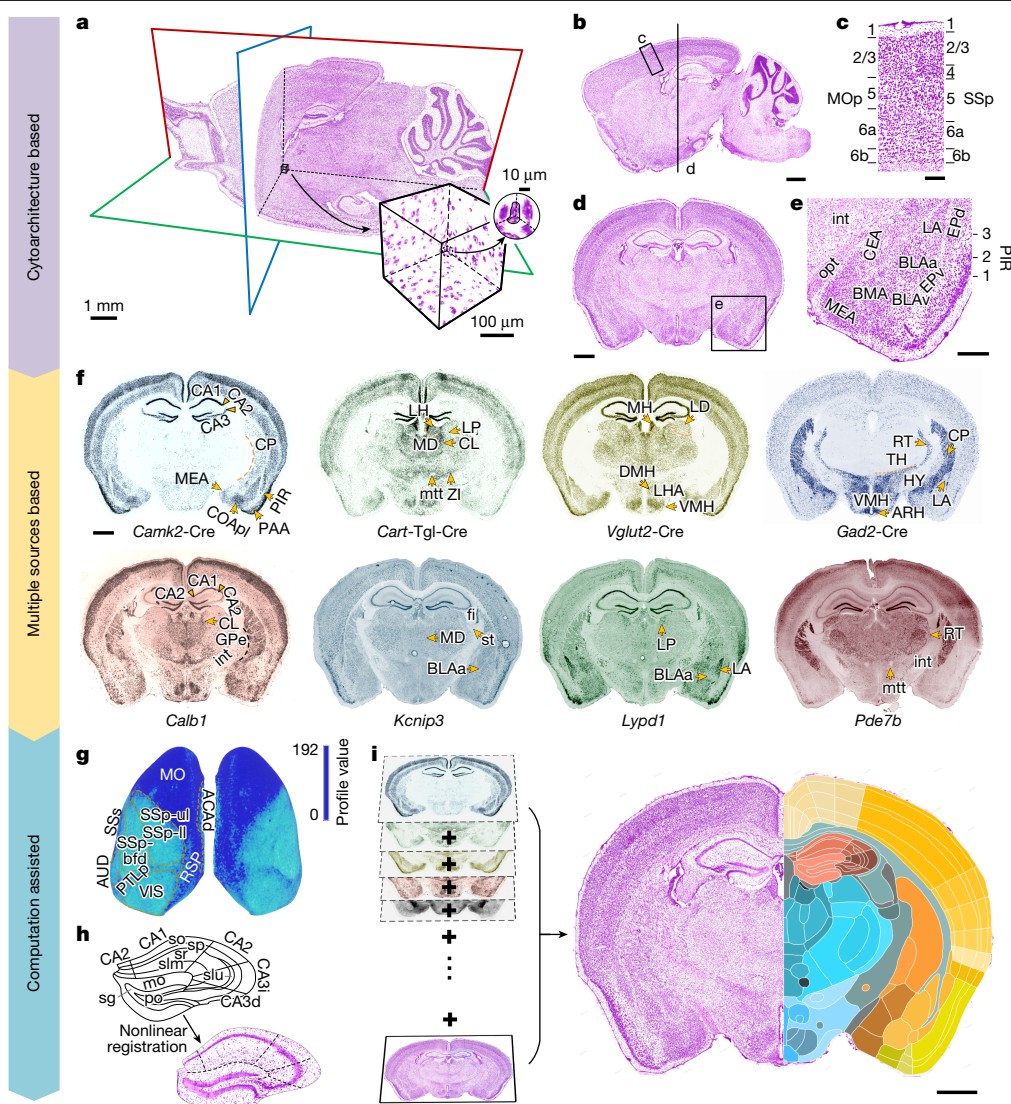

**Fig. 2 | Delineating brain regions and nuclei of STAM using high-resolution 3D cytoarchitectural image dataset and other supplementary approaches.** **a**, Original coronal, sagittal and horizontal planes from the MOST-Nissl dataset. The cube at the intersection of planes represents a randomly selected volume, shown magnified on the lower right. One corner of this volume is further enlarged to reveal neuronal somas with single-cell resolution. **b**, Sagittal section from the STAM cytoarchitectural image. **c**, Zoom-in of the cortical area in **b** showing the transition from five- to six-layer lamination. **d**, Coronal plane from the same dataset at the position indicated in **b**. **e**, Zoom-in of the boxed region in **d**. **f**, Examples of auxiliary datasets used for delineation. Arrows and labels highlight texture features indicative of structural borders. Image types and annotation approaches noted at the bottom; all images pseudocoloured. The second row is adapted from the Allen Mouse Brain Atlas (mouse.brain-map. org/experiment/show/79556672, mouse.brain-map.org/experiment/show/ 71587887, mouse.brain-map.org/experiment/show/72008305 and mouse. brain-map.org/experiment/show/71670683). **g**, Cortical region discrimination based on texture from layer 4 of a propidium iodide-stained dataset. Orange dashed lines mark boundaries, text labels indicate identified cortical areas. **h**, Ammon's horn (CA) region delineation assisted by image registration. Top, schematic redrawn of CA's subdivisions. Bottom, the corresponding STAM coronal plane overlaid with registered boundaries (dashed lines). **i**, Multi-source images registered to the same coronal level, offering integrated information for structural identification. Right, colour-coded labels of illustrated brain regions. All projection thicknesses are 20 μm, except **a** in which the single-plane thickness is 1 μm. Scale bars, 1 mm (**b**,**d**,**f**,**i**), 0.2 mm (**c**), 0.5 mm (**e**,**h**). BMA, basomedial amygdalar nucleus; CEA, central amygdalar nucleus; DMH, dorsomedial nucleus of the hypothalamus; EPd, endopiriform nucleus, dorsal part; EPv, endopiriform nucleus, ventral part; int, internal capsule; LA, lateral amygdala; MEA, medial amygdalar nucleus; MOp, primary motor area; opt, optic tract; PIR, piriform area; SSp, primary somatosensory cortex. Abbreviations for other brain structures correspond to the full names found in the Supplementary Information.

## 3D cytoarchitecture image with 1-μm resolution

Using an improved Nissl staining method and the micro-optical sectioning tomography (MOST) bright-field imaging technique, we obtained one 3D cytoarchitecture image dataset with a resolution of $0.35 × 0.35 × 1\ \mu m^3$, achieving micrometre-level resolution in both horizontal and axial directions[12]. The original data were processed into an isotropic 1-μm resolution in three sectional directions, and then mapped to the CCF (v.3, the same after) to achieve global morphological correction[13]. The corrected dataset, referred to as the MOST-Nissl dataset, has dimensions of 11,400 × 9,000 × 14,000 pixels and is further used for atlas construction (Fig. 1a).

The Nissl staining images obtained encompass neurons and glial cells throughout the entire brain, providing a recognizable representation of the shape and size of individual cells, which could not be observed on an averaged template created from individual specimens (Fig. 2a). The obtained high-resolution MOST-Nissl dataset provides rich cytoarchitecture information, including cell diversity and distribution patterns

in different brain structures, revealing their boundaries (Extended Data Fig. 1). For example, by examining the changes in lamination patterns of cells in the images, we can determine boundaries between different cortical areas (Fig. 2b,c). Also, by observing the discrepancies in density, size and morphology of somas, we can identify distinct subcortical regions (Fig. 2d,e).

Moreover, the isotropic 1-μm resolution of the MOST-Nissl dataset offers an advantage in observing the continuous changes of any specific anatomical structure on the 2D planes. This capability benefits the accurate determination of anatomical locations where the key features of certain brain structures 'appear' or 'disappear', thereby obtaining their subtle 3D topography. Using the small triangular nucleus of the septum in the cerebral nuclei (CNU) as an example, we observed its appearance along the anterior–posterior axis with 1-μm axial steps in the MOST-Nissl dataset, starting from the interior side of the septofimbrial nucleus and disappearing at the ventral side of the dorsal fornix (Supplementary Video 1). By contrast, traditional stereotaxic brain atlases, with larger intervals between coronal sections, cannot precisely reveal the entire triangular nucleus of the septum on the coronal plane and risk misinterpreting its remnants as the septofimbrial nucleus[6,7].

In addition to small nuclei, fibre bundles represent another category of morphologically complex brain structures. Taking the olfactory limb of the anterior commissure as an example, the MOST-Nissl images show this structure located in the anterior part of the mouse brain with symmetric branches on both hemispheres. Along the anterior–posterior direction, we can observe the location of the intersection point of these branches (Extended Data Fig. 1g–n), indicating that the spatial reference for brain-wide positioning is at the single-cell resolution.

## Atlas levels on canonical planes

Using cytoarchitectonic information as the foundation, supplemented by existing mouse brain atlases and other reference datasets, including distributions by genetically defined neuronal types[8,14,15], we initially focused on coronal section images to delineate different brain structures. In brief, the 20-μm thickness projected Nissl-stained coronal sections provided the main templates to identify anatomical structures, with aligned auxiliary coronal images from other datasets providing extra information (Fig. 2f, Extended Data Figs. 2–5 and Supplementary Table 1). The list of used datasets from our laboratory is provided in Supplementary Table 2. We also calculated the cytoarchitectural profiles along the depth of the cortex and mapped its distribution for delineating different cortical areas[16] (Fig. 2g). The delineation of hippocampal formation from previously published literature was also introduced through nonlinear registration[17] (Fig. 2h). The accuracy of all registrations has been evaluated. In most cases, the average Dice score for evaluation was above 0.8, indicating acceptable alignment. By incorporating the information from several sources, we created a comprehensive set of coronal atlas levels with abundant labels for brain structures (Fig. 2i). We ensured seamless adjacency of these labels to eliminate any 'terra nullius' between neighbouring structures.

Subsequently, we computed the obtained coronal atlas levels into sagittal and horizontal planes. By referencing the continuous cytoarchitectural features of the MOST-Nissl dataset on these two planes, we applied smoothing and optimization to the drawn boundaries to address the common 'jigsaw phenomenon' observed when sectional images are resliced into other planes in 3D space[18]. As the shapes of most brain structures are irregular, to avoid excessive smoothing that might lead to the loss of correct anatomical features, we resliced the optimized delineation back to the coronal plane for further examination. The boundaries of each brain structure in the three canonical anatomical planes of STAM underwent many iterations of examinations.

Once all traditional canonical anatomical planes were acquired, we developed a canonical plane visualization platform, comprising 700 coronal levels, 256 sagittal levels and 367 horizontal levels, each with a projection thickness of 20 μm. Specifically on the coronal plane, we integrated the delineation and nomenclature from Paxinos and Franklin's *The Mouse Brain in Stereotaxic Coordinates* (MBSC), based on the work in refs. 19,20. This will benefit the users of this atlas by providing an online version, and also enriches the delineation of brain structures, such as the subdivisions of caudoputamen. Furthermore, we also visualized the nonlinearly mapped specific gene-type neuron distribution datasets overlaid with STAM (Supplementary Table 3).

## Topography of whole-brain structures

By aligning and resampling the illustrated coronal atlas levels into a 3D image stack with isotropic 10-μm resolution, we generated the 3D topography of each brain structure through surface reconstruction (Fig. 3a). With careful balancing, we preserved the 3D topographies of structures, minimizing artefacts from the reconstruction process. The resulting models retain many anatomical details, as seen in Fig. 3a,b. Brain structures were reconstructed from the most-detailed level of the brain structural ontology, with higher-order structures hierarchically assembled from their constituent parts, creating a complete set of brain structures across different anatomical levels (Fig. 3c and Supplementary Video 2).

The reconstructed topographies highlight the changing and irregular nature of brain areas. For example, the thalamus retains fine anatomical features such as fibre-penetrating holes (Fig. 3d). Benefitting from the details provided by the high-resolution MOST-Nissl dataset, we could reconstruct fine structures, especially stratifications. One example is the three-layer structure of the olfactory tubercle, including the islands of Calleja embedded within it, which are hard to reconstruct on non-cytoarchitectural images (Fig. 3e). Another example is the somatosensory area, in which we not only visualized its 3D morphology and spatial relationship with surrounding sensory regions, but also subdivided the second somatosensory area into six layers (Fig. 3f).

We developed a 3D visualization platform for STAM that incorporates the 3D atlas label images, reconstructed 3D models and single-neuron morphology datasets from various sources[21]. These data are deeply integrated, allowing us to calculate the brain regions supplied by specific vascular branches and the branches that pass through particular brain regions. The same procedures apply to the relationships between brain regions and single-neuron morphology data. We provide an online query service for this integrated information, facilitating systematic analyses based on many types of neuroinformation (Fig. 3g).

## The brain-wide positioning system

To establish intracranial datum marks, we first selected eight anatomical structures, including the anterior commissure, nucleus ambiguous, corpus callosum, dentate gyrus granular layer (DGsg), dorsal raphe, lateral amygdalar nucleus, medial geniculate complex and the facial nerve (VIIn), from the MOST-Nissl dataset that are easily recognizable in cytoarchitecture images. We defined geometric features such as the centre and endpoints of these structures as intracranial datum marks, totalling 18 points (Extended Data Fig. 6a). Their specific names and coordinates can be found in Supplementary Table 4. The selection of intracranial datum marks considers the anterior–posterior, left–right and superior–inferior directions within the brain. These datum marks were determined on the basis of their surrounding cytoarchitecture information in three anatomical orientation images by neuroanatomists (Extended Data Figs. 6b and 7).

To bridge STAM and traditional skull-based stereotaxic coordinates, we then used a fluorescent MOST (fMOST) technique to acquire a dataset containing both skull and brain tissue with propidium iodide staining, providing a 3D image of the entire mouse head with a horizontal resolution of 0.325 μm per pixel and axial resolution of 1 μm per pixel.

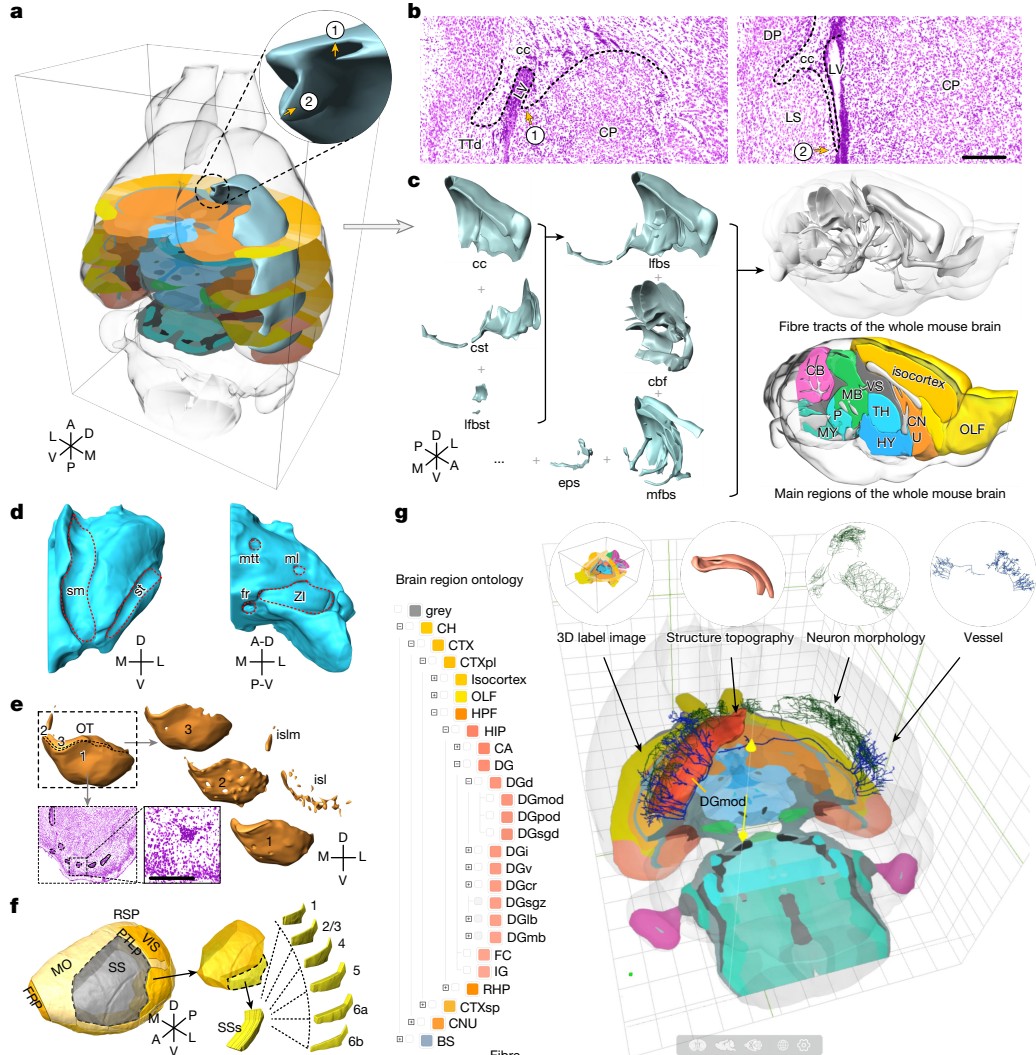

**Fig. 3 | Three-dimensional reconstruction of brain structures of STAM.**
**a**, The corpus callosum (cc) reconstructed from label image stacks (grey model); inset shows detailed topography, including two distinct tips (1, 2). The transparent model wrapping the label images is the outline of STAM. **b**, Coronal cytoarchitectural images corresponding to the tips confirm they are authentic anatomical features, not reconstruction artefacts. Dashed lines mark the corpus callosum boundaries. The numbers in this panel mark the same topographic features as in **a**. **c**, Fibre tracts reconstructed hierarchically (left), integrated into a whole-brain fibre system (upper right). Lower right shows STAM's full structural composition: olfactory system (OLF, yellow), isocortex (dark yellow), CNU (orange), thalamus (TH, light blue), hypothalamus (HY, dark blue), midbrain (MB, green), cerebellum (CB, magenta), pons (P, cyan), medulla (MY, dark cyan), fibre tracts (light grey) and ventricles (VS, dark grey), overlaid on a transparent brain outline. **d**, Reconstructed surface of the thalamus, showing indentations and holes formed by fibre bundles (for example, mammillothalamic tract (mtt),

medial lemniscus (ml), fasciculus retroflexus (fr)) and adjacent grey matter (for example, the zona incerta (ZI)). **e**, Reconstructed surfaces of olfactory tubercle (OT) and its subdivisions (right), based on coronal cytoarchitectural images (lower left). Dashed curves indicate boundaries; the magnified view highlights details. **f**, Hierarchical reconstruction of the isocortex, including the somatosensory area and its subregion supplemental somatosensory area (SSs), further subdivided into six cortical layers (right). **g**, Snapshot of STAM's 3D visualization platform. Checked structure 'DGmod' is shown in red. Overlaid objects include a neuron (green), vascular branch (blue) and horizontal atlas slice (bottom). Four insets show each element separately, with arrows indicating locations. Transparent contour marks the brain outline. Scale bars, 0.2 mm (**b**), 0.5 mm (**e**). A, anterior; P, posterior; D, dorsal; V, ventral; M, medial; L, lateral; A-D, anterodorsal; P-V, posteroventral. Abbreviations for other brain structures correspond to the full names found in the Supplementary Information.

This dataset allowed us to obtain the 3D structures of cranial datum marks, namely bregma and lambda (Extended Data Fig. 6c,d). We extracted the contour of the MOST-Nissl dataset and the cranial cavity from the 3D image dataset of the entire mouse head, and aligned them, establishing the spatial correspondence between cranial and intracranial datum marks[13] (Extended Data Fig. 6e–g and Supplementary Table 4).

As STAM's datum marks are distributed throughout the entire brain and some of them can be identified in magnetic resonance imaging (MRI) and/or immunohistochemistry images, they serve as landmarks for constructing spatial mapping relationships between non-whole-brain data and certain non-optical imaging modalities.

We established a spatial mapping relationship between STAM and the Waxholm Space (WHS), a MRI-based atlas[22,23] (Extended Data Fig. 6h–j).

## Visualizing STAM on arbitrary-angle planes

We visualized 2D atlas planes at arbitrarily selected cutting angles, with isotropic 1-μm resolution and any desired projection thickness between 1 and 20 μm (Fig. 4a,b and Supplementary Video 3). This capability distinguishes our STAM from traditional atlases and CCF that could only provide the three canonical planes.

One advantage of arbitrary-angle planes is the ability to observe anatomical features that are not visible in traditional planes. For example,

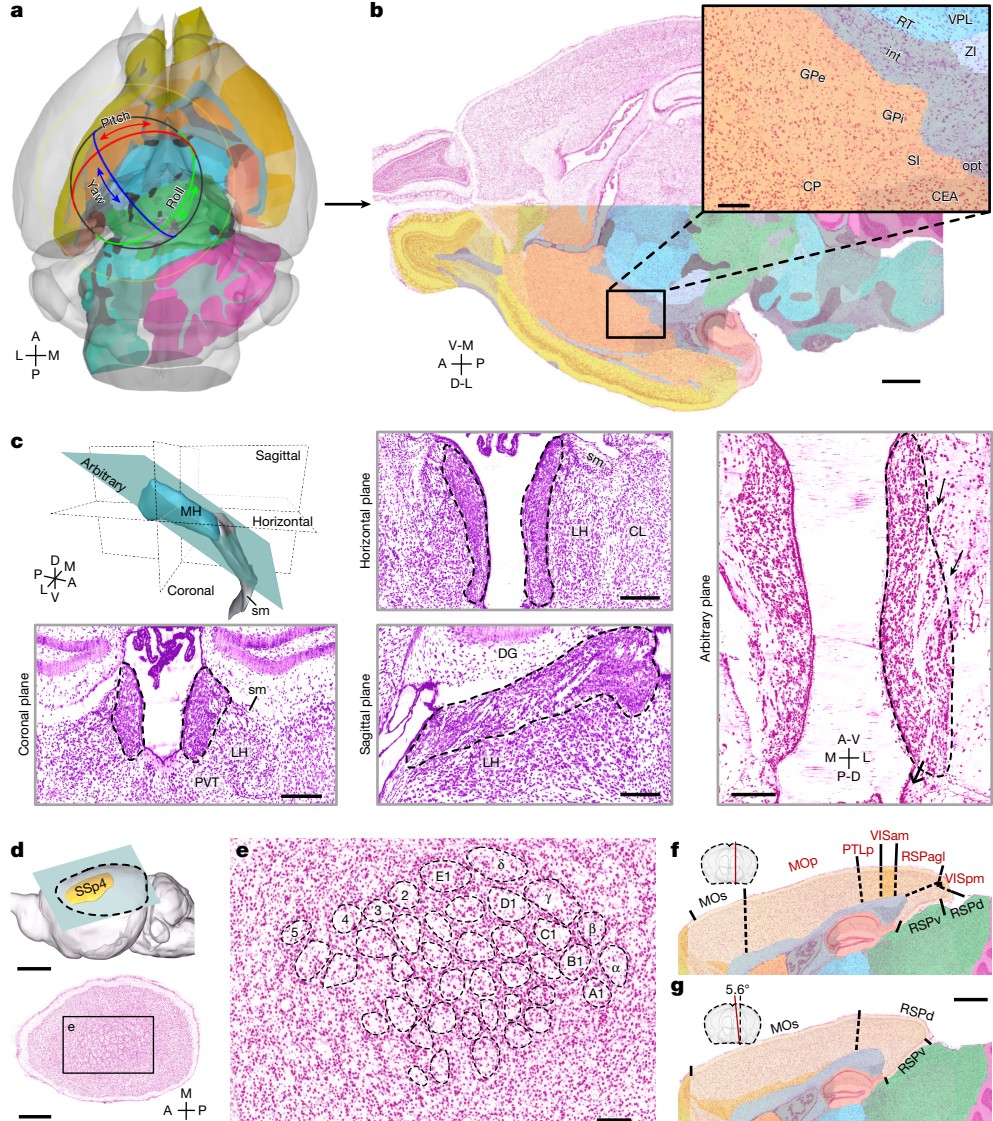

**Fig. 4 | Arbitrary-angle reslicing of the 3D cytoarchitectural image of STAM with isotropic 1-µm resolution. a**, Arbitrary-angle atlas level shown within the transparent 3D brain outline of STAM; a red–green–blue ring widget allows interactive pitch, roll and yaw adjustments. **b**, Corresponding cytoarchitectural image resliced from the 3D MOST-Nissl dataset with the same angles of **a**; top right shows a magnified region. **c**, Top left, the 3D view of the medial habenula (MH, blue) and surrounding fibre structure (stria medullaris (sm), silver). Dashed planes indicate canonical slices; the blue plane represents a non-canonical orientation from dorsoposterior to anteroventral. The top middle, bottom left and bottom middle correspond to the dashed planes in the top left. Right, resliced image at the same angle as the blue plane on the top left, revealing the fibre bundle through MH (black arrows) visible only in this orientation. **d**, Identification of the layer 4 barrel field in the primary somatosensory cortex using a non-canonical plane. The top panel indicates the slicing orientation; the bottom reveals the resliced image from the MOST-Nissl dataset, with visible barrel-like cytoarchitectural structures. **e**, Magnified region from **d**, with dashed outlines highlighting barrels. **f,g**, Cortical subdivisions on canonical (**f**) and non-canonical (**g**, yaw 5.6°) sagittal slices. Red labels emphasize distribution differences. Inset coronal views show plane positions (red lines). Scale bars, 1 mm (**b**), 200 µm (**b** (inset)), 0.3 mm (**c** (left and middle)), 0.2 mm (**c** (right),**e**), 1 mm (**d** (bottom),**f**), 4 mm (**d** (top)). D-L, dorsolateral; V-M, ventromedial; A-V, anteroventral; P-D, posterodorsal; CP, caudoputamen; GPe, globus pallidus, external segment; GPi, globus pallidus, internal segment; RSPagl, retrosplenial area, lateral agranular part; RT, reticular nucleus of the thalamus; SI, substantia innominata; VISam, anteromedial visual area; VISpm, posteromedial visual area; VPL, ventral posterolateral nucleus of the thalamus. Abbreviations for other brain structures correspond to the full names found in the Supplementary Information.

although the medial habenula is easily recognized on coronal, horizontal and sagittal plane, the complete morphology of the fibre bundle traversing the entire medial habenula is only visible from a resliced plane at a specific angle, running anteroventral to dorsoposterior (Fig. 4c). The barrel field of primary somatosensory cortex is another feature structure that can only be observed from a deviated angle of view (Fig. 4d,e).

Given the complex 3D morphology of many brain structures and the rapid transitions between anatomical regions, even a slight angle shift can produce varied atlas planes. Figure 4f,g compares canonical and non-canonical sagittal planes, showing how a mere 5.6° yaw angle causes the primary motor area, posterior parietal association areas, retrosplenial area, lateral agranular part, anteromedial visual area and posteromedial visual area to disappear from the oblique plane. This not only indicates the importance of constructing an accurately delineated atlas, but also demonstrates the necessity of providing non-canonical atlas planes, which can facilitate brain anatomical and physiological investigations.

## Neuronal circuits mapping

The 3D space of STAM provides a foundation for spatial localization of subtle neuroinformation, such as neuronal circuits. We integrated 1,644 single-neuron morphological data obtained through fMOST imaging technology and public databases, registered them onto STAM, identified their soma locations and projection targets and analysed the connectivity between brain regions[24]. This enabled us to establish a comprehensive connectivity map of the entire mouse brain at a single-neuron projection level (Fig. 5a,b). This connectivity map depicts possible connections between any two brain structures, and the queried single-neuron morphology data can be visualized, with the locations of all the branching points and terminals labelled in the 3D space. Benefitting from the deep integration of multi-source neuroinformation, atlas levels on canonical or non-canonical planes could be visualized alongside neuron morphologies, facilitating the 3D spatial localization of neuronal circuits.

The STAM can also be used to localize afferent and efferent connections for any brain structure of interest (Fig. 5c). Using the newly annotated hippocampal structure 'Field CA1 of hippocampus, stratum oriens, dorsal domain' (CA1sod) as an example, our 3D visualization platform enables the observation of afferent neurons from the diagonal band nucleus (NDB) and lateral septal nucleus, rostral part (LSr) in different colours, along with their somas, branching points, terminals and the brain structures of STAM where they are localized (Fig. 5d).

We used the STAM to analyse the connection and projection pattern of the mapped neuron morphology data. Figure 5e visually compares the projection pathways from 'Field CA3 of hippocampus, pyramidal layer, intermediate domain' (CA3spi) and 'Field CA3 of hippocampus, stratum oriens, dorsal domain' (CA3sod). We found that one efferent neuron from CA3spi primarily projects to the ipsilateral and contralateral sides of hippocampus, with only a few axons reaching the lateral septal nucleus, caudal part (LSc) and LSr (Extended Data Fig. 8a). By contrast, another efferent neuron from CA3sod sends most fibres to deeper brain structures, such as the medial septal nucleus, dorsal peduncular area and taenia tecta. We observed the preferred projection among subregions and sublayers, as described in ref. 25. Furthermore, the efferent neuron from CA3spi primarily projects to the dorsal and intermediate domains of the Field CA3, whereas the ventral domain is sparsely projected. The efferent neuron from CA3sod primarily projects to the stratum oriens and stratum radiatum layers of CA1–CA3, with minimal projections to other layers. These differences are quantified in Fig. 5f. The subtle subdivisions of the hippocampus introduced to STAM, along with the localization of terminals, serve as important references for understanding neural information transmission and encoding in the hippocampus.

In addition to the spatial localization of projection terminals, we used STAM to localize the branching points of neuronal circuits. For the efferent neuron from CA3spi shown in Fig. 5e, we observed that its fibres primarily branch within the fimbria and septofimbrial nucleus before further projecting to the target nuclei LSr and LSc (Extended Data Fig. 8b). A similar branching pattern was found for its contralateral projections, which first give off branches in alveus before entering CA fields (Extended Data Fig. 8c). For the efferent neuron from CA3sod, we observed that its projections to CA fields mainly traverse through the stratum oriens layers along the hippocampal axes, branching into neighbouring layers as they progress (Extended Data Fig. 8d,e). By localizing both terminals and branching points, we can depict the complex projection patterns of individual neurons registered to STAM, supporting computational modelling of signal propagation dynamics at the single-neuron level[26].

## New annotations of STAM

We carefully compared the differences and similarities in the delineation between STAM and CCF, listing a total of 236 newly drawn brain regions and nuclei, as well as the criteria to determine these structures (Supplementary Table 5). Most comes from the finer and continuous cytoarchitecture information provided by the 1-μm resolution MOST-Nissl dataset. Some of these structures involve finer layering, primarily distributed in olfactory areas and CNU, whereas others involve more detailed subregions, mainly distributed in the hypothalamus and midbrain (Extended Data Fig. 9a–f). In particular, for structures such as the islands of Calleja that show island-like shapes, with individual particles having diameters of only a few tens of micrometres and scattered among other structures, STAM can still demonstrate their location and complete 3D topography (Extended Data Fig. 9f).

Other newly annotated brain structures mainly come from multi-source images registered onto the MOST-Nissl dataset, including those from specific gene-type neuron distributions, in situ hybridization (ISH) and immunohistochemistry. Specifically, we refined the boundaries of the nucleus accumbens (ACB) and zona incerta by identifying their subdomains using *Thy1*-Cre and *Vglut2*-Cre neuron distribution images obtained through fMOST technology (Extended Data Fig. 10a–d). We also used the ISH images from an online database[15] to define subdivisions of the periaqueductal grey (PAG) (Extended Data Fig. 10e). Furthermore, we aligned the subdomains of the hippocampal formation, which were delineated on the basis of combined connectivity and molecular markers incorporated in previous studies[17,27], onto STAM.

## Miscellaneous web services

To facilitate the application of STAM in neuroscience researches, we developed a series of web services for miscellaneous needs. Besides the visualization of 2D atlas levels and 3D topographies, we provide open access to this isotropic 1-μm resolution Nissl-stained dataset, allowing users to select any 3D spatial range of interest and choose various down-sampling rates for downloading. This openly accessible 3D cytoarchitecture brain image with 1-μm resolution reveals abundant and intact anatomical details of the entire mouse brain.

Leveraging the isotropic 1-μm resolution advantage of STAM, we also managed a cloud service for the spatial localization of brain slices stained with propidium iodide or 4′,6-diamidino-2-phenylindole (DAPI), offering a more automated approach for using brain atlases in neuroscience research, reducing manual comparison and registration calculations.

We also established a spatial mapping between STAM and CCF, enabling the nonlinear mapping of single-neuron morphology data and 3D whole-brain images from CCF to STAM, and vice versa. This tool suite includes an online mapping service, desktop plugins for Fiji and ImageJ and a Python-wrapped application program interface.

We used the defined datum marks to construct a micrometre-resolution brain-wide structural positioning system and developed a virtual stereotaxic surgery service for intelligent path planning. For example, when targeting the small, deeply embedded dopaminergic group A13, the service can automatically calculate an appropriate injection path while avoiding user-specified regions (Fig. 5g). This function is beneficial in situations in which certain cortical structures are integral to the entire neural circuitry and thus require protection, ensuring they remain undisturbed by the intrusion of injection equipment. We believe this service can facilitate presurgical planning for mouse brains, reducing failure risk and minimizing animal use, thus promoting animal welfare.

All web services have been integrated into a single entry-point (Supplementary Table 3). By offering open access to all the neuroinformation of our services, integrating them with mutual queries and providing an easy download routine for the atlas data as well as calculated data, we adhere to the FAIR principles of findability, accessibility, interoperability and reusability[28]. We believe that the various developed web services can meet diverse needs in neuroscience research, and provide a solid spatial localization foundation for integrating neural information from different sources and modalities.

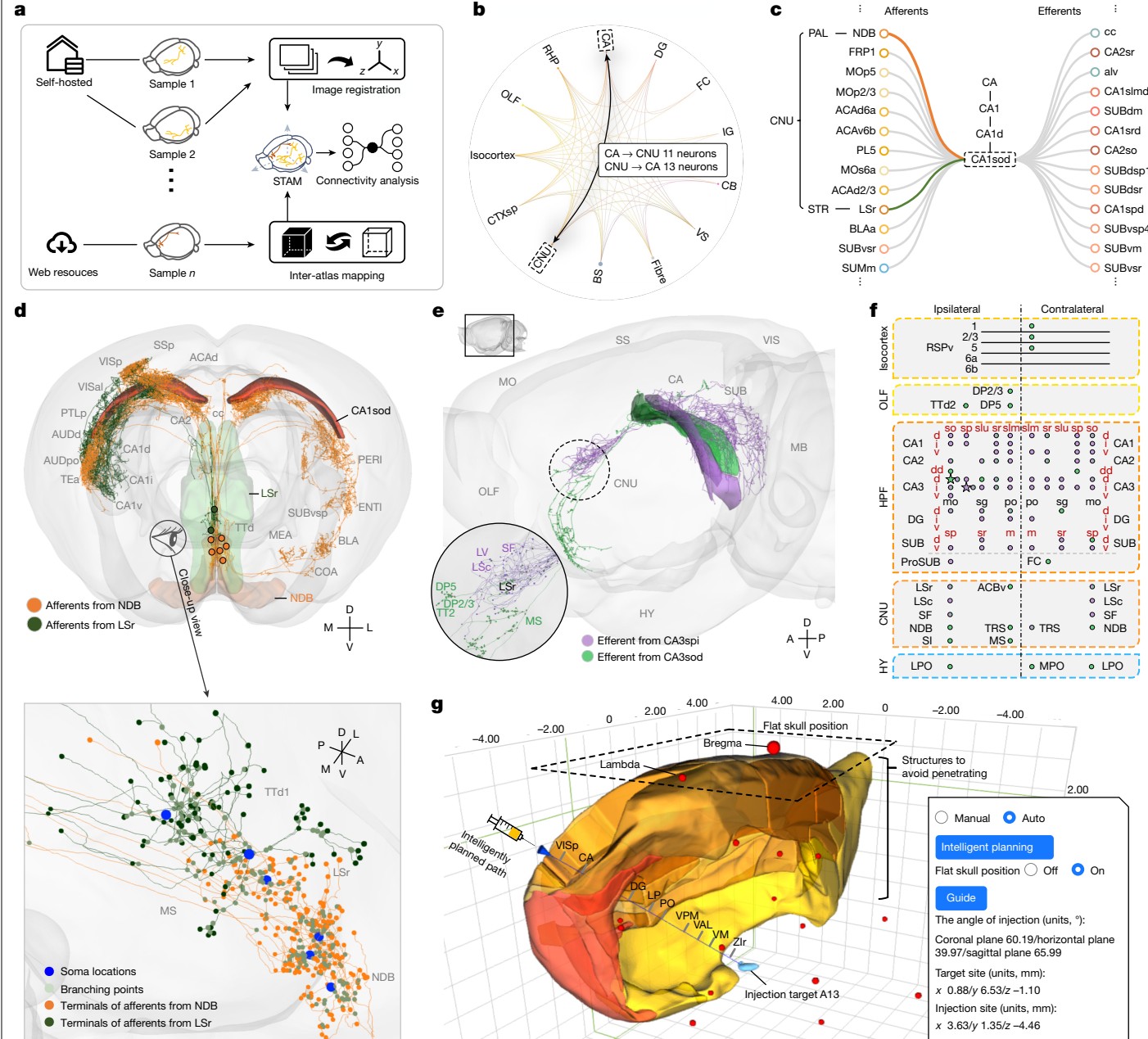

**Fig. 5 | Mapping neuronal circuits using STAM. a**, Pipeline for integrating single-neuron morphological data into STAM. **b**, A whole-brain connectivity diagram based on 1,644 neurons from many sources. A bold line between CA and CNU indicates the presence of projections, with adjacent text showing the number of neurons in each direction. **c**, The queried afferents and efferents of CA1sod displayed on the visualization platform. Green and orange lines indicate inputs from LSr and NDB, respectively. **d**, Top, afferent neurons originating from LSr (green) and NDB (orange), with somas marked by matching coloured dots. Models of CA1sod, LSr and NDB are overlaid with fibre trajectories and labelled target structures, with the transparent model of the brain outline of STAM as the background. The label texts mark the brain region and nuclei that the visualized neurons terminate. Bottom, magnified view indicated by the 'eye' icon in the top part of this panel, showing fibres (lines), somas, terminals and branch points (dots) colour-coded by type. **e**, Visualized comparison of efferent projections from CA3spi (purple) and CA3sod (green). 3D views of topography and projection patterns, with local insets highlighting region-specific differences. Purple and green labels mark distinct targets; black labels (for example, LSr) indicate shared targets. **f**, The summary diagram showing projection patterns of the two efferent neurons in **e**, with dots for terminals and stars for somas. **g**, Stereotactic injection planning using STAM. Dopaminergic A13 group (A13, target, light blue) and structures to avoid (red to orange) are visualized. The injection path shown in blue; bregma (red sphere) defines the origin. The panel on the right shows the calculated angle and coordinates. Abbreviations for other brain structures correspond to the full names found in the Supplementary Information. Panels **a** and **g** adapted from iSlide under a Creative Commons licence CC0 1.0.

## Discussion

This study used MOST technology to capture a 3D Nissl-stained cytoarchitectonic image dataset with isotropic 1-µm resolution. By combining immunohistochemistry, ISH, neural circuit labelling and distribution patterns of specific gene-type neurons from various reference datasets, we constructed a stereotaxic topographic atlas, STAM. This atlas enables exploration of cytoarchitecture images of the mouse brain from not only the coronal, sagittal and horizontal planes, but also at any angle at single-cell resolution. We reconstructed the 3D topography of 916 brain structures, offering a 3D visualization platform that integrates various neuronal data, including single-neuron morphology for

mutual query. At the core of STAM is a brain-wide positioning system, serving as the foundation for a suite of informatics tools designed to meet diverse neuroscience needs. These tools include inter-atlas mapping for 3D imaging and neuron morphology data, virtual surgery for stereotaxic surgery planning and brain slice image registration to a 3D reference atlas. We further established a connection between the Nissl staining sections from brainmaps.org to the corresponding planes from STAM to compare the cytoarchitecture information from different resources[29].

Before our work, there were several published stereotaxic atlases of the mouse brain[6,7]. However, inconsistencies in the anatomical planes from different directions, along with the large and non-uniform axial spacing between adjacent slices, made it challenging to align slices and reconstruct the authentic topography of brain structures. To address these issues, we used one set of 3D Nissl-stained images that simultaneously provides single-cell resolution image sequences of coronal, sagittal, horizontal and even arbitrarily oriented planes from the same brain.

There also exist 3D mouse brain atlases, represented by the CCF and WHS. However, the images used to build these atlases lack cytoarchitecture information, making it difficult to clearly depict fine details necessary for accurate delineation of brain structures, as shown in Extended Data Fig. 9g–j. Furthermore, the autofluorescence image datasets lack skull information, preventing the establishment of the cranial landmarks and skull-based coordinate system. This limitation hinders their use in typical neuroscience tasks, such as guiding stereotaxic surgeries. By contrast, the cytoarchitectural texture of our atlas aids in identifying more accurate structural boundaries. The reconstructed 3D topographies also show richer and more detailed anatomical features. Furthermore, we established a spatial positioning system spanning both the cranial and intracranial regions, making it the highest spatial resolution 3D brain atlas for any mammalian species to date. With these innovative features, our newly constructed STAM serves as a suited atlas for integrating results from single-cell level brain mapping projects, including spatial transcriptome and single-neuron morphology reconstructions.

There also exist limitations for our atlas construction pipeline. Despite the high-quality requirements for atlas images and the complex workflow resulting from management and quality control in collaborative brain-region annotation by many individuals, it took us roughly 10 years to establish a comprehensive solution for sample preparation, image acquisition and processing, atlas creation and the development of miscellaneous services. Considering the notably greater effort required for mapping non-human primate or even human brain atlases, in the future, we will leverage recent advancements in deep learning technology to automate and expedite the atlas creation by learning the texture information of different brain structures in cytoarchitecture images[14].

In future work, we aim to develop STAM into a more comprehensive atlas and provide more advanced analytical tools. Although the current version of the atlas focuses on the central nervous system of the mouse, we plan to extend this framework to include primate brains, such as the marmoset. Given the labour-intensive nature of constructing this atlas, we will develop deep learning-based methods for high-throughput automated atlas construction, leveraging the anatomical structure labelling of STAM. We also intend to integrate more neurological data into the STAM. Beyond multi-omics data and more comprehensive neuronal circuit information, STAM also holds promise for integrating pathological features such as amyloid plaques associated with Alzheimer's disease. The high-resolution spatial localization offered by STAM would facilitate the study of the spatial distribution of plaques and their potential impact on neuronal circuits throughout the brain, thereby advancing our understanding of neurodegenerative disorders and brain alterations in mouse models. Ultimately, our goal is to create a user-friendly, all-in-one open-access platform, serving as a foundational infrastructure of brainsmatics study in the near future.

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

# Methods

## Mice

All animal procedures were conducted in accordance with a protocol approved by the Institutional Animal Ethics Committee of HUST-Suzhou Institute for Brainsmatics, Suzhou, China. We used C57BL/6J mice, and transgenic mice that were crossed with Cre-recombinase-expressing mice and fluorescent-reporter mice (LSL-H2B-green fluorescent protein (GFP)). The Cre-recombinase-expressing mice used include Cart-Cre, Sert-Cre, Vglut2-Cre, Thy1-Cre, PlxinD1-CreER, Camk2-Cre, GAL-Cre, GAD2-Cre, Vgat-Cre, CRH-Cre, Emx1-Cre, Fezf2-CreER, TH-Cre, Chat-Cre, DAT-Cre, SOM-Cre and PV-Cre. The sample size and details can be seen in Supplementary Table 2. We used both male and female mice of 6–36 weeks old for brain image acquisition. The mice used in the experiments were housed in a standard specific pathogen-free animal facility under controlled conditions: a 12-hour light–dark cycle (7:00 to 19:00), room temperature maintained at 20–26 °C and relative humidity regulated between 40 and 70%.

## Nissl dataset acquisition

An adult C57BL/6J mouse (8-week-old, male) was anaesthetized with 1% sodium pentobarbital solution and then perfused intracardially with 4% paraformaldehyde (PFA) (in 0.01 M PBS, pH 7.4, 4 °C). Following this procedure, the whole brain was removed and postfixed in 4% PFA at 4 °C for 24 h. After fixation, the brain was washed in 0.01 M PBS for 2 days. Next, the brain was stained with a 0.25% thionine solution (in 0.1 M acetic acid-sodium acetate buffer solution, pH 6.0) for 10 days. For staining and the subsequent dehydration and infiltration steps, the solution was constantly kept on a rotary shaker (rotating diameter 20 mm, speed 1 revolution per second). Then, the stained brain was dehydrated in a graded series of ethanol and acetone solutions: 50% ethanol for 2 h; 16 iterations of 70% ethanol for 12 h each; 85, 95 and 100% ethanol for 2 h each; a 1:1 ratio of 100% ethanol and 100% acetone for 2 h; 100% acetone for 12 h and another 100% acetone step for 1 day. Subsequently, the dehydrated brain was infiltrated in 50, 75 and 100% Spurr resin concentrations for 8 h for each, then infiltrated in fresh 100% Spurr resin for 3 days. All the staining, dehydration and Spurr resin infiltration processes were performed at room temperature under rocking conditions. Finally, the brain was put in a silicone mould filled with 100% Spurr resin solution and then polymerized in an oven at 60 °C for 36 h. The Spurr reagents (SPI) were newly prepared. The 100% Spurr resin contained 10 g of ERL-4221, 7.6 g of DER-736, 26 g of nonenyl succinic anhydride and 0.2 g of dimethylaminoethanol. The 50 and 75% Spurr solutions (wt/wt) were prepared from 100% acetone solution and 100% Spurr solution[30].

Afterwards, the MOST system was used to acquire the high-resolution 3D dataset of a Nissl-stained resin-embedded mouse brain[12]. The specimen was consecutively cut into 1-µm-thick sections, while line-scanning imaging (×40, 0.8 numerical aperture objective) was performed simultaneously. The continuous whole-brain imaging lasted for about 10 days. The raw dataset, comprising more than 11,000 coronal sections with a voxel resolution of $0.35 \times 0.35 \times 1 \mu m^3$, totals larger than 0.74 terabytes. Because the adjacent sections obtained through MOST imaging are self-registered, this dataset can be directly reconstructed into an image volume without inter-slice alignment, allowing for arbitrary slicing along any angle and projection at any thickness. All animal experiments were approved by the Institutional Animal Ethics Committee of HUST-Suzhou Institute for Brainsmatics, and the same applies in the following data acquisition process.

## Immunohistochemistry dataset acquisition

An adult C57BL/6J mouse (8-week-old, male) was used for immunostaining with NeuN (a neuron-specific nuclear protein) and NF160 (a protein found in neuronal axons), following a protocol described in ref. 31. Briefly, the slices were washed with PBS, blocked with 5% bovine serum albumin and then incubated with the primary antibodies overnight at 4 °C. After washing with PBS, the secondary antibodies, namely Alexa Fluor 488 goat anti-mouse immunoglobulin G (IgG) (Invitrogen, catalogue no. A11029, 1:1,000 dilution) and Alexa Fluor 594 goat anti-rabbit IgG (Invitrogen, catalogue no. A11037, 1:1,000 dilution) were applied for 2 h at room temperature. Imaging was done afterwards with a multichannel fluorescence slide microscope (Olympus VS120), and ImageJ software was used for processing (National Institutes of Health).

## Specific gene-type neuron distribution dataset acquisition

The Cre-recombinase-expressing mice, including Cart-Cre, Sert-Cre, Vglut2-Cre, Thy1-Cre, PlxinD1-CreER, Camk2-Cre, GAL-Cre, GAD2-Cre, Vgat-Cre, CRH-Cre, Emx1-Cre, Fezf2-CreER, TH-Cre, Chat-Cre, DAT-Cre, SOM-Cre and PV-Cre, were crossed with fluorescent-reporter mice (LSL-H2B-GFP) to assay Cre expression. In the cross-hybridized strains, the fluorescent proteins labelled different types of neuron. The mice were anaesthetized and perfused with PBS followed by 4% PFA, and the whole brain was postfixed at 4 °C for 24 h. Samples were embedded with glycol methacrylate resin. Briefly, each intact brain was dehydrated in a graded ethanol series and glycol methacrylate series (70, 85 and 100%), followed by being embedded in a vacuum oven at 48 °C for 24 h for polymerization.

After that, the structured illumination fMOST was applied to acquire 3D datasets of all mouse brain samples at the voxel resolution of $0.32 \times 0.32 \times 2 \mu m$ (ref. 24). Briefly, for each sample, optical sectioning images of the top 2-µm-thick layer were acquired on the block face, then a diamond knife was used to remove the top layer so that a newly exposed surface was created to repeat the imaging-sectioning loop until the whole sample was sectioned and imaged. During the imaging process, propidium iodide was applied for real-time staining of the cytoarchitecture. Then we obtained the whole-brain dataset containing two channels: the green channel for the GFP signal that represents specific neurons and the red channel providing cytoarchitectonic information. For each brain, the image acquisition procedure lasted for about 3 days taking roughly 5,500 coronal images.

## Whole-head dataset acquisition

The adult C57BL/6J mouse (8-week-old, male) was anaesthetized and perfused with PBS, followed by 4% PFA. Then the whole head was removed and postfixed in 4% PFA at 4 °C for 48 h. The intact sample was subsequently dehydrated in a graded ethanol series, immersed in a graded LR-White resin (Ted Pella Inc.) series and embedded in a vacuum oven at 38 °C for 3 days. The embedded skull-brain sample was then imaged by a high-definition fMOST system[32]. During the imaging process, real-time counter-staining was also implemented to obtain propidium iodide-stained cytoarchitecture. A skull-brain dataset with a voxel resolution of $0.32 \times 0.32 \times 1 \mu m^3$ was finally obtained in 45 days, with a size of $61,460 \times 58,005 \times 22,048$ and a total raw data volume of 77.4 terabytes.

## MOST-Nissl dataset normalization

To eliminate individual differences from using one single brain sample, and to correct the overall shape and spatial distribution of various brain structures, we nonlinearly registered the MOST-Nissl dataset to the average brain template in the CCF at a 1-µm resolution using the BrainsMapi registration tool. Only the brain outline was selected as the anatomical landmark for this registration, to ensure symmetry along the mid-sagittal plane in 3D space and the registered dataset positioned in the required orientation for constructing the brain atlas. We evaluated the registration result using the Dice score, achieving a value above 0.8 for the brain outline, which is generally considered satisfactory, as shown in the 'outline' row and the 'Global registration for normalization' column of Supplementary Table 6.

After the registration, the normalized MOST-Nissl dataset was then resliced along the three image axes to obtain coronal, sagittal and

horizontal image sequences with a voxel size of 1 µm. For each direction, minimum intensity projections were applied according to the actual thickness needed for atlas construction, ensuring that the images were within the optimal thickness range for identifying brain regions and nuclei based on cytoarchitecture[13].

## Multiple-source auxiliary image registration

As the normalized MOST-Nissl dataset is a 3D image with isotropic 1-µm resolution, it serves as an ideal template for registering both 2D and 3D auxiliary images from different sources. Its high spatial resolution offers the potential for improved registration accuracy. The auxiliary image registration helps validate the parcellation of STAM, provides extra clues for identifying exact boundaries of specific brain regions and supplements cranial datum marks absent in the MOST-Nissl dataset. In most registration cases, the average Dice score for evaluation was above 0.8, indicating acceptable alignment. Below are the details for auxiliary image registration.

**Specific gene-type neuron distribution to Nissl.** Using cytoarchitectural information from the propidium iodide channel of the specific gene-type neuron distribution dataset, we identified boundaries of structures such as the hippocampal region and cerebellum as references. These structures were then nonlinearly registered at 1-µm resolution to the normalized MOST-Nissl dataset using the BrainsMapi tool. The resulting deformation field was subsequently applied to the GFP channel of the neuron distribution image, achieving coregistration of specific gene-type neuron distribution information with cytoarchitectural information. We quantitatively evaluated the registration quality across 22 specific neuron distribution datasets, using 12 brain regions as the statistical benchmarks. In most cases, the average Dice score exceeded 0.8, as shown in Extended Data Fig. 4e. Specifically, we evaluated the registration performance of a Camk2-Cre neuron distribution dataset for delineating borders among CA1, CA2 and CA3, with the results shown in Extended Data Fig. 3 and quantitative evaluation results in Supplementary Table 7.

**Immunohistochemistry and ISH to Nissl.** On the basis of the cytoarchitectural information provided by the immunohistochemical image, we identified the coronal plane in the normalized MOST-Nissl dataset that is closest to its axial position. Using a 2D nonlinear registration method provided by the ANTs tool, we registered the immunohistochemical data onto the selected coronal plane. The ISH dataset used in this study is derived from the Allen Institute's gene expression dataset, and the registration steps are the same as those for immunohistochemical images. The quantitative evaluation of the registration quality is given in Supplementary Table 8.

**Whole-head data to Nissl.** The whole-head dataset we acquired includes both cranial and intracranial brain tissue. We manually extracted the contour of the brain tissue from the whole-head dataset as the reference and then used the BrainsMapi registration tool to nonlinearly register the whole-head dataset to the MOST-Nissl dataset in 3D space. The quantitative evaluation of the registration quality is given in Supplementary Table 9.

The specific gene-type neuron distribution and whole-head datasets mentioned above are 3D high-resolution datasets obtained using MOST imaging technology, with data sizes ranging from terabytes to tens of terabytes. Traditional image registration algorithms struggle to handle such large datasets. Therefore, before registration, we preconverted them into a multi-resolution archived TDat format and performed registration in a parallel computing environment. As an example, for an uncompressed specific neuron distribution dataset of roughly 10 terabytes, we used a computing cluster with five nodes in which each node was configured with four CPU (E5-2600) cores and 16 GB of memory. The registration process took roughly 1 h.

## Brain structure delineation

**The strategy and pipeline used for parcellation.** The micrometre-resolution MOST-Nissl dataset provides detailed cytoarchitectural information, forming the basis for parcellating most brain structures. Given that neurons have a diameter of about 10 µm and cytoarchitecture is defined by clusters of cell bodies, we assume that brain-region boundaries shift at a 10-µm scale. Therefore, we chose to illustrate the boundaries of each brain structure at 20-µm axial intervals along standard sections from the normalized, 1-µm resolution MOST-Nissl dataset. Each coronal plane was projected with a 20-µm thickness to accumulate enough cells, aiding in distinguishing cell density differences between regions. These projected coronal sections were then handed over to neuroanatomical experts to identify structures on the images. When delineating the structures, the experts could use our 1-µm resolution MOST-Nissl dataset to observe and track the continuous cytoarchitectural changes along the axial direction, helping to determine the start and end points of a given brain structure. Resliced sagittal and horizontal sections also provided references when coronal views were unclear.

To further validate brain structural parcellation and assist in identifying challenging areas, we incorporated auxiliary images from various labelling and staining methods. Because these auxiliary images have already been registered to the corresponding Nissl-stained coronal planes, as the previous section described, the coregistered image stacks were imported into vector drawing software where illustrators switched between and observed coronal planes of different modalities. They manually outlined the boundaries of various structures, following the principle of 'seamless adjacency' to ensure no 'terra nullius' between neighbouring structures. For special cases in which certain brain regions undergo appearance, disappearance or notable morphological changes in the axial direction, we can appropriately reduce the axial spacing to capture their fine 3D morphology. After the drawing process, the vectorized structure boundaries on the coronal planes at 20-µm intervals were automatically converted into structure annotations with a resolution of 10 µm in both horizontal and axial directions on raster images. The raster images were resliced onto coronal and horizontal planes, and the anatomical structure boundaries on coronal and horizontal planes were manually smoothed and optimized.

**Isocortex.** Using the cytoarchitectural information from Nissl staining, we differentiated the isocortex into different layers along the radial direction. The discrepancies in laminar appearances of different cortical areas on cytoarchitectural images also provided supporting evidence to define the regional boundaries along the tangential direction. For example, a dense layer of cells between layers 1 and 2/3 in the ventral part of the retrosplenial area served as a landmark to determine the boundary between the ventral and dorsal parts of the retrosplenial area. In high-resolution images, the lateral part of the entorhinal area showed a layer of small cells in the fourth layer, helping to determine the boundary between the lateral part of the entorhinal area and the perirhinal area. In addition to manual observation, we calculated the grey level index along the radial direction on propidium iodide-stained cytoarchitectural images to assist in determining the boundaries between visual areas and the primary somatosensory area. For some boundaries that could not be identified in cytoarchitectural images, we used neuron distribution datasets to determine the positions of cortical areas such as auditory, infralimbic and orbital areas. Furthermore, we referenced cortical parcellation methods from other literature to determine the boundaries of regions such as gustatory, visceral, temporal association and ectorhinal areas[11].

**Hippocampus.** Parcellating the hippocampus is similar to the cortex, requiring both layering along the radial direction and zoning along the tangential direction. The MOST-Nissl dataset provided sufficient

cytoarchitectonic features to aid in the differentiation of the pyramidal layer, granular layer, molecular layer and so on. In addition, we used specific neuron distribution data to determine the boundaries of CA1, CA2 and CA3 along the tangential direction. Furthermore, through high-precision nonlinear registration, we coregistered more detailed information about hippocampal subregions from the reference literature onto our atlas[17]. The hippocampal subregion information used for registration is available at https://cic.ini.usc.edu/ (refs. 33,34).

**Other subcortical structures.** Distinct boundaries of the brain structures in the thalamus, hypothalamus and midbrain, as well as white matter fibres and ventricles, can be identified on cytoarchitectural images. For some challenging-to-identify subcortical structures, we introduced other methods for assistance. For example, specific gene-type neuron distribution images helped identify nuclei such as pedunculopontine nucleus, subregions such as the dorsal and ventral parts of ACB, the posterior part of basolateral amygdalar nucleus, the lateral part of the mediodorsal nucleus of the thalamus, the lateral part of the medial preoptic nucleus and the ventral part of medial geniculate complex. We also used the Allen ISH dataset to confirm nuclei such as the retrorubral area of midbrain reticular nucleus, subparaventricular zone and subregions of the PAG. We validated the parcellation patterns of subcortical nuclei by referencing representative studies[8,14].

## Reconstruction and visualization

Based on the 10-µm resolution annotation image, we used the marching cubes algorithm to systematically reconstruct the 3D morphology along the hierarchical nomenclature of the brain, ranging from fine nuclei to larger brain regions. The Laplacian smoothing algorithm provided by the Visualization Toolkit was used to simplify the surface models, preserving details while reducing the data volume and computational overhead for visualization[35]. Blender software (Blender Foundation) was then used to further compress the model dataset and convert it into an fbx format suitable for online presentation. Using the open-source framework Three.js, we established an online platform for browsing and interacting with the entire brain's 3D surface models. The Zoomify tool (Zoomify, Inc., www.zoomify.com) was used to enable real-time web browsing of coronal, sagittal and horizontal brain slice image sequences at 1-µm horizontal resolution. Furthermore, the neuroglancer framework (Google Inc., www.neuroglancer.org) was used to achieve real-time arbitrary-angle reslicing of the 3D image dataset with an isotropic 1-µm resolution.

## The brain-wide positioning system

**Defining intracranial datum marks.** We manually selected eight anatomical structures distributed throughout the entire brain of the normalized MOST-Nissl dataset used for atlas construction. Structures such as the anterior commissure, nucleus ambiguous, corpus callosum and facial nerve VIIn are also clearly visible in images of immunohistochemistry staining sections, including acetylcholinesterase staining, whereas the DGsg, lateral amygdalar and the medial geniculate complex are identifiable in the images acquired by MRI. On the basis of the reconstructed detailed 3D morphology, a total of 18 easily recognizable geometric feature points were further selected from these anatomical structures, such as the midpoint of the anterior commissure, and the posterior–dorsal endpoint of the dorsal raphe. The selection of these points considers the anterior–posterior, left–right and superior–inferior directions within the brain. The initial coordinates for each geometric feature point were determined on the basis of the reconstructed 3D models.

Using the 1-µm resolution MOST-Nissl dataset, local volumes were cropped around each feature point, with dimensions of 1,000 pixels for length, width and height. These volumes were imported into a custom-designed MATLAB (MathWorks, Inc.) App capable of

simultaneously generating 20-µm thickness projection images on coronal, sagittal and horizontal planes. Several experts with neuroanatomical knowledge observed and determined the precise coordinate positions of these feature points on these standard planes. The results from various experts were then consolidated to establish the intracranial datum marks.

**Defining cranial datum marks.** We down-sampled the acquired whole-head images from an adult male C57 mice to isotropic 20-µm resolution. These images were imported into Amira Software (Thermo Fisher Scientific) for volume rendering to determine the initial coordinates of the bregma and lambda points, which served as cranial datum marks. Subsequently, we cropped the surrounding images of these two points from the original resolution whole-head images. These cropped images were then placed into Amira Software for 3D reconstruction. On the basis of the reconstructed morphology of the skull sutures, the precise coordinates of the two points were determined.

**Calculating the spatial relationship among datum marks.** We used a nonlinear registration algorithm to register the full-head dataset to the MOST-Nissl dataset, obtaining a spatial mapping relationship between the two sets of images[13]. This mapping was then applied to the coordinates of intracranial datum marks, allowing us to establish the precise spatial relationship between intracranial and cranial datum marks.

**Constructing the brain-wide stereotaxic coordinate system.** The coordinate system was established using the length, width and height directions of the normalized MOST-Nissl dataset as the $x$, $y$ and $z$ axes, respectively. This coordinate system could be oriented based on surgical requirements, using any cranial or intracranial datum marks as the origin.

## Inter-atlas mapping and neuronal circuits mapping

**STAM and CCF.** We initially used only the brain outline as the landmark, registered the MOST-Nissl dataset onto the CCF for global correction, ensuring that its position and azimuth in 3D space kept consistent with the CCF. For more precise mapping, we selected several brain regions as anatomical landmarks to register the STAM onto CCF, using the BrainsMapi tool. Because of differences in brain-region definitions and boundaries between the two atlases, this mapping achieves a brain-region level precision with the average Dice score greater than 0.8, as shown in the 'Fine registration for inter-atlas mapping' column of Supplementary Table 6.

**STAM and WHS.** Using STAM's brain-wide positioning system, which offers rich reference data for image registration, we first selected five pairs of intracranial datum marks visible in both Nissl and MRI images to linearly align WHS with STAM. Then, several brain regions were used as landmarks to perform a nonlinear registration of WHS onto STAM by means of the BrainsMapi tool. This method can also be applied to establish spatial mapping relationships between STAM and other atlases.

**STAM and MBSC.** We integrated the delineation and nomenclature of MBSC into STAM, using supplementary data from ref. 19, which connect MBSC's nomenclature with that of the ARA. First, we used these data to construct a hierarchically organized MBSC nomenclature. We then extracted anatomical label slices of MBSC from these supplementary data, determined their spatial relationship to the corresponding CCF slices and accurately located them in STAM's coordinate system, which is initially registered to the CCF. These label slices were converted into vectorized borders and visualized in STAM's 2D viewer. Owing to the 100-µm interval of the MBSC slices, direct 3D reconstruction is challenging, so we show them at present in 2D, and only one in every five atlas levels in STAM corresponds to an MBSC atlas level.

**Neuronal circuits mapping.** Using the fMOST imaging technology[24], we can obtain datasets containing both neural circuit and propidium iodide imaging channels. The cytoarchitectural images from the propidium iodide channel provide anatomical features that allow us to establish a spatial mapping between neural circuit images and STAM. For single-neuron circuit datasets without propidium iodide images, the intrinsic fluorescence contours of cell-dense regions can be used to identify landmarks such as DGsg-mid, enabling spatial mapping with STAM. Moreover, the method of constructing precise spatial mapping between STAM and CCF can be generalized to other 3D mouse brain atlases.

## Nomenclature

The construction of STAM primarily used the nomenclature of the ARA and Brain maps v.4.0, with extra references to the MBSC for the delineation of certain subregions[6–8]. During the construction of STAM, we also annotated some new subregions and nuclei. For the newly defined subregions in the hippocampal area, we directly adopted the names provided in the literature, as these subregions follow the naming conventions of ARA. For instance, for the subregions identified in ACB, PAG and zona incerta, we adhered to the naming conventions of ARA, using the nucleus name followed by the lowercased abbreviation defining their locations. The naming of other newly defined brain areas and nuclei in STAM also follows the ARA naming conventions.

## Web service construction

**Open access to the MOST-Nissl dataset.** We converted the 1-µm resolution MOST-Nissl dataset into TDat format. The TDat-formatted image dataset is stored as multi-resolution image pyramid, and is diced into cubes with the same 256 voxels in each direction. We developed a server-side program cropping any local image from the TDat-formatted MOST-Nissl dataset, with given information about the desired spatial range and resolution. The program returned an organized 3D image in .tiff format to the client-side for downloading.

**Brain slice registration.** We selected two sets of propidium iodide- and DAPI-stained whole-brain datasets acquired by the fMOST system and nonlinearly registered them to the MOST-Nissl dataset. Using the registered data as a foundation, we generated a training set by extracting slices from different angles and positions. This training set was then input into a prediction network built on the SVRnet framework[36]. On the basis of the trained prediction network, we established an online registration web service for 2D brain slices to the 3D brain atlas. This service receives a single propidium iodide- or DAPI-stained brain slice image uploaded by the user. The image is input into the corresponding mode of the prediction network in the backend, where the prediction network calculates the slice parameters, computes the corresponding brain atlas slice and returns the result to the user. Simultaneously, the slice angle and position parameters are passed to the arbitrary-angle reslice browsing service, which returns 1-µm resolution MOST-Nissl dataset images at the same angle as the user's brain slice.

**Virtual surgery planning.** We used the 10-µm resolution annotation image of STAM to create a table, recording the list of brain structures that every 10-µm sampled point from the 3D space of STAM belonged to. Using this table, we then developed a server-side program calculating the brain structures that any given injection path passed through, and returned the result to the client-side for visualization. This program is used for both the manual and intelligent modes. We also developed a program detecting a path that avoided to pass through the user-given structures. This program randomly emitted rays from the user-assigned injection target, and decided whether any of these emitted rays did not penetrate the given structures. If no ray fitted the requirement, the program emitted new rays again until a qualified path was found or the time limit of calculation was hit. This program is used for intelligent mode only.

**Mutual query of neuroinformation.** After multi-type neuroinformation was registered onto the STAM, we computed the brain regions or nuclei where their key nodes were located. For each neuron morphology data, the key nodes are its soma, terminals and branching points, the coordinates of which are recorded in a .swc format file. On the basis of the calculated locations, we could construct a table describing the relationship between the brain structure list and neuroinformation list. We then developed a query service based on this table and integrated this service to the webpage.

As mentioned previously, all these web services are integrated into single entry-point. They are organized as different tabs in one page, facilitating fast switching between different services. Users can get familiar with these services both by reading the manuals and following the website tours provided on this webpage.

## Reporting summary

Further information on research design is available in the Nature Portfolio Reporting Summary linked to this article.

## Data availability

The 1-µm resolution MOST-Nissl dataset used to construct STAM, the labelling images and vectorized boundaries of brain structures, are available through https://atlas.brainsmatics.cn/STAM/. Readers can browse this link and find the desired way to navigate or download our data, or query Supplementary Table 3 to visit the specific gene-type neuron distribution datasets used for validating STAM, the comparison of the MOST-Nissl dataset with the Nissl staining sections from brainmaps.org and more results for Extended Data Figs. 5, 7 and 10. The ARA and CCF referred in this study can be accessed by https://atlas.brain-map.org. The WHS data were downloaded from https://www.nitrc.org/projects/incfwhsmouse. The brain-region labels of MBSC used in our coronal plane visualization service are available at Dryad (https://doi.org/10.5061/dryad.t1g1jwsxw)[37]. The ISH image data from Allen Institute used in this study can be accessed at https://mouse.brain-map.org/. The neuron morphology data used by STAM's neuronal connectivity web service include datasets from the Brain Image Library (https://www.brainimagelibrary.org/), under the following BIL ID, which is used as the identifier to query dataset at https://api.brainimagelibrary.org/web/: ace-ban-out, ace-ban-owl, ace-ban-own, ace-ban-pad, ace-ban-pal, ace-ban-pan, ace-ban-pay, ace-ban-pen, ace-ban-pet, ace-ban-pie, ace-ban-pig, war, wax, wet, ace-die-age, ace-ban-rig, who, ace-did-who, ace-add-vat, ace-add-vex, ace-ban-pot, ace-add-wag, ace-ban-pry, ace-ban-pun, ace-add-was, ace-ban-put, ace-add-web, ace-ban-ran, ace-ban-rat, ace-ban-raw, ace-ban-red, ace-ban-rid, win, wit, zoo, all, ace-zip, ace-ace, ace-act, ace-add, ace-age, ace-aim, ace-air, ace-and, ace-ant, ace-ape, ace-arm, ace-art, ace-ash, ace-ask, ace-ban-rip, ace-die-ant, ace-did-win, ace-ban, ace-bat, ace-bay, ace-bed, ace-bet, ace-bid, ace-big, ace-bin, ace-bit, ace-bog, ace-boo, ace-box, ace-bug, ace-bun, ace-bus and ace-cab. Source data are provided with this paper.

## Code availability

ImageJ (v.1.53k) is available at https://imagej.net/ij/download.html. The ANTs tool (v.2.1.0) is available at https://github.com/ANTsX/ANTs. The BrainsMapi tool (no version number available) is available upon request. The Visualization Toolkit (v.9.3.0) is available at https://github.com/Kitware/VTK. The Blender software (v.4.3) is available at https://www.blender.org/download/. The Three.js (v.0.157.0) is available at https://github.com/mrdoob/three.js. The Zoomify tool (Enterprise Developer 4) we purchased is no longer maintained by the producer

at https://zoomify.com/. The neuroglancer framework (v.2.29) is available at https://github.com/google/neuroglancer. MATLAB software (v.2023b) is a commercial production, and can be purchased at https://www.mathworks.com/products/matlab.html. Amira Software (v.6.1.1) is commercial software and can be purchased through https://www.thermofisher.com/. The SVRnet package (no version number available) is available at https://github.com/farrell236/SVRnet.

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

**Acknowledgements** The STAM was developed over a decade of work, beginning in the summer of 2014, when we first attempted to depict the distribution of brain structures on acquired Nissl-stained images. Throughout this 10-year effort, many individuals contributed to the development of the pipeline, data preparation and evaluation of results. We thank Z. Ding, Z. Wang, X. Liu, Z. Zhao, X. Peng, Q. Zhou, Y. Chen, L. Su, J. Li, H. Zhou, H. Lu, Z. Duan, X. Li, P. Luo, X. Zhang, C. Zhang, X. Zhang, C. Zhou, L. Deng, T. Luo, Y. Yu, L. Tan, Y. Jia, L. Liu, S. Cheng, W. Guo, G. Chen, R. Guo, Y. Shen, K. Bai, X. Hu, X. Song, D. Liu, T. Lei, S. Qin, S. Luo, R. Xiao, M. Yao, W. Sun, G. Fan, M. Zhang, L. Wei, Y. Cao, Z. Wang, J. Jin, T. Tang, Z. Xie, K. Zhang, T. Zhang, W. Li, W. Chen, Y. Luo, M. Liao, Y. Wang, S. Wang, J. Lu, Y. Wang, Y. Xiao, Y. Di, Q. Ye, Z. Wang and Y. Gu for their participation in this work. This work was supported by finance from the STI2030-Major Projects (grant nos. 2022ZD0205201, 2021ZD0201001, 2021ZD0201002 and 2021ZD0200203), National Natural Science Foundation of China grants (nos. 32192412, T2122015, 61890953 and 61721092) and grant no. NIHU19MH114821.

**Author contributions** H.G. and Q.L. conceived and supervised the project. X. Li, B.L., M.R. and S.C. prepared the brain samples. T.J. and X.J. acquired the imaging data needed. G.Y., Y. Li, Z.L., H.N., C.T., W.D. and W.S. developed the software. A.L., X. Li and H.G. managed the project. Z.F., Y. Luo, X. Liu, H.N., L.C., Q.Z., A.T., C.T., Z.X. and H.D. performed anatomical delineation. X. Chai, X. Lu, Z.W., M.Z., C.D., S.B. and J.Z. investigated and analysed the data. X. Chen, S.Y., Z.Z. and J.L. implemented web visualization. Z.F., X. Li, H.D., H.G. and Q.L. prepared and revised the manuscript.

**Competing interests** The authors declare no competing interests.

**Additional information**
**Correspondence and requests for materials** should be addressed to Hong-wei Dong, Hui Gong or Qingming Luo.

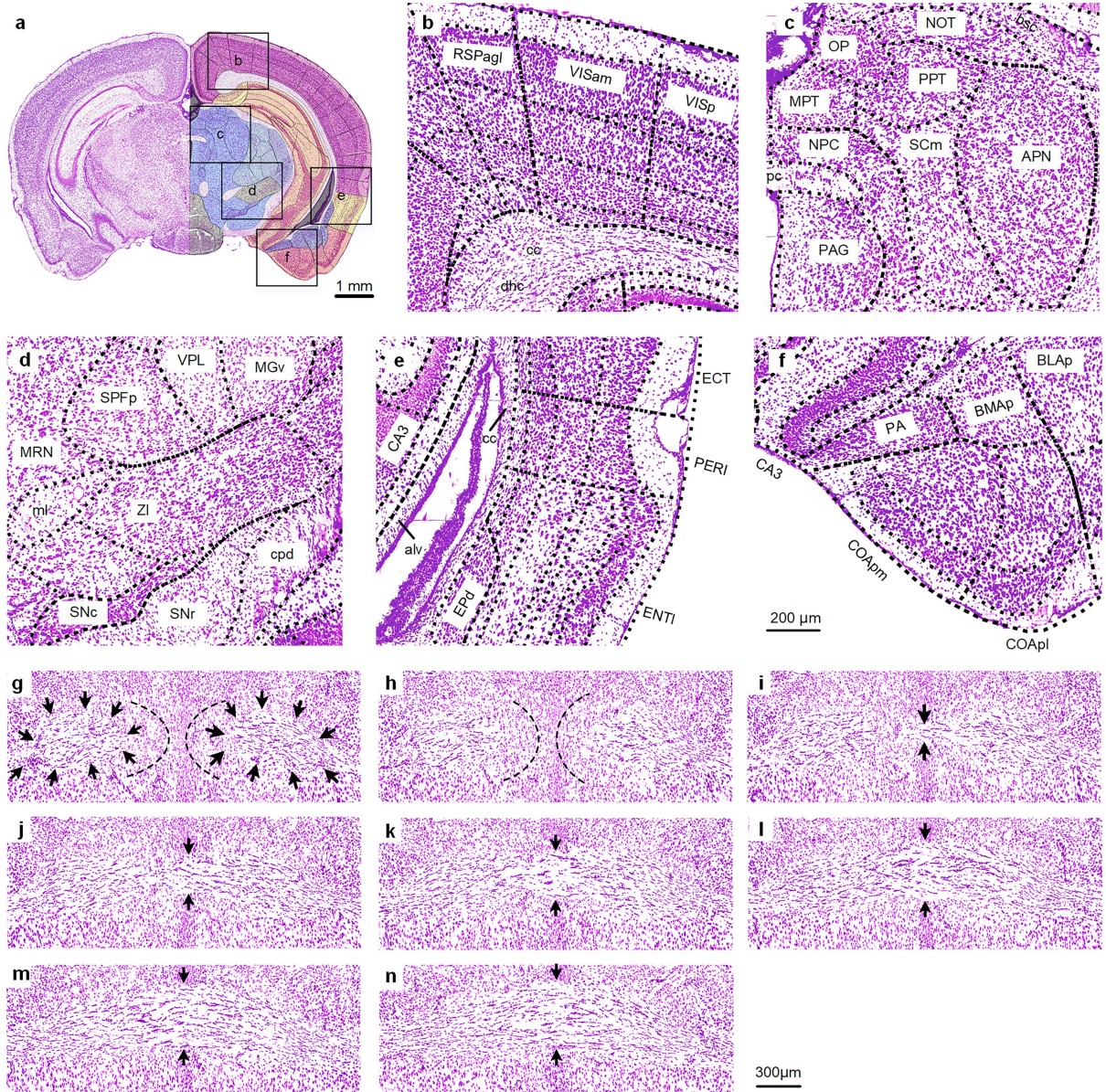

**Extended Data Fig. 1 | The cytoarchitecture details of the MOST-Nissl dataset. a**, A typical coronal image that includes the brain regions isocortex, TH, HY, MB, HPF, OLF, and CTXsp, each highlighted by black rectangles. **b-f**, Magnified views of the boxed areas in (**a**): isocortex and cc in (**b**); MB in (**c**); HY and TH in (**d**); HPF in (**e**); and CTXsp and OLF in (**f**). Dashed black lines mark regional boundaries. All images are 20 μm-thick projections from the MOST-Nissl dataset. Scale bar in (**f**) applies to (**b-f**). **g-n**, Eight consecutive coronal sections near the anterior commissure (ac), spaced at 1 μm intervals. Black arrows highlight the ac, and dashed lines delineate the medial tips of its bilateral branches. Each panel represents a 20 μm-thick projection. Scale bars are shown at lower right in each image. Scale bar located at the bottom right applies to (**g-n**). Full region names are listed in the supplementary information and apply throughout.

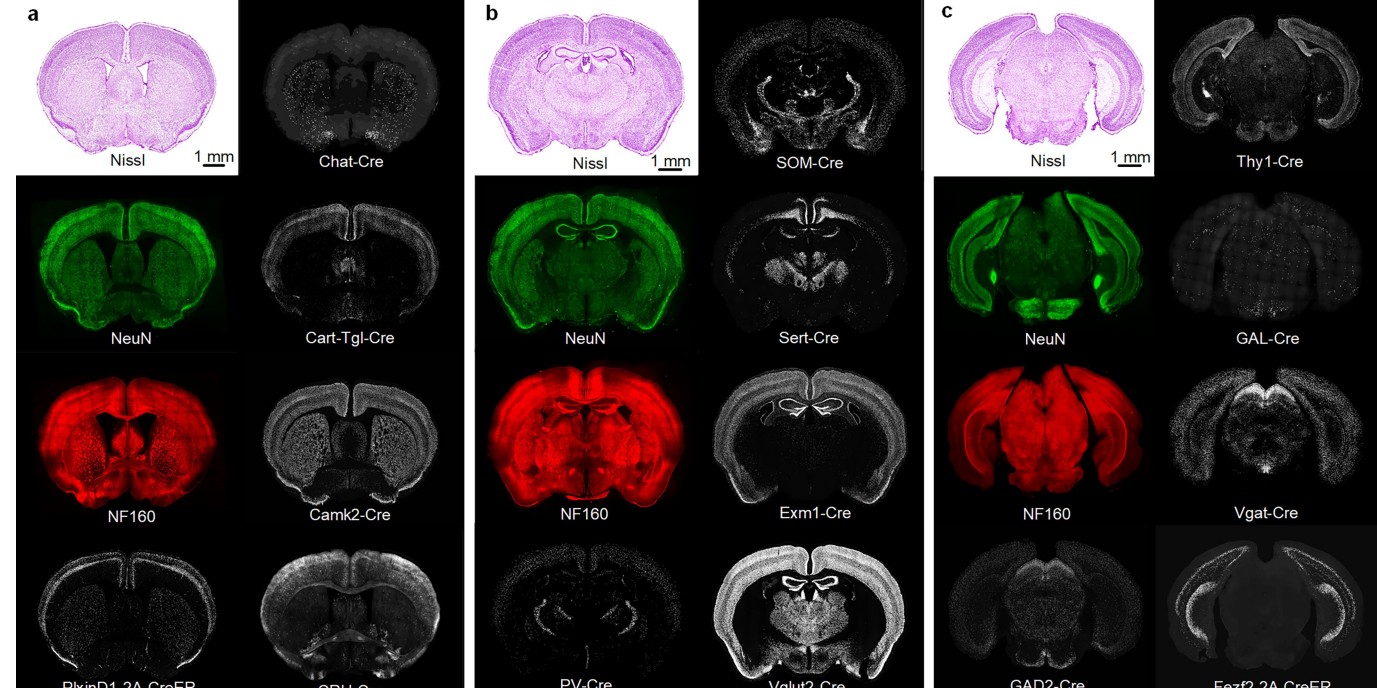

**Extended Data Fig. 2 | Overview of the datasets employed to illustrate STAM. a-c**, Three representative coronal planes of the atlas with corresponding source images. Magenta images are Nissl-stained, while green and red indicate immunohistochemical stains. Black-and-white panels show distributions of 15 genetically defined neuron types, including PlxinD1-2A-CreER, Chat-Cre, Cart-Tgl-Cre, Camk2-Cre, CRH-Cre, PV-Cre, SOM-Cre, Sert-Cre, Exm1-Cre,

Vglut2-Cre, GAD2-Cre, Thy1-Cre, GAL-Cre, Vgat-Cre, Fezf2-2A-CreER. Labels below each panel specify the staining method or transgenic line. The cytoarchitecture coronal sections shown in three groups is the representatives of n = 700 coronal sections with 20 μm-thick projection derived from the MOST-Nissl dataset. Scale bars are shown on the cytoarchitectural image of each panel, respectively.

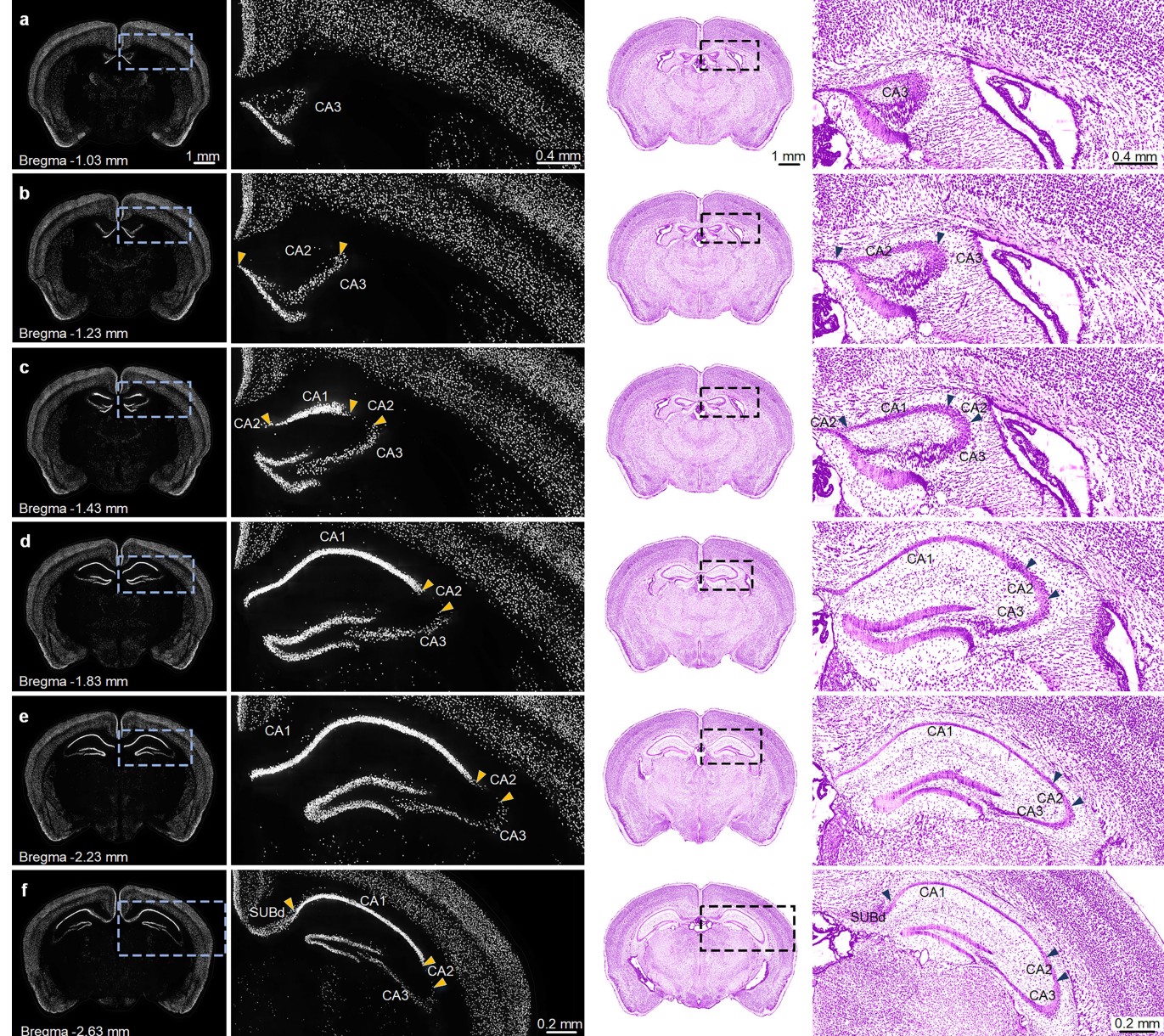

**Extended Data Fig. 3 | Use of Camk2-Cre neuron distribution to delineate CA subdivisions. a-f,** Coronal sections illustrating the boundaries between CA1, CA2, and CA3. In each panel, the first column shows full Camk2-Cre neuron distribution coronal images; the second column magnifies the CA region (light-blue dashed box), where yellow arrows mark natural boundaries based on neuronal density and pattern. The third column displays the corresponding Nissl-stained images; the fourth column shows magnified CA regions (black dashed box), with black arrows indicating boundary positions migrated from the neuron distribution images. Scale bars in the first and third columns of (**a**) apply to the same columns in (**a-f**); those in the second and fourth columns apply to (**a-e**).

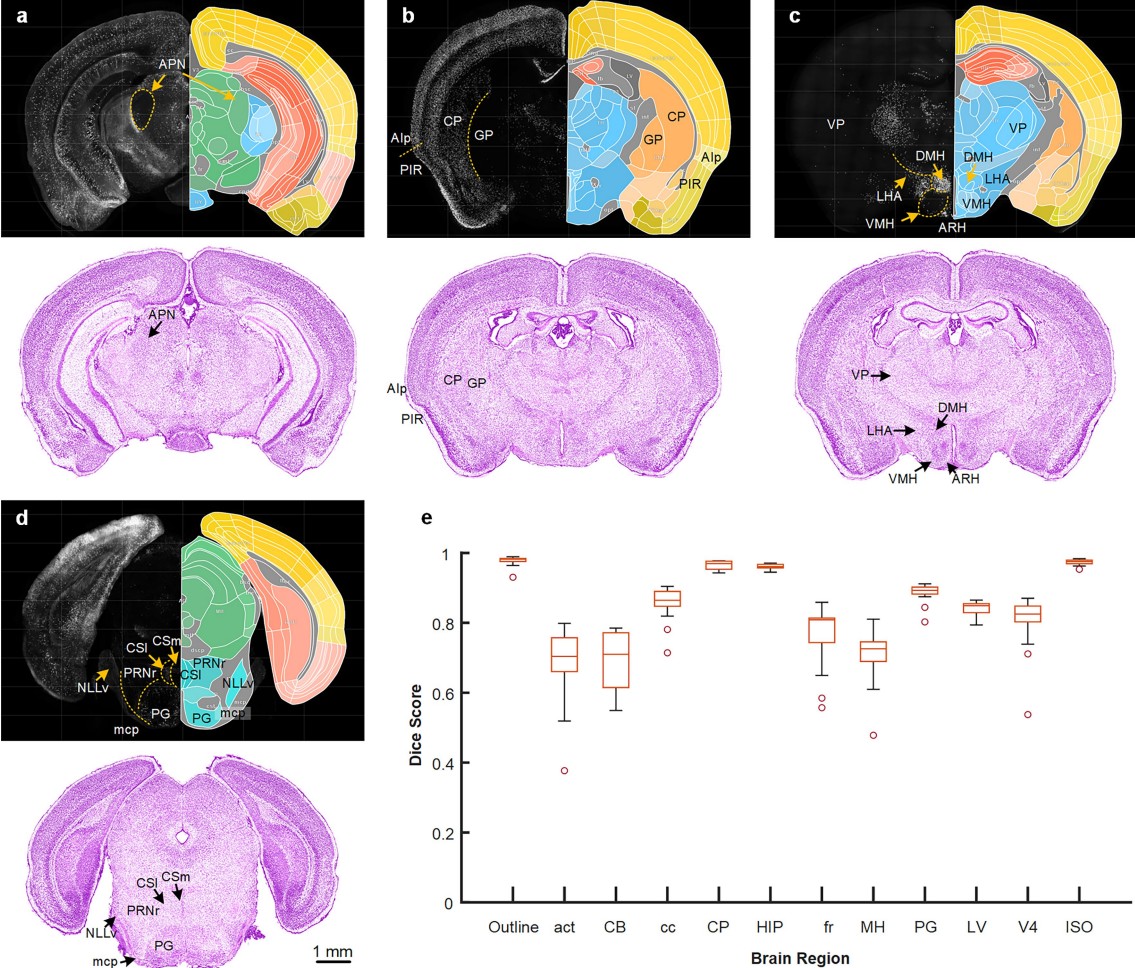

**Extended Data Fig. 4 | Specific gene-type neuron distributions aid structural delineation. a-d**, Representative coronal planes from SOM-Cre, Camk2-Cre, GAL-Cre and CRH-Cre datasets respectively. In each, the left half of the first row shows structure boundaries (yellow dashed lines) inferred from neuron distribution patterns, and the right half shows the corresponding STAM atlas delineation. The images on the second row are Nissl-stained sections at the same location, with the black texts as the migrated brain region identifications from the corresponding specific neuron distribution images. The left half image shown in each panel is the representative coronal images from one specific neuron distribution dataset. Scale bar applies to (**a-d**) and is located below (**d**). **e**, Quantitative evaluation of registration accuracy. Dice scores are shown as box plots for each brain region (X-axis), with Y-axis displaying the interquartile range (orange box), median (orange line), minimum/maximum values (whiskers), and outliers (orange circles). Note that not all brain regions shown in (**e**) are identifiable in all specific gene-type neuron distribution dataset (n, number of biologically independent datasets for evaluating the accuracy of registering certain brain structure. n = 22 for Outline, CB, cc, HIP, PG, LV and V4; n = 20 for act; n = 19 for fr and MH; n = 13 for ISO; n = 12 for CP).

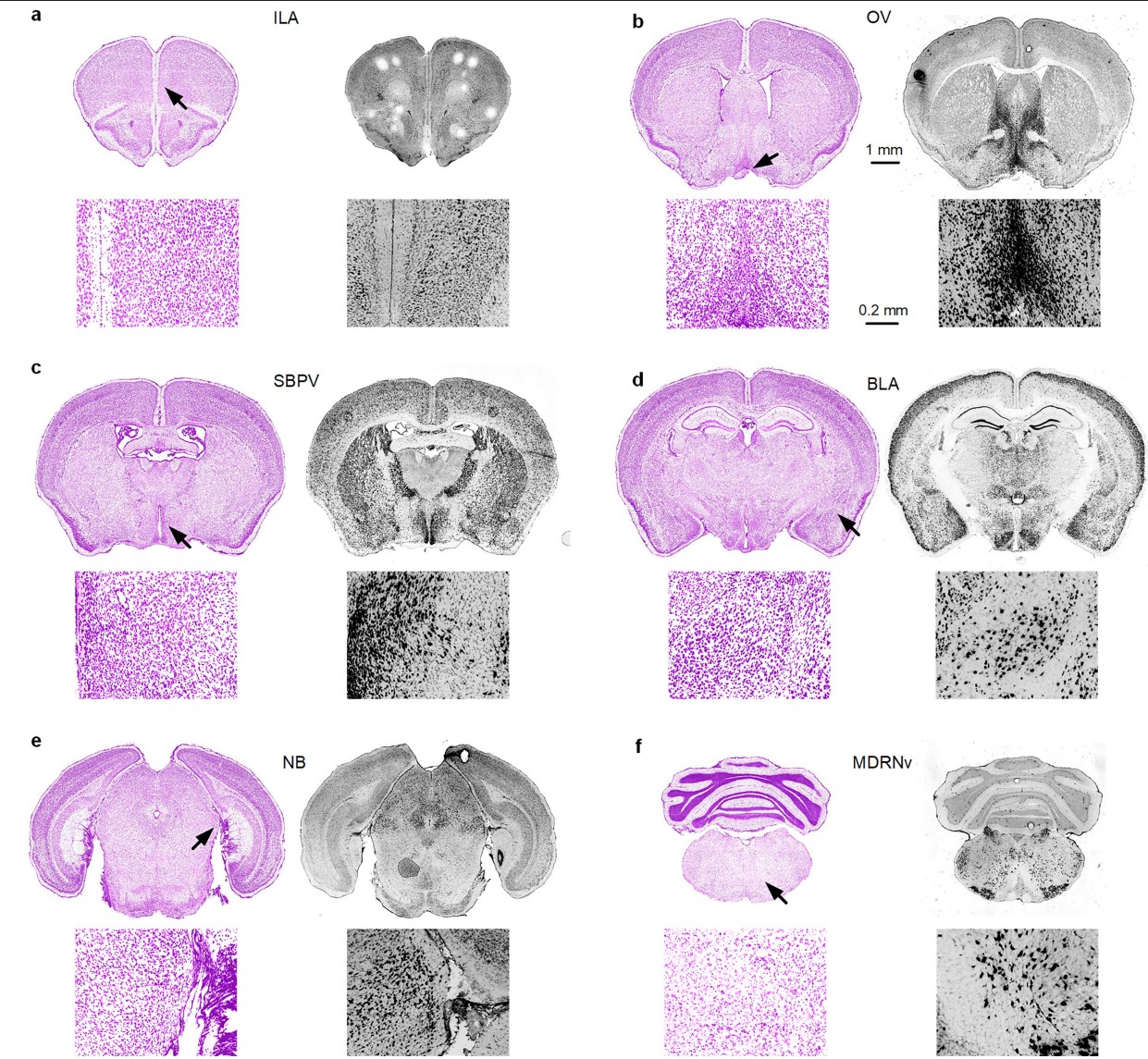

**Extended Data Fig. 5 | Delineation of anatomical structures using in situ hybridization (ISH) images. a-f**, ISH images provide structural clues for delineating anatomical regions. For each panel, the top left shows a Nissl-stained coronal plane with the target structure, and the top right is a corresponding ISH image. Black arrows mark the structure of interest. The bottom left and bottom right present magnified views of the region of interest indicated on the Nissl and the corresponding ISH images, respectively. ISH images from top right and bottom right of (**a**) adapted from the Allen Mouse Brain Atlas: mouse. brain-map.org/experiment/show/70562124. For the ISH images of other panels, links can be retrieved using the Gene Symbol and Probe Name provided in Supplementary Table 1. Scale bar is shown in (**b**) and applies to all panels. Additional results can be accessed via the link provided in the Data availability section.

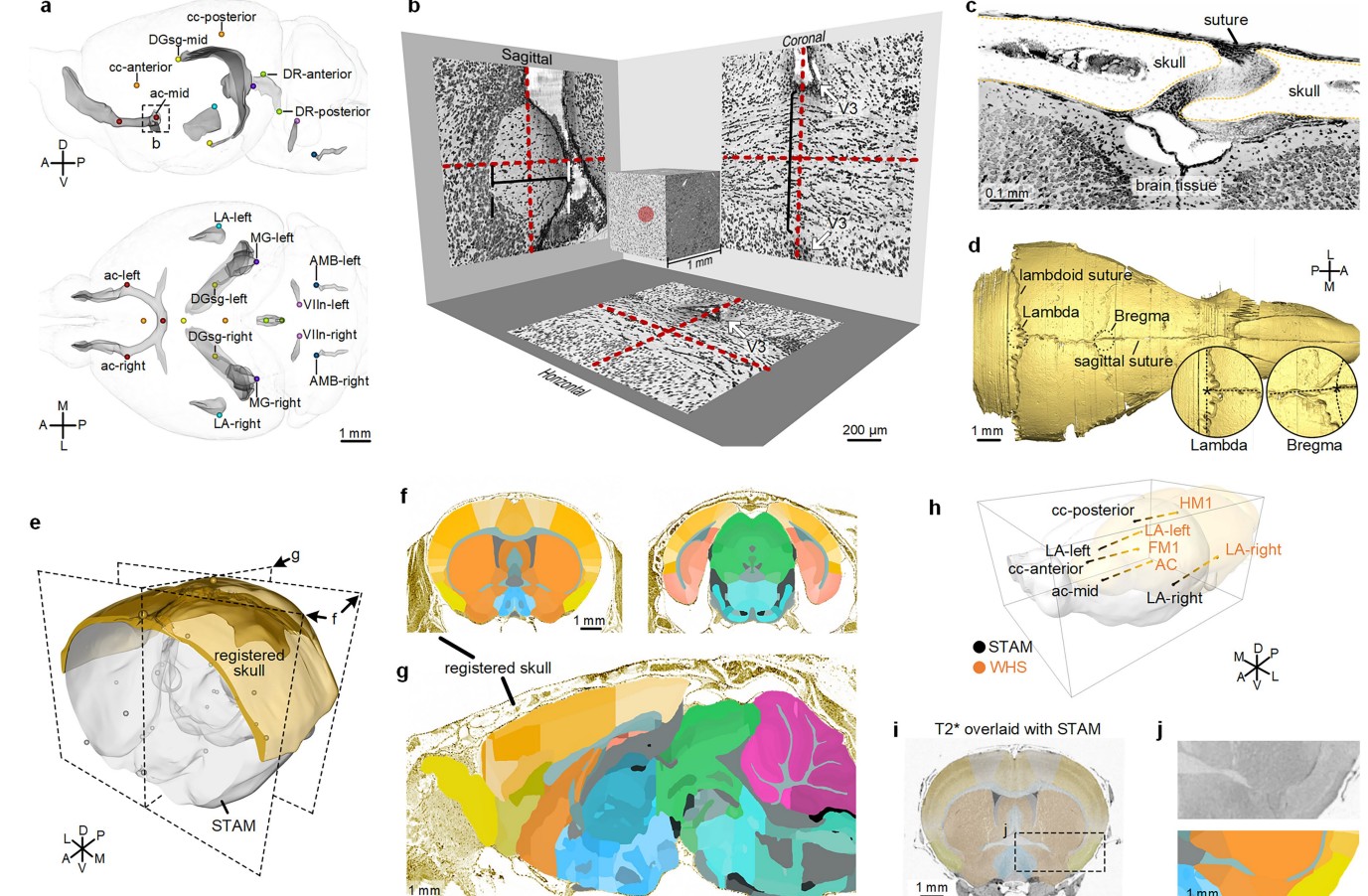

**Extended Data Fig. 6 | The datum marks of STAM. a**, Intracranial datum marks shown on sagittal and horizontal views, overlaid on a transparent outline. Datum marks defined by different anatomical features are color-coded; key brain structures rendered in dark gray. **b**, Construction of datum mark ac-mid, derived from the center of the anterior commissure (ac). Cross-sectional images from a 1000×1000×1000 voxel block with auxiliary lines and red crosshairs indicating ac-mid. Red dot, the defined ac-mid. **c**, zooms-in to a local region of MOST-acquired whole head dataset used to define cranial datum marks; image inverted for clarity. Yellow dashed lines enclose skull. **d**, 3D-reconstructed skull with Bregma and Lambda marked (dashed circles), and enlarged views showing anatomical guidelines. **e**, Registration of the head dataset to STAM. Transparent gray model: STAM brain; gold: skull. Grey and gold dots indicate registered intracranial and cranial datum marks, respectively. **f, g** Coronal and sagittal slices showing alignment of STAM (color-labeled) and registered head data (golden background). The locations of the three planes presented by the dashed lines in (**e**). **h**, Correspondence between STAM and WHS datum marks: STAM (gray model, black dots) and WHS (light-yellow model, orange dots). Dashed lines show paired marks for cross-atlas registration. FM1, frontal middle 1; AC, anterior commissure, WHS origin; HM1, hippocampus middle 1. These datum marks are defined by WHS, while the remaining ones are defined by STAM and can be queried in Supplementary Table 4. **i**, Registration of WHS onto STAM showing STAM region labels overlaid on a coronal T2* MRI slice from WHS. **j**, Magnified view of the same region, comparing the STAM label image (below) with the registered WHS T2* image (above). In **i** and **j**, the outline and planes of WHS were derived from NITRC atlas data (https://www.nitrc.org/projects/incfwhsmouse) and are adapted from ref. 22, PLOS, under a Creative Commons licence CC BY 4.0.

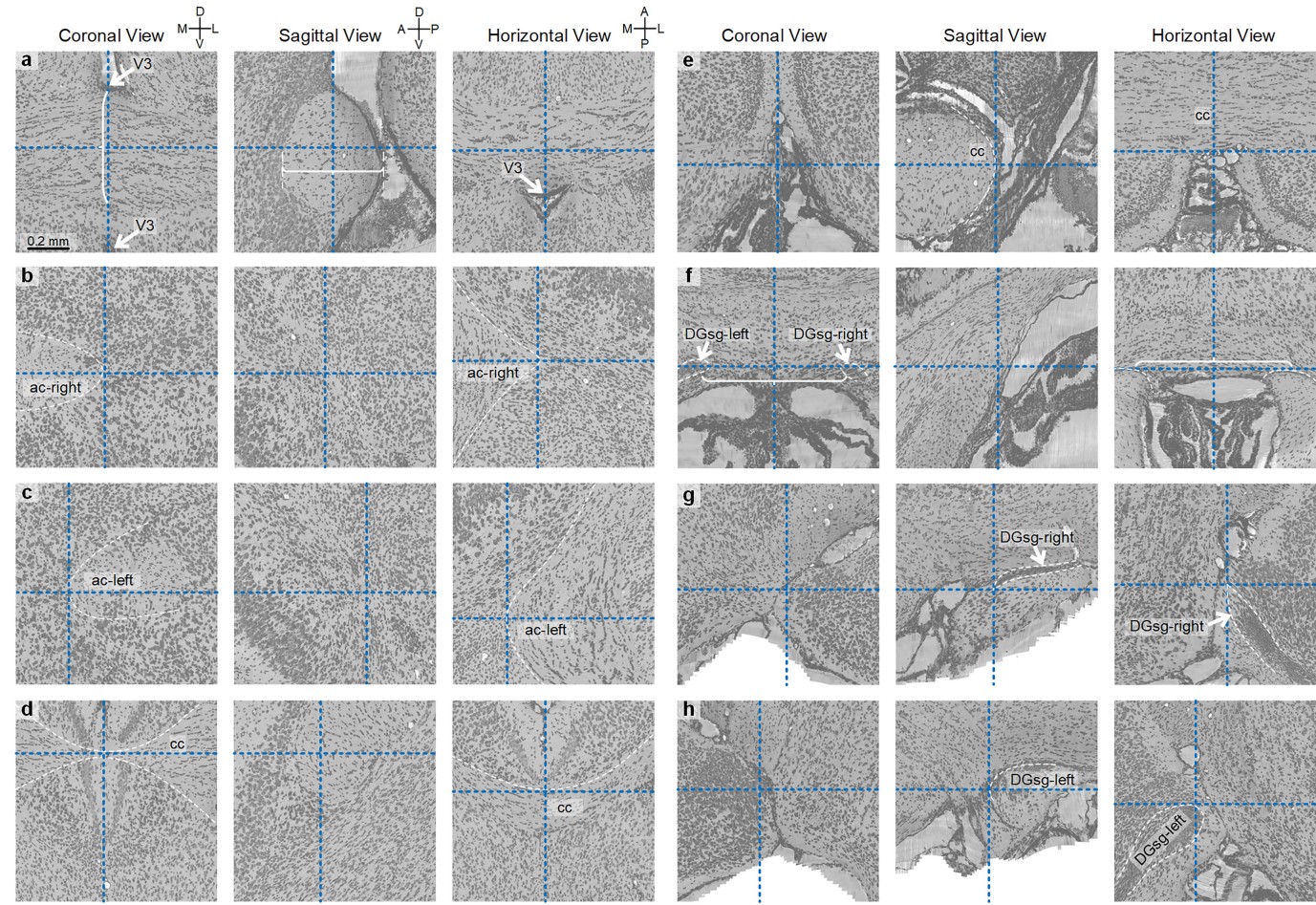

**Extended Data Fig. 7 | Cytoarchitectural criteria for defining intracranial datum marks of STAM. a-h,** Cytoarchitectural features of 8 intracranial datum marks displayed on coronal, sagittal, and horizontal planes. (**a**) ac-mid, (**b**) ac-right, (**c**) ac-left, (**d**) cc-anterior, (**e**) cc-posterior, (**f**) DGsg-mid, (**g**) DGsg-right, (**h**) DGsg-left. Blue dashed lines indicate the precise positions of each datum mark in the respective anatomical planes. In selected panels, white dashed lines outline relevant structural boundaries to aid identification. Scale bar in (**a**) applies to the images of all panels. The cytoarchitecture images of all 18 datum marks can be accessed via the link provided in the Data availability section.

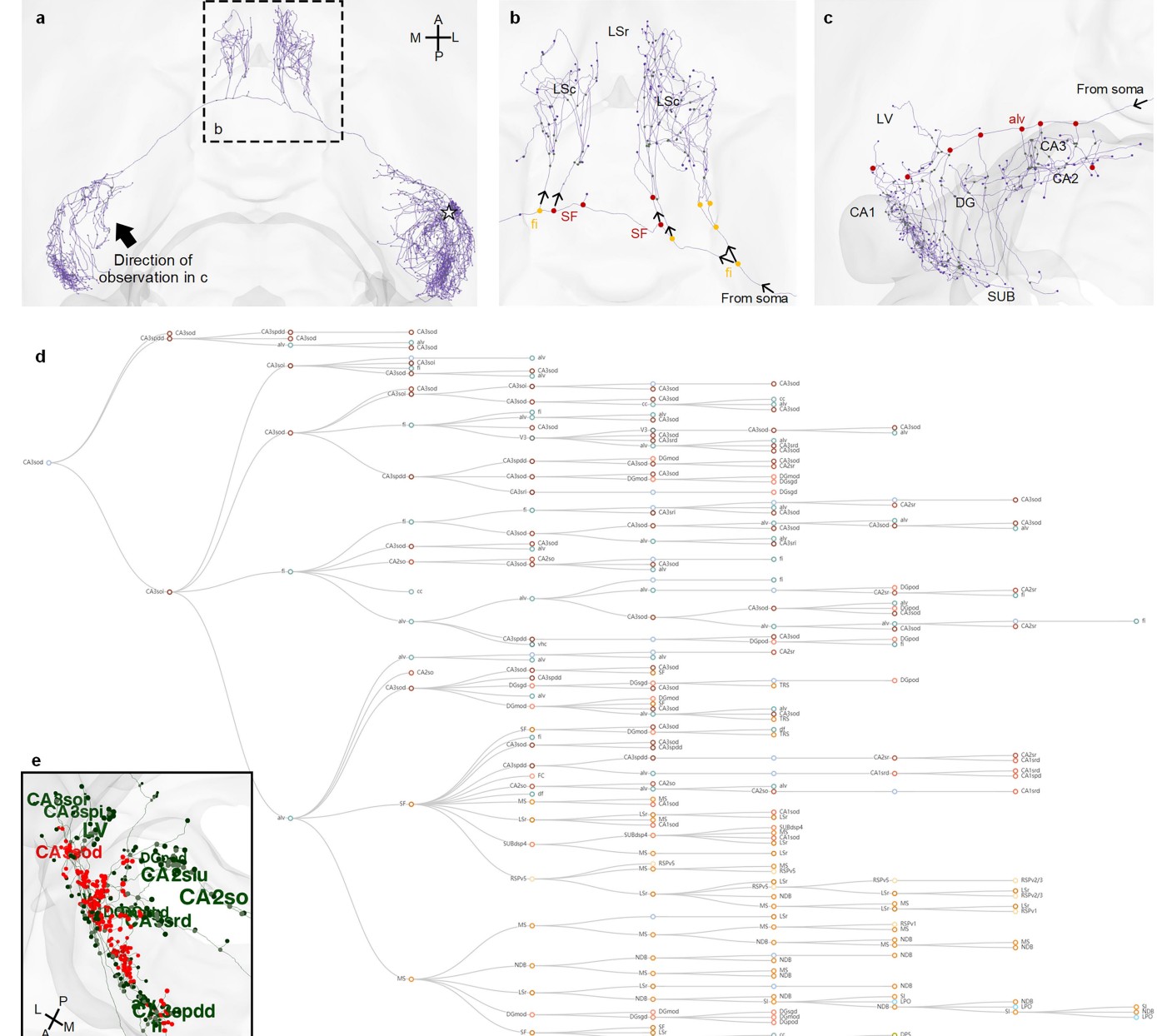

**Extended Data Fig. 8 | Localization of branching points and projection pattern analysis of a single neuron in STAM. a**, Magnified view of the neuron shown in Fig. 5e, with the soma marked by a white star. **b**, Zoom-in of the dashed box in (**a**), highlighting branching points in fi and SF (yellow and red dots, respectively). Black arrows indicate neuron projection directions; black text labels mark projection targets. Same applies hereafter. **c**, View along the observing direction indicated by the arrow in (**a**), showing red branching points in alv. **d**, Diagram of the neuron's projection pattern. The leftmost circle represents the soma's location; other circles indicate target or intermediate structures with branching points. Lines represent projection paths and their directionality. **e**, Snapshot of the STAM neuronal circuit viewer, highlighting terminals and branching points located in CA3sod in red, and the remaining colored in green.

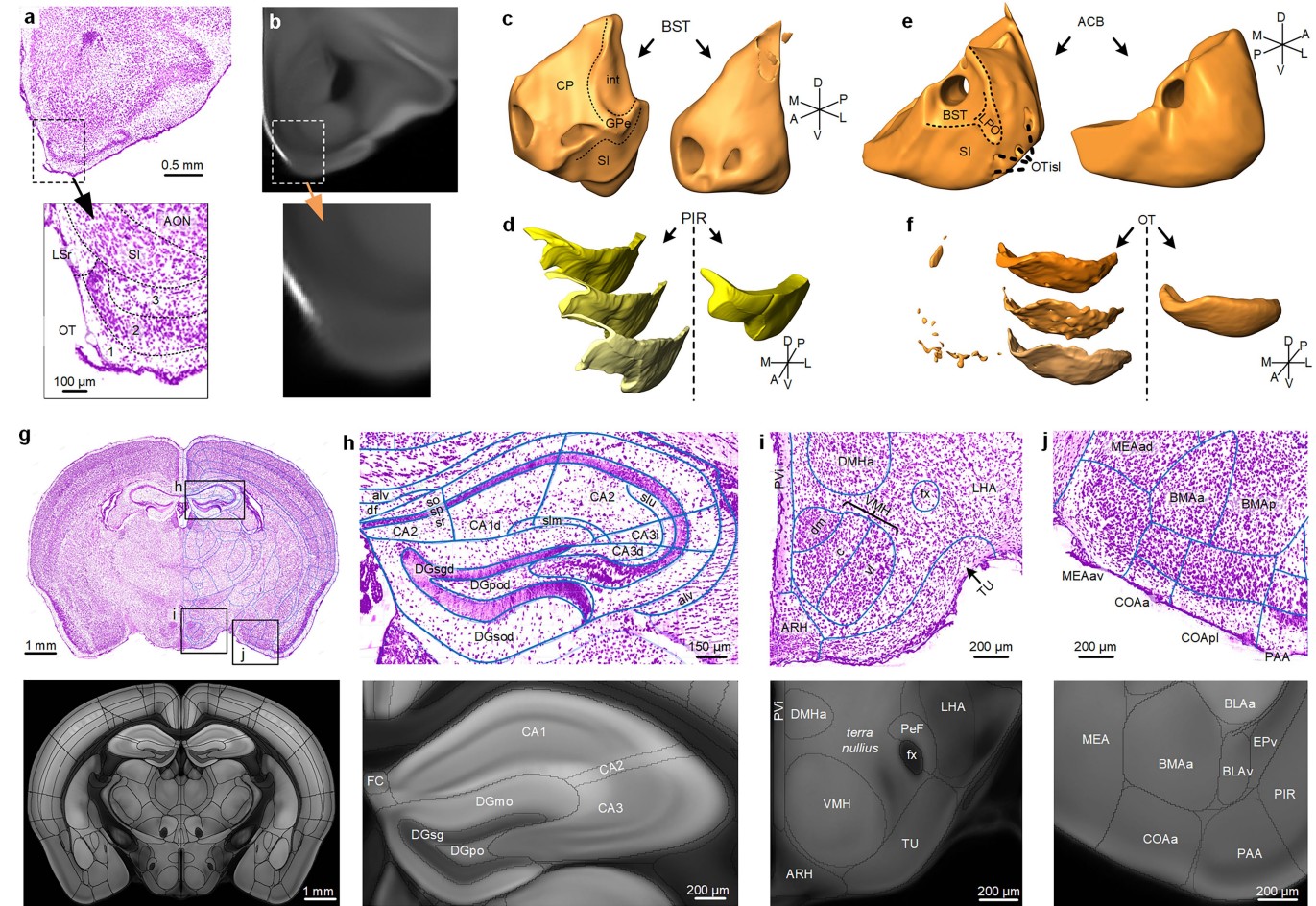

**Extended Data Fig. 9 | Comparison between STAM and CCF. a-b**, Top: magnified Nissl-stained coronal section from STAM (**a**) and corresponding average coronal section from CCF (**b**), both centered on the caudal nucleus (CNU). Bottom: further magnification of the boxed areas on the top row, with dashed lines outlining anatomical boundaries. **c-f**, 3D reconstructions of BST, PIR, ACB, and OT from STAM (left) and CCF (right). Vertical dashed lines in (**d**) and (**f**) separate the two atlases; labels indicate adjacent structures, with dashed boundaries marking separations. **g**, Coronal section comparisons between STAM and CCF, overlaid with delineated structure outlines (blue boundaries for STAM and black boundaries form CCF). **h-j**, Enlarged regions from boxed areas in (**g**),

showing detailed annotations of anatomical subdivisions. To avoid overcrowding, abbreviations for substructures (e.g., CA layers, hypothalamic, and amygdalar nuclei) are shown without their parent structure names. The word '*terra nullius*' in the bottom row of (**i**) indicates unlabeled regions in CCF. The original images and boundaries are snapshotted from the online visualization services of STAM and CCF respectively, while the texts are annotated manually. The boundaries on the coronal sections of STAM are bolded for better observations. Panel (**b**) and (**g-j**) contain images adapted from the Allen Mouse Brain Atlas, atlas.brain-map.org. Scale bars: (**a**) applies to (**a-b**); (**g-j**) beneath each panel. Orientation markers in (**c-f**) respectively.

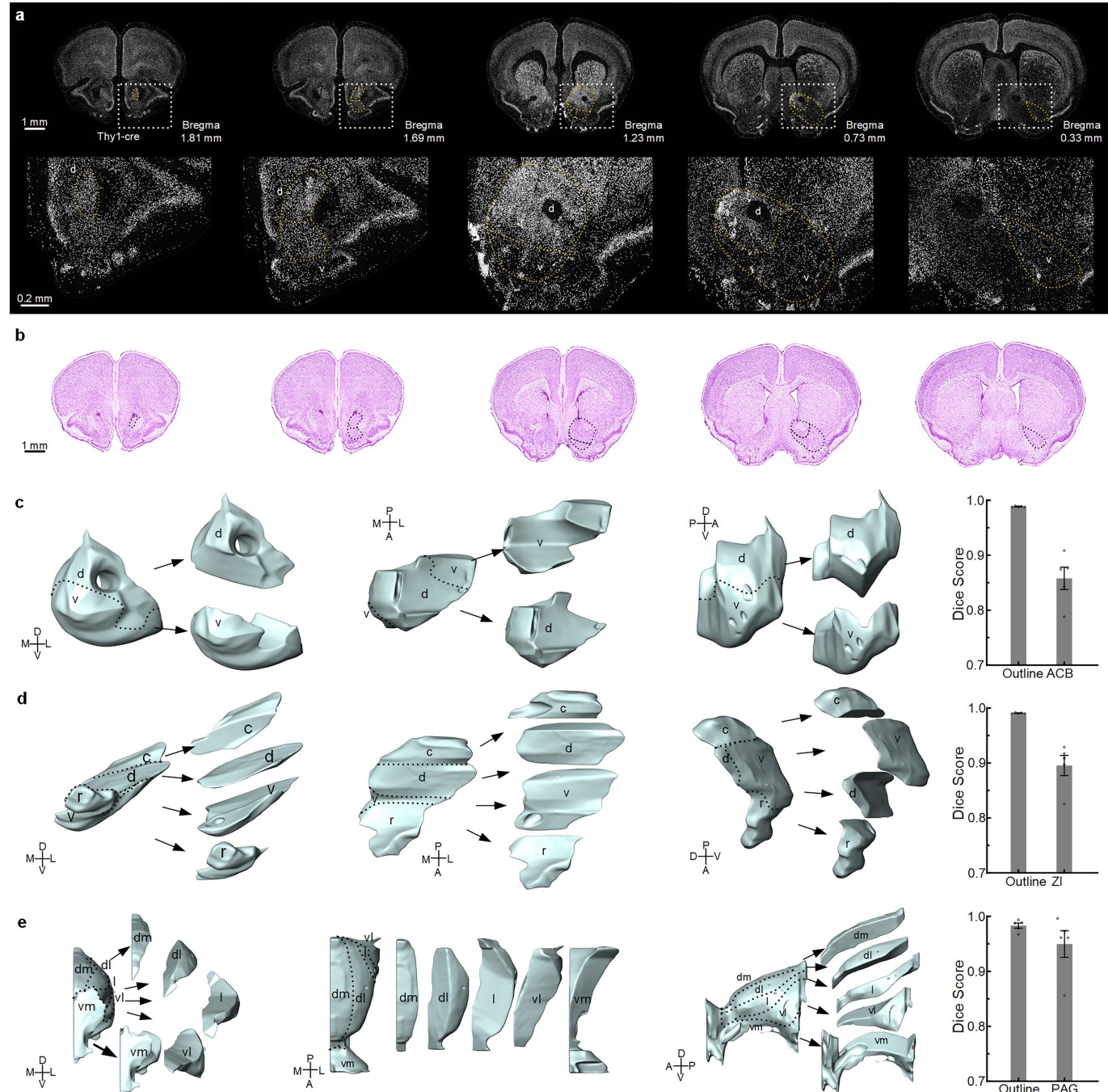

**Extended Data Fig. 10 | Subdivision and validation of newly delineated nuclei in STAM. a-c**, The delineation and reconstruction of subdivisions for ACB based on Thy1-Cre datasets. (**a**) Structural features used for delineation across coronal sections; second row shows magnified regions of the boxed areas on the top row, highlighting key structural distinctions. (**b**) Corresponding Nissl-stained images at the same location for comparison, with black dashed line migrated from the first row of (**a**). (**c**) 3D reconstructions of the newly defined ACB subdivisions with bar plot evaluating registration accuracy. d and v correspond to the subdivided dorsal and ventral parts of ACB.

**d-e**, 3D reconstructions of the newly defined ZI and PAG subdivisions, with bar plot evaluating registration accuracy. d, v, r and c in (**d**) correspond to the subdivided dorsal, ventral, rostral and caudal parts of ZI. dm, dl, l, vm and vl in (**e**) correspond to the subdivided dorsomedial, dorsolateral, lateral, ventromedial and ventrolateral parts of PAG. Bar plots show Dice scores for each region (X-axis), with mean ± s.e.m. (Y-axis, range 0 to 1), assessing registration accuracy (from n = 5 biologically independent coronal slices). Additional results can be accessed via the link provided in the Data availability section.

# Reporting Summary

## Statistics

For all statistical analyses, confirm that the following items are present in the figure legend, table legend, main text, or Methods section.

| n/a | Confirmed | |
|---|---|---|
| ☐ | ☒ | The exact sample size (*n*) for each experimental group/condition, given as a discrete number and unit of measurement |
| ☐ | ☒ | A statement on whether measurements were taken from distinct samples or whether the same sample was measured repeatedly |
| ☒ | ☐ | The statistical test(s) used AND whether they are one- or two-sided<br>*Only common tests should be described solely by name; describe more complex techniques in the Methods section.* |
| ☒ | ☐ | A description of all covariates tested |
| ☒ | ☐ | A description of any assumptions or corrections, such as tests of normality and adjustment for multiple comparisons |
| ☐ | ☒ | A full description of the statistical parameters including central tendency (e.g. means) or other basic estimates (e.g. regression coefficient) AND variation (e.g. standard deviation) or associated estimates of uncertainty (e.g. confidence intervals) |
| ☒ | ☐ | For null hypothesis testing, the test statistic (e.g. *F*, *t*, *r*) with confidence intervals, effect sizes, degrees of freedom and *P* value noted<br>*Give P values as exact values whenever suitable.* |
| ☒ | ☐ | For Bayesian analysis, information on the choice of priors and Markov chain Monte Carlo settings |
| ☒ | ☐ | For hierarchical and complex designs, identification of the appropriate level for tests and full reporting of outcomes |
| ☒ | ☐ | Estimates of effect sizes (e.g. Cohen's *d*, Pearson's *r*), indicating how they were calculated |

*Our web collection on statistics for biologists contains articles on many of the points above.*

## Software and code

Policy information about availability of computer code

| | |
|---|---|
| Data collection | No software for collecting data is used for this study. |
| Data analysis | The ImageJ (v 1.53k) is available at https://imagej.net/ij/download.html.<br>The ANTs tool (v 2.1.0) is available at https://github.com/ANTsX/ANTs.<br>The BrainsMapi tool (No version number available) is available upon request.<br>The Visualization Toolkit (v 9.3.0) is available at https://github.com/Kitware/VTK.<br>The Blender software (v 4.3) is available at https://www.blender.org/download/.<br>The Three.js (v 0.157.0) is available at https://github.com/mrdoob/three.js.<br>The Zoomify tool (Enterprise Developer 4) we purchased is no longer maintained by the producer https://zoomify.com/.<br>The neuroglancer framework (v 2.29) is available at https://github.com/google/neuroglancer.<br>The Matlab software (v 2023b) is a commercial production, and can be purchased at https://www.mathworks.com/products/matlab.html.<br>The Amira Software (v 6.1.1) is a commercial software, and can be purchased through https://www.thermofisher.com/.<br>The SVRnet package (No version number available) is available at https://github.com/farrell236/SVRnet. |

For manuscripts utilizing custom algorithms or software that are central to the research but not yet described in published literature, software must be made available to editors and reviewers. We strongly encourage code deposition in a community repository (e.g. GitHub). See the Nature Portfolio guidelines for submitting code & software for further information.

# Data

Policy information about availability of data

All manuscripts must include a data availability statement. This statement should provide the following information, where applicable:

- Accession codes, unique identifiers, or web links for publicly available datasets
- A description of any restrictions on data availability
- For clinical datasets or third party data, please ensure that the statement adheres to our policy

The 1 μm resolution MOST-Nissl dataset used to construct STAM, the labeling images, and vectorized boundaries of brain structures, are available through https://atlas.brainsmatics.cn/STAM/. Readers can browse this link and find the desired way to navigate or download our data, or query Supplementary Table 3 to visit the specific gene-type neuron distribution datasets used for validating STAM, the comparison of the MOST-Nissl dataset with the Nissl-staining sections from brainmaps.org, and more results for Extended Data Fig. 5, Extended Data Fig. 7, and Extended Data Fig. 10. The ARA and CCF referred in this study can be accessed by https://atlas.brain-map.org. The WHS data were downloaded from https://www.nitrc.org/projects/incfwhsmouse. The brain region labels of MBSC used in our coronal plane visualization service are obtained through https://datadryad.org/dataset/doi:10.5061/dryad.t1g1jwsxw. The ISH image data from Allen Institute used in this study can be accessed by https://mouse.brain-map.org/. The neuron morphology data used by STAM's neuronal connectivity web service include datasets from Brain Image Library (https://www.brainimagelibrary.org/), under the following BIL ID, which is used as the identifier to query dataset at https://api.brainimagelibrary.org/web/: ace-ban-out, ace-ban-owl, ace-ban-own, ace-ban-pad, ace-ban-pal, ace-ban-pan, ace-ban-pay, ace-ban-pen, ace-ban-pet, ace-ban-pie, ace-ban-pig, war, wax, wet, ace-die-age, ace-ban-rig, who, ace-did-who, ace-add-vat, ace-add-vex, ace-ban-pot, ace-add-wag, ace-ban-pry, ace-ban-pun, ace-add-was, ace-ban-put, ace-add-web, ace-ban-ran, ace-ban-rat, ace-ban-raw, ace-ban-red, ace-ban-rid, win, wit, zoo, all, ace-zip, ace-ace, ace-act, ace-add, ace-age, ace-aim, ace-air, ace-and, ace-ant, ace-ape, ace-arm, ace-art, ace-ash, ace-ask, ace-ban-rip, ace-die-ant, ace-did-win, ace-ban, ace-bat, ace-bay, ace-bed, ace-bet, ace-bid, ace-big, ace-bin, ace-bit, ace-bog, ace-boo, ace-box, ace-bug, ace-bun, ace-bus, ace-cab.

# Research involving human participants, their data, or biological material

Policy information about studies with human participants or human data. See also policy information about sex, gender (identity/presentation), and sexual orientation and race, ethnicity and racism.

| | |
|---|---|
| Reporting on sex and gender | Not applicable |
| Reporting on race, ethnicity, or other socially relevant groupings | Not applicable |
| Population characteristics | Not applicable |
| Recruitment | Not applicable |
| Ethics oversight | Not applicable |

Note that full information on the approval of the study protocol must also be provided in the manuscript.

# Field-specific reporting

Please select the one below that is the best fit for your research. If you are not sure, read the appropriate sections before making your selection.

☒ Life sciences ☐ Behavioural & social sciences ☐ Ecological, evolutionary & environmental sciences

For a reference copy of the document with all sections, see nature.com/documents/nr-reporting-summary-flat.pdf

# Life sciences study design

All studies must disclose on these points even when the disclosure is negative.

| | |
|---|---|
| Sample size | No formal statistical method was used to predetermine sample size. The sample sizes were chosen based on established practices in atlas development, where a single, carefully prepared sample is used as the 'basis' to ensure consistent cytoarchitecture information, assisted by auxilary image modalities to supplement extra details for structure delineation. Specifically, 3 adult C57BL/6J mice were used: one for generating the whole-brain Nissl-stained image dataset for atlas construction (the 'basis' sample), one for immunohistochemical labeling, and one for obtaining the whole-head sample used to define cranial landmarks. In addition, 22 transgenic mice were used to obtain specific neuronal distribution patterns. These sample sizes are considered sufficient and appropriate for the structural delineation purpose, as this study focuses on the construction of a brain atlas, which typically relies on high-quality representative samples rather than group comparisons or hypothesis testing. |
| Data exclusions | No data were excluded from the analyses. |
| Replication | This study focuses on the construction of a high-resolution mouse brain atlas based on Nissl-stained cytoarchitectural imaging and is supported by various auxiliary imaging modalities. Given the structural and anatomical nature of the data, the findings are not subject to experimental variability in the traditional sense. The protocol of this study follows established practices in stereotaxic atlas development, therefore, we confirm no findings were identified that could not be reproduced. |

| Randomization | This study does not involve allocation of animals into experimental groups, as it focuses on brain atlas construction based on high-resolution structural imaging. All samples were used for specific and predetermined purposes (e.g., one for Nissl-stained imaging, one for immunohistochemistry, one for intact head imaging, and 22 transgenic mice for specific neuron labeling). Therefore, randomization and control of covariates were not applicable in this context. |
| --- | --- |
| Blinding | Blinding was not relevant to this study. The research did not involve group allocation, treatment comparisons, or outcome assessments subject to observer bias. All imaging data were acquired and analyzed based on predetermined anatomical and structural criteria for brain atlas construction, ensuring objectivity in data interpretation. |

# Reporting for specific materials, systems and methods

We require information from authors about some types of materials, experimental systems and methods used in many studies. Here, indicate whether each material, system or method listed is relevant to your study. If you are not sure if a list item applies to your research, read the appropriate section before selecting a response.

## Materials & experimental systems

| n/a | Involved in the study |
| --- | --- |
| ☐ | ☒ Antibodies |
| ☒ | ☐ Eukaryotic cell lines |
| ☒ | ☐ Palaeontology and archaeology |
| ☐ | ☒ Animals and other organisms |
| ☒ | ☐ Clinical data |
| ☒ | ☐ Dual use research of concern |
| ☒ | ☐ Plants |

## Methods

| n/a | Involved in the study |
| --- | --- |
| ☒ | ☐ ChIP-seq |
| ☒ | ☐ Flow cytometry |
| ☒ | ☐ MRI-based neuroimaging |

## Antibodies

| Antibodies used | For the immunohistochemistry, the mouse brain was sectioned at 70 µm thickness by a vibratome (Leica, VS1200S). the slices were washed with PBS, blocked with 5% bovine serum albumin, and then incubated with the primary antibodies (anti- NeuN, mouse, Covance, SIG-39860, 1:1000 dillution; anti- NF160, rabbit, Abcam, ab64300, 1:1000 dillution) overnight at 4 °C. After washing with PBS, the secondary antibodies, namely Alexa Fluor 488 goat anti-mouse immunoglobulin G (IgG) (Invitrogen, Carlsbad, CA, United States, A11029, 1:1,000 dilution), and Alexa Fluor 594 goat anti-rabbit IgG (Invitrogen, Carlsbad, CA, United States, A11037, 1:1000 dilution) were applied for 2 hours at room temperature. Imaging was done afterwards with a multichannel fluorescence slide microscope (Olympus VS120, Tokyo, Japan). |
| --- | --- |
| Validation | These are all well characterized commercial antibodies.The specificity of the primary and secondary antibodies was validated by the manufacturers.<br>anti-NeuN: https://www.antibodypedia.com/gene/32645/RBFOX3/antibody/1457501/SIG-39860<br>anti-NF160: https://www.labome.com/product/Abcam/ab64300.html |

## Animals and other research organisms

Policy information about studies involving animals; ARRIVE guidelines recommended for reporting animal research, and Sex and Gender in Research

| Laboratory animals | 3 C57BL/6J mice were used, 1 for Nissl staining, 1 for immunohistochemistry, 1 for acquiring the whole head sample. Another 22 transgenetic mice were used for acquiring specific gene-type neuron distribution images. The C57BL/6J mice were all 8-week old, while the age of 22 transgenetic mice varies from 6-week to 36-week. The details can be queried in the Supplementary Table 2. |
| --- | --- |
| Wild animals | The study did not involve wild animals. |
| Reporting on sex | The findings apply to both male and female mice. The information about sex is reported in Supplementary Table 2. For the 3 mice used to construct our atlas, the sex is male. For the mouse samples used to collect the neuron distribution datasets with specific gene types, there are 8 male samples and 14 female samples. |
| Field-collected samples | The study did not involve sample collected from the fields. |
| Ethics oversight | the Institutional Animal Ethics Committee of HUST-Suzhou Institute for Brainsmatics |

Note that full information on the approval of the study protocol must also be provided in the manuscript.

## Plants

Seed stocks

Not applicable

Novel plant genotypes

Not applicable

Authentication

Not applicable

