## [Peer Review File · Nature]

A mouse brain stereotaxic topographic atlas with isotropic 1 μm resolution

Corresponding Author: Professor Qingming Luo

Version 0:

Reviewer comments:

Referee #1

(Remarks to the Author)

The study presents a high-resolution 3D reference atlas of the mouse brain, addressing the critical need for spatial localization at single-cell resolution in multi-omics research. The authors employed an innovative 3D Nissl-stained image dataset with isotropic 1 μm resolution, achieved through continuous micro-optical sectioning tomography, to construct a 3D mouse brain stereotaxic atlas. Additionally, they developed an informatics-based platform for visualizing and sharing atlas images, offering functionalities such as brain slice registration and neuronal circuit mapping. I agree with the authors that this atlas represents a powerful neuroinformatics tool for studying the brain at single-cell resolution. This work is a highly significant achievement, especially given the limitations of traditional rodent brain atlases, which rely on manually annotated Nissl-stained coronal sections spaced hundreds of micrometers apart. These traditional methods restrict the observation of continuous structural changes and accurate 3D reconstruction.

The article is well-written, with excellent and elegant illustrations. I have no specific concerns regarding the quality or rigor of the work. My only suggestion is for the authors to briefly discuss an additional potential application of their atlas: its use in studying brain alterations in mouse models of diseases, particularly Alzheimer's disease. The atlas could facilitate the registration and integration of single-cell-level brain mapping data with specific localizations of pathological features, such as amyloid plaques. Including a brief comment on this would further emphasize the broad utility of the atlas.

Referee #2

(Remarks to the Author)

This article presents the construction of an atlas of the entire mouse brain in Nissl at the single cell level, using micro-optical sectioning tomography. It also presents the development of the visualisation and sharing of atlas imaging. This is a major step in atlas making and it is a resource that is definitely required by scientists who increasingly need higher resolution and comprehensiveness. The work is highly original in scope and quality. Displaying the whole brain provides for everyone working on the organ. Obtaining the quality of Nissl staining (given it is impregnation of the whole brain) is unprecedented. The approach is valid because the authors did find the subdivisions of cortex, amygdala etc expected in the locations they display. That is, the work presented herein is amazing. However, there are issues to be addressed, including a major improvement to be obtained if the authors followed my advice.

A minor issue is the implementation of a number of corrections to expression I am sending you in tracked changes in the attachment. Another minor issue is their mistaken identification of the bregma landmark as indicated in their Fig 4D magnified images (skull diagram): they have misrepresented bregma by presenting the suture above the olfactory bulb, also inverting it. Likewise they inverted the magnified image of the lambda reference point.

There is an issue relating to allocating credit. The authors state in Line 906, they chose to base their delineations primarily on the Allen Reference Atlas (ARA) "with only occasional references" to the Paxinos and Franklin atlas, indeed, paradoxically, they refer to the 2001 edition of the latter atlas (an atlas which has had three subsequent editions, the current

published in 2019). If the reliance on the Paxinos and Franklin atlas was more than “occasional,” the authors would need to adjust the epithet used. The authors would have been aware that when comparing the above two atlases Chon et (2019; see quoted paragraph below) found the Franklin and Paxinos (FP)(Paxinos and Franklin) atlas to be more accurate and more cited than the Allen Reference Atlas.

Be that as it may, the major issue I see in this Ms is the choice of the Allen Reference Atlas for nomenclature and abbreviations, when the Paxinos and Franklin nomenclature and abbreviations are already used in atlases of rats, mice, marmosets, rhesus monkeys, bats, gerbils, birds, and humans as well as developmental atlases (for homologous structures). The authors' work would be better integrated with that on other species if the nomenclature and abbreviations of Paxinos and Franklin were also available, especially in view of the authors' stated goal to extend their work to non-human primate and humans. The work of the authors would be that much more integrated with that of other scientist working on the mouse and other species, including humans, if they also availed it in the nomenclature and abbreviations of Paxinos and Franklin. Please note the paragraph below where Chon ...& Kim (2019) compare the Allen Reference Atlas (HW Dong) to that of Franklin and Paxinos (FP). Chon et al & Kim also used the Paxinos and Franklin nomenclature in their work, something which the authors of the current manuscript could emulate. That is, the authors can provide their existing delineations with the Paxinos and Franklin nomenclature and abbreviations as Chon... & Kim have done, something that will benefit their atlas and the broader community which uses the list Paxinos and colleagues carefully developed over the last 43 years. BELOW IS THE LINK TO THE CHON...& KIM ARTICLE AND THE MOST RELEVANT PARAGRAPH.

<https://www.nature.com/articles/s41467-019-13057-w>

“Unfortunately, significant discrepancies exist between the anatomical labels on the ARA [Allen Reference Atlas] and the FP [Franklin & Paxinos] labels. For example, these two atlases often have discordant anatomical borders and 3D coordinates as well as different names for the same structures^{12,20}. To make it worse, the labels in the Allen CCF released in 2017 (CCFv3) also introduced significant changes from its original ARA labels that were based on 2D Nissl stained sections. This has created confusion and misinterpretation of experimental results²¹. These issues motivated us to create a unified and highly segmented anatomical labeling system in the adult mouse brain based on the Allen CCF. We decided to use the FP labels for our initial anatomical labeling because it represents one of the most popular adult mouse brain atlases with detailed segmentations, and because a huge body of prior research is based on the FP labels^{12,18}. Here, we adopt the FP labels into the Allen CCF by rigorous alignment using an MRI based atlas and cell type specific transgenic mice marking for distinct anatomical areas^{18,22}. We also further segment labels where cell types could be distinguished within single anatomically defined regions. The resulting labels create a unique opportunity for comprehensive comparisons between the two most frequently used anatomical labels in a common space. Furthermore, we use topographically distinct cortico-striatal projection patterns to add segmentations to the dorsal striatum, which is unsegmented in the existing atlases.”

Referee #3

(Remarks to the Author)

Summary:

The authors used block-face optical imaging to image an entire Nissl-stained mouse brain at a voxel size of 0.3 x 0.3 x 1 micrometer. They used this dataset as the core of a high-resolution brain atlas. They have made this atlas and a suite of related tools available online. While not all the claims made in the current manuscript are fully backed up by the reported data, the dataset and the related tools represent a significant amount of work and a significant contribution to the field. Given the value of the data and tools to the neuroscience community, I would like to see them published in a high-profile journal like Nature. Prior to publication, I would like to see the issues below dealt with. Almost all of these issues can be resolved by qualifying claims and clarifying limitations in the text.

Writing style:

I think the manuscript suffers in several places from making claims that are not directly supported by evidence. These claims are sometimes stylistic: “This authentic cytoarchitectural texture information aids in identifying genuine and accurate boundaries of brain structures.” All these adjectives constitute claims that could be explicitly quantified in the manuscript. There are two places where I believe these types of claims cross the line into being seriously misleading and I have listed them below. Outside of these two points, I think the text would be better received if the authors toned down some of this type of language.

Atlas resolution:

One of the problems with this manuscript is that it states that the atlas is micron-resolution. The original Nissl-stained brain was imaged with a 0.3 x 0.3 x 1 um voxel size. There is a 1 um isotropic version of this image volume that is reslicable. However, the regional topography atlas is at 10 x 10 x 10 um (?). STAM operates using every 20th section. More generally, the resolution of the regional delineations is not equal to the voxel size. Regional delineations are made based on manual evaluations of the relative distribution of fields of cell bodies in every 20th section of the Nissl dataset. The resolution of these decisions is complicated by the fact that adjacent sections were available and that other types of labeling from other datasets were leveraged to make decisions.

I think the authors have been transparent with these limitations when describing individual results, but there is no part of the text that synthesizes these different resolutions in a way that is easy for the reader to make sense of. These differences in voxel size, precision and, accuracy is an important point because different kinds of atlases have different levels of predictive accuracy for different kinds of questions and the small voxel size of the source data is the key feature of this project. The primary alternative approach to the proposed manuscript would be to combine many lower resolution datasets that would lose details that vary between individuals but that might have a higher predictive accuracy between individuals.

One solution would be to quantify the accuracy of different aspects of the atlas. For example, what was the section-to-section variation in boundary position prior to smoothing? In the absence of additional quantification of accuracy, I would also be satisfied if a paragraph or two bringing together all the issues of voxel size vs resolution vs segmentation and registration accuracy were included in the limitations section of the discussion.

Small regions:

In line 302, the authors claim that structures only a few tens of micrometers can be identified. That would seem to be an important claim if that resolution of object identification is a general feature of their atlas. But it is difficult to imagine how such regions could be reliably identified in a single piece of tissue where regions are defined by the relative distribution of 10 micrometer wide cell bodies. I think that claim needs to be backed up by side-by-side images of the raw data used to segment the islands and the segmentation. Preferably, the images would be in non-supplementary figures.

Nissl staining:

The manuscript makes several claims about the richness of information that is available in the Nissl volume that is not supported by the figures. They claim that size, shape, and diversity of cells can be derived from the Nissl staining in their volume. This claim is true for Nissl staining, in general, but is not clear how much of this information is available in their atlas. First, there should be images somewhere in the non-supplemental manuscript that make the quality of the raw data easy to judge. Second, there should be some qualification about what features of the Nissl staining are available in which version of the atlas. When I imagine Nissl staining for an entire brain, I imagine images like <https://brainmaps.org/ajax-viewer.php?datid=116&sname=4g2>. The images I found on STAM were of much lower quality. I was not able to zoom in enough to see much subcellular detail. The contrast was also too high. Non-cell body signal was almost completely saturated. I believe the usefulness of the atlas would be improved if additional (even empty) zoom was enabled in the display and if the contrast could be reduced to make background signal visible.

The high contrast of the available data also relates to a problem with the manuscripts claim that fiber tracts can be mapped in the Nissl dataset (line 123). This claim needs much more explanation. Brain atlases are often based on combinations of Nissl staining and Myelin staining because fiber tracts are difficult to see in Nissl staining. It is possible to infer large fiber tracts from the cell body staining. Smaller fiber tracts could be visible if the contrast were not set so high. As it stands, leaving readers with the impression that an atlas exists that reveals micron-resolution fiber tracts is misleading.

Website performance:

My experience with the performance of the website was not great. Data loading was slow, making it difficult to navigate through tissue. There are many wonderful tools in STAM. Isotropic reslice is great. The injection planner is great. Unfortunately, data loading was so slow that I would rather go to brainmaps.org to see annotated Nissl stains. Using Chrome, the website appears to be consuming large amounts of memory and CPU. Also, the website is also tagged by Chrome as "Not Secure". If there are settings that will improve performance, please make those clear.

Comprehensive connectivity map:

In line 255, the authors claim that they were able to make a comprehensive connectivity map. The authors mapped the projection of some number of neurons into their atlas. I could not find the number of neurons reported, but most of my connectivity queries turned up "no available neurons". The map is not comprehensive and it is, arguably, not connectivity. Please revise the language and be explicit about how many cells are currently in the system.

Accuracy of registration:

The authors state "The accuracy of all the registration calculations has been evaluated." Please provide the evaluations. Please include, at least, a summary statement of these evaluations in the non-supplemental text.

Minor issue:

For limitations: If you cite 10 years of annotation, it would be helpful to translate that into work hours/years. One annotator, a hundred annotators?

Version 1:

Reviewer comments:

Referee #1

(Remarks to the Author)

I suggest making slight modifications to the paragraph provided by the authors in response to my comments.

Original paragraph:

'Additionally, we intend to integrate more neurological data into the STAM. Besides the multi-omics data and more comprehensive neuronal circuit information, STAM also holds promise for integrating the pathological features, such as amyloid plaques associated with Alzheimer's disease. Given the small size and scattered distribution of amyloid plaques throughout the brain, the high-resolution spatial localization provided by STAM would enable the study of the spatial distribution of disease-related changes, thereby advancing our understanding of neurodegenerative disorders and brain alterations in mouse models'.

Revised paragraph:

'Beyond multi-omics data and more comprehensive neuronal circuit information, STAM also holds promise for integrating pathological features such as amyloid plaques associated with Alzheimer's disease. The high-resolution spatial localization provided by STAM would enable the study of the spatial distribution of plaques and their potential impact on neuronal circuits throughout the brain, thereby advancing our understanding of neurodegenerative disorders and brain alterations in mouse models'.

Furthermore, an important aspect of this work is that single-neuron morphology data can be visualized, including the locations of all branching points and terminals. However, upon re-reading the manuscript, I believe this aspect should be illustrated in greater detail, as the labeled neurons are shown at low magnification. Including a supplementary figure to provide a more detailed view of labeled neurons, particularly terminal arborizations, could enhance the manuscript.

Referee #2

(Remarks to the Author)

Line 1: Delete "The" from the title and insert "A" in its place. The "The" implies that there can be no other "mouse brain stereotaxic...". The suitable pronoun is "A".

Line 68. A cranial landmark (eg Bregma line) or a fiducial mark made by a needle as it passes through the brain is a reference point or line; calling it a "data mark" (rather than a reference point) is not correct.

Line 158. The correct reference to Paxinos and Franklin is: Paxinos G and Franklin KBJ, Paxinos and Franklin's the Mouse Brain in Stereotaxic Coordinates, Compact 5th Edition, Academic Press, San Diego, 2019.

Referee #3

(Remarks to the Author)

The authors have addressed the issues I raised in my initial review with additions to the texts, clarifications, and additions to STAM. My straining eyes would still benefit from allowing a couple more steps of magnification (until I can see pixels). Regardless, I am happy to recommend this manuscript for publication.

Referees' comments:

Referee #1 (Remarks to the Author):

The study presents a high-resolution 3D reference atlas of the mouse brain, addressing the critical need for spatial localization at single-cell resolution in multi-omics research. The authors employed an innovative 3D Nissl-stained image dataset with isotropic 1 μm resolution, achieved through continuous micro-optical sectioning tomography, to construct a 3D mouse brain stereotaxic atlas. Additionally, they developed an informatics-based platform for visualizing and sharing atlas images, offering functionalities such as brain slice registration and neuronal circuit mapping. I agree with the authors that this atlas represents a powerful neuroinformatics tool for studying the brain at single-cell resolution. This work is a highly significant achievement, especially given the limitations of traditional rodent brain atlases, which rely on manually annotated Nissl-stained coronal sections spaced hundreds of micrometers apart. These traditional methods restrict the observation of continuous structural changes and accurate 3D reconstruction.

The article is well-written, with excellent and elegant illustrations. I have no specific concerns regarding the quality or rigor of the work. My only suggestion is for the authors to briefly discuss an additional potential application of their atlas: its use in studying brain alterations in mouse models of diseases, particularly Alzheimer's disease. The atlas could facilitate the registration and integration of single-cell-level brain mapping data with specific localizations of pathological features, such as amyloid plaques. Including a brief comment on this would further emphasize the broad utility of the atlas.

Reply:

Thanks very much for your comments. To address the suggestion regarding the potential application of STAM in studying brain alterations in disease models, particularly for localizing amyloid plaques in Alzheimer's disease, we have added a brief discussion about the application prospects in brain diseases, which is highlighted in bold orange text in lines 458–465.

Referee #2 (Remarks to the Author):

This article presents the construction of an atlas of the entire mouse brain in Nissl at the single cell level, using micro-optical sectioning tomography. It also presents the development of the visualisation and sharing of atlas imaging. This is a major step in atlas making and it is a resource that is definitely required by scientists who increasingly need higher resolution and comprehensiveness. The work is highly original in scope and quality. Displaying the whole brain provides for everyone working on the organ. Obtaining the quality of Nissl staining (given it is impregnation of the whole brain) is unprecedented. The approach is valid because the authors did find the subdivisions of cortex, amygdala etc expected in the locations they display. That is, the work presented herein is amazing. However, there are issues to be addressed, including a major improvement to be obtained if the authors followed my advice.

A minor issue is the implementation of a number of corrections to expression I am sending you in tracked changes in the attachment. Another minor issue is their mistaken identification of the bregma landmark as indicated in their Fig 4D magnified images (skull diagram): they have misrepresented bregma by presenting the suture above the olfactory bulb, also inverting it. Likewise they inverted

the magnified image of the lambda reference point.

Reply:

We appreciate your acknowledgment of our work, and thanks for providing the tracked changes in the attachment. Due to file format conversion, it was somewhat challenging to track the changes; therefore, we carefully reviewed the corrections and revised the manuscript accordingly. The corrected expressions are now marked in bold orange text in lines 55–56, 66-68, 82-83, 100, 107, 110, 117, 128-130, 132-133,140, 167-168, 170, 184, 196-197, 203, 255, 308, 321-322, 324, 327, 330, 351, 362, 384-385, 454, 500, and 521.

And thanks for noticing the mistake in Figure. 4d. We have corrected the magnified images of Bregma and Lambda which were inverted.

Regarding the location of Bregma, we adopted the approach to determine the Bregma from a recent review ^[1]. This review examined various studies and noted inconsistencies in how Bregma is determined. For example, in the fifth edition of Franklin & Paxinos' mouse brain atlas, it is written that '*Bregma and lambda are the intersections of the midline suture with the line of best fit along the bregmoid and lambdoid suture, respectively.*' ^[2] However, this statement is not consistent with the accompanying figure from the same book, as shown below:

[FIGURE REDACTED]

[TEXT REDACTED]

As summarized in [1], '*Both atlases used the same skull figure showing the Bregma at the crossing point between the coronal and the sagittal sutures which is not compatible with the description mentioned before*'. To resolve this inconsistency, [1] recommended the method by Blasiak et al., which employed a parabola to best-fit the coronal suture, as shown in the figure below ^[3]. The **red point** indicates the '**old Bregma**' (i.e. the crossing point of sutures), while the **blue point** represents the '**new Bregma**' using a parabola drawn as a dashed line. [3] demonstrated that the '**new Bregma**' would lead to '*significantly smaller errors in the anteroposterior and mediolateral axes*'.

[FIGURE REDACTED]

[TEXT REDACTED]

We adopted this recommended approach for two reasons.

First, as we zoomed into the micron-resolution level, we found that there doesn't exist a 'intersection of the coronal and sagittal sutures', because the left and right 'limbs' of the coronal suture didn't converge at a single point on the sagittal suture. This can be seen from the reconstructed skull shown below. As seen in sub-figure B, the dash-lined circles indicate different converging points of the left and right limbs of the coronal suture.

[FIGURE REDACTED]

Secondly, after adopting the 'new Bregma' proposed by [1], we observed that the coronal atlas level immediately beneath the 'new Bregma', which came from the co-registered MOST-Nissl dataset, corresponded well with the typical 'Bregma 0 mm' atlas level as seen in traditional brain stereotactic atlases. This atlas level is shown in the sub-figure A below. For comparison, we also selected the atlas level beneath the "old Bregma" (see sub-figure B below), and found that only the new Bregma position produced the expected consistence.

[FIGURE REDACTED]

Finally, in the Figure 4, we noticed that the original dashed guidelines in sub-figure d, which followed the natural path of the coronal sutures, led to ambiguity. Therefore, we replaced them with a "best-fit" curve following the approach by Blasiak et al. [3] in the updated version of Figure. 4, and modified the legend text in lines 610-611 accordingly.

Reference

[1] Cecyn, M. N. & Abrahao, K. P. Where do you measure the Bregma for rodent stereotaxic surgery? *IBRO Neuroscience Reports* **15**, 143–148 (2023).

[2] Franklin, K. & Paxinos, G. *The Mouse Brain in Stereotaxic Coordinates, Compact*. (Academic Press, San Diego, CA, 2019).

[3] Blasiak, T., Czubak, W., Ignaciak, A. & Lewandowski, M. H. A new approach to detection of the bregma point on the rat skull. *J Neurosci Methods* **185**, 199–203 (2010).

There is an issue relating to allocating credit. The authors state in Line 906, they chose to base their delineations primarily on the Allen Reference Atlas (ARA) “with only occasional references” to the Paxinos and Franklin atlas, indeed, paradoxically, they refer to the 2001 edition of the latter atlas (an atlas which has had three subsequent editions, the current published in 2019). If the reliance on the Paxinos and Franklin atlas was more than “occasional,” the authors would need to adjust the epithet used. The authors would have been aware that when comparing the above two atlases Chon et al (2019; see quoted paragraph below) found the Franklin and Paxinos (FP)(Paxinos and Franklin) atlas to be more accurate and more cited than the Allen Reference Atlas.

Be that as it may, the major issue I see in this Ms is the choice of the Allen Reference Atlas for nomenclature and abbreviations, when the Paxinos and Franklin nomenclature and abbreviations are already used in atlases of rats, mice, marmosets, rhesus monkeys, bats, gerbils, birds, and humans as well as developmental atlases (for homologous structures). The authors’ work would be better integrated with that on other species if the nomenclature and abbreviations of Paxinos and Franklin were also available, especially in view of the authors’ stated goal to extend their work to non-human primate and humans. The work of the authors would be that much more integrated with that of other scientist working on the mouse and other species, including humans, if they also availed it in the nomenclature and abbreviations of Paxinos and Franklin. Please note the paragraph below where Chon ...& Kim (2019) compare the Allen Reference Atlas (HW Dong) to that of Franklin and Paxinos (FP). Chon et al & Kim also used the Paxinos and Franklin nomenclature in their work, something which the authors of the current manuscript could emulate. That is, the authors can provide their existing delineations with the Paxinos and Franklin nomenclature and abbreviations as Chon... & Kim have done, something that will benefit their atlas and the broader community which uses the list Paxinos and colleagues carefully developed over the last 43 years. BELOW IS THE LINK TO THE CHON...& KIM ARTICLE AND THE MOST RELEVANT PARAGRAPH.

<https://www.nature.com/articles/s41467-019-13057-w>

“Unfortunately, significant discrepancies exist between the anatomical labels on the ARA [Allen Reference Atlas] and the FP [Franklin & Paxinos] labels. For example, these two atlases often have discordant anatomical borders and 3D coordinates as well as different names for the same structures^{12,20}. To make it worse, the labels in the Allen CCF released in 2017 (CCFv3) also introduced significant changes from its original ARA labels that were based on 2D Nissl stained sections. This has created confusion and misinterpretation of experimental results²¹. These issues motivated us to create a unified and highly segmented anatomical labeling system in the adult mouse brain based on the Allen CCF. We decided to use the FP labels for our initial anatomical labeling because it represents one of the most popular adult mouse brain atlases with detailed segmentations, and because a huge body of prior research is based on the FP labels^{12,18}. Here, we adopt the FP labels into the Allen CCF by rigorous alignment using an MRI based atlas and cell type specific

transgenic mice marking for distinct anatomical areas^{18,22}. We also further segment labels where cell types could be distinguished within single anatomically defined regions. The resulting labels create a unique opportunity for comprehensive comparisons between the two most frequently used anatomical labels in a common space. Furthermore, we use topographically distinct cortico-striatal projection patterns to add segmentations to the dorsal striatum, which is unsegmented in the existing atlases.”

Reply:

Thank you for your suggestion. We have incorporated the nomenclature and abbreviations from the Franklin and Paxinos (F&P) atlas into STAM, based on the work by [1]. Initially, our delineations were based primarily on the Allen Reference Atlas (ARA); however, recognizing the need for a more balanced approach, we have now integrated the F&P delineations into STAM’s 2D viewer (see sub-figure B below), alongside the ARA-based ones (see sub-figure A below). This will benefit the users of F&P’s atlas by providing an online version of this atlas, and will also enrich the delineation of brain structures, such as the subdivisions of caudal putamen (also see the black texts in sub-figure B). **We cited [1] and briefly introduced this integration of F&P in the Results and Methods sections**, in lines 156–160 and 938-947 respectively with bold orange text. We also changed the expression about the nomenclature used in line 959.

In addition, based on the work of linking the F&P’s and ARA’s nomenclature systems by [1], we have implemented navigation buttons to allow users to switch between corresponding brain structures (see sub-figure C). This integration bridges the gap between different ontology systems and facilitates the broader integration of neuroscientific knowledge across species.

[FIGURE REDACTED]

The detail of integrating two nomenclatures is described as followed: we first downloaded the supplementary data from Chon et al. [1], which carefully connects F&P's nomenclature with that of the ARA. We then used this downloaded data as the basis for constructing a hierarchically organized F&P's nomenclature and visualized it in our web service. Specifically, we:

1. Extracted the anatomical label slices of F&P from the supplementary data provided by [1].
2. Utilized the corresponding CCF background coronal images to determine the spatial relationships between these F&P label slices and the CCF.
3. Considering that STAM is registered to the CCF, we accurately locate the FP label slices in our STAM's coordinate system.
4. Converted these label slices into vectorized borders and overlaid them onto the original images in STAM's 2D viewer.

Since the F&P anatomical label slices are provided at 100 μm intervals, their direct 3D reconstruction become challenging. Therefore, we currently display them in 2D space, even though some slices do not perfectly alignment with the coronal plane of STAM. Also due to this 100- μm interval, only 1 in every 5 atlas levels in STAM has a corresponding F&P atlas level. These levels are marked with "F&P" labels on their preview thumbnails, as shown in sub-figure D above.

Reference

[1] Chon, U., Vanselow, D. J., Cheng, K. & Kim, Y. Enhanced and unified anatomical labeling for a common mouse brain atlas. *Nat Commun* **10**, 5067 (2019).

Referee #3 (Remarks to the Author):

Summary:

The authors used block-face optical imaging to image an entire Nissl-stained mouse brain at a voxel size of 0.3 x 0.3 x 1 micrometer. They used this dataset as the core of a high-resolution brain atlas. They have made this atlas and a suite of related tools available online. While not all the claims made in the current manuscript are fully backed up by the reported data, the dataset and the related tools represent a significant amount of work and a significant contribution to the field.

Given the value of the data and tools to the neuroscience community, I would like to see them published in a high-profile journal like Nature. Prior to publication, I would like to see the issues below dealt with. Almost all of these issues can be resolved by qualifying claims and clarifying limitations in the text.

Writing style:

I think the manuscript suffers in several places from making claims that are not directly supported by evidence. These claims are sometimes stylistic: "This authentic cytoarchitectural texture information aids in identifying genuine and accurate boundaries of brain structures." All these adjectives constitute claims that could be explicitly quantified in the manuscript. There are two places where I believe these types of claims cross the line into being seriously misleading and I have

listed them below. Outside of these two points, I think the text would be better received if the authors toned down some of this type of language.

Reply:

Thanks for your comments. We have investigated the whole manuscript, and removed or revised expressions that may be too subjective, including not only certain adjectives but also some verbs. All changes are highlighted in orange, with deleted text shown as strikethrough and revised text in bold orange. These changes can be seen in lines 34-35, 66, 73, 102-103, 108, 114, 124, 168, 176, 217, 238, 257, 289, 298, 330, 376, 390-391, 396, and 403.

Atlas resolution:

One of the problems with this manuscript is that it states that the atlas is micron-resolution. The original Nissl-stained brain was imaged with a $0.3 \times 0.3 \times 1 \mu\text{m}$ voxel size. There is a $1 \mu\text{m}$ isotropic version of this image volume that is reslicable. However, the regional topography atlas is at $10 \times 10 \times 10 \mu\text{m}$ (?). STAM operates using every 20th section. More generally, the resolution of the regional delineations is not equal to the voxel size. Regional delineations are made based on manual evaluations of the relative distribution of fields of cell bodies in every 20th section of the Nissl dataset. The resolution of these decisions is complicated by the fact that adjacent sections were available and that other types of labeling from other datasets were leveraged to make decisions.

I think the authors have been transparent with these limitations when describing individual results, but there is no part of the text that synthesizes these different resolutions in a way that is easy for the reader to make sense of. These differences in voxel size, precision and, accuracy is an important point because different kinds of atlases have different levels of predictive accuracy for different kinds of questions and the small voxel size of the source data is the key feature of this project. The primary alternative approach to the proposed manuscript would be to combine many lower resolution datasets that would lose details that vary between individuals but that might have a higher predictive accuracy between individuals.

One solution would be to quantify the accuracy of different aspects of the atlas. For example, what was the section-to-section variation in boundary position prior to smoothing? In the absence of additional quantification of accuracy, I would also be satisfied if a paragraph or two bringing together all the issues of voxel size vs resolution vs segmentation and registration accuracy were included in the limitations section of the discussion.

Reply:

Thanks for your comments and suggestions. The original image resolution is $0.35 \times 0.35 \times 1 \mu\text{m}$ per voxel, which is further resampled to an isotropic $1 \mu\text{m}$ resolution image volume to enable arbitrary-angle reslicing; the delineation of brain region boundaries is basically performed on every 20th coronal planes; and the isotropic $10 \mu\text{m}$ resolution regional topography atlas is derived from the stacked coronal atlas levels, which is referred to as the '10 μm resolution annotation image' in the 'Reconstruction and visualization' part of the Methods section. We processed and analyzed images of varying resolutions at different stages of the workflow.

To avoid the possible ambiguity, we arranged a new part named ‘Precision of atlas illustrating’ in the Discussion section. In this part, we first introduced the above-mentioned strategies of atlas illustrating, then discussed reasons that we adopted the 20 μm interval. The modified or newly added sentences in lines 422-444 are marked as bold text with orange color.

Additionally, some points still require clarification:

1. The coronal planes used to create the atlas are not just taken from every 20th coronal plane, but from a projected image created by combining each plane with its 19 neighboring planes. This approach is necessary because 20- μm thickness projected images provide clearer structural borders than 1 μm thickness images, where many borders are not distinct.

2. With the isotropic 1 μm resolution, we can generate projections at any desired location to observe continuous changes along specific axes, rather than using fixed intervals. This allows us to track the appearance, disappearance, or drastic morphological changes of brain structures for delineation. For example, Video. S1 shows the appearance and disappearance of the triangular nucleus of the septum (TRS) along the coronal direction. The slices shown in the movie change continuously at 1 μm intervals, with each slice representing a projected image with 20- μm thickness.

Specially, we focused on the appearance of TRS using the figure below. The sub-figure A shows part of the 5171st coronal section from MOST-Nissl dataset of STAM, with a small region marked as B1 where the left part of the TRS will appear. The sub-figure B displays the continuous change of this small region marked as B1 in A on 20 consecutive coronal sections from 5171st to 5190th section of the MOST-Nissl dataset. In B1 and B2, the 'seed' regions, marked with dashed lines, will gradually grow into the TRS, which is also marked with dashed line in B20.

Small regions:

In line 302, the authors claim that structures only a few tens of micrometers can be identified. That would seem to be an important claim if that resolution of object identification is a general feature of their atlas. But it is difficult to imagine how such regions could be reliably identified in a single piece of tissue where regions are defined by the relative distribution of 10 micrometer wide cell bodies. I think that claim needs to be backed up by side-by-side images of the raw data used to

segment the islands and the segmentation. Preferably, the images would be in non-supplementary figures.

Reply:

Thanks for your comment. The identification of small regions is indeed a feature of our atlas due to the isotropic 1 μm resolution of the MOST-Nissl dataset. To ensure accurate delineation of small structures, such as islands of Calleja (OTisl) in the olfactory tubercle, we first delineated them on coronal sections with 1 μm horizontal resolution, which provided us the information of distinct somas for structure recognition. The delineated boundaries were then stacked and processed in 3D with isotropic 10 μm resolution. While single neurons were no longer resolvable at this stage, the pre-delineated boundaries provided a reliable reference for further refinement. We have included magnified images in Figure 3e to further highlight these details, with its figure legend text modified and highlighted with bold orange in lines 573-578.

(A snapshot from the new version of Figure. 3)

Nissl staining:

The manuscript makes several claims about the richness of information that is available in the Nissl volume that is not supported by the figures. They claim that size, shape, and diversity of cells can be derived from the Nissl staining in their volume. This claim is true for Nissl staining, in general, but is not clear how much of this information is available in their atlas. First, there should be images somewhere in the non-supplemental manuscript that make the quality of the raw data easy to judge. Second, there should be some qualification about what features of the Nissl staining are available in which version of the atlas. When I imagine Nissl staining for an entire brain, I imagine images like <http://brainmaps.org/ajax-viewer.php?datid=116&sname=4g2>. The images I found on STAM were of much lower quality. I was not able to zoom in enough to see much subcellular detail. The contrast was also too high. Non-cell body signal was almost completely saturated. I believe the

usefulness of the atlas would be improved if additional (even empty) zoom was enabled in the display and if the contrast could be reduced to make background signal visible.

The high contrast of the available data also relates to a problem with the manuscripts claim that fiber tracts can be mapped in the Nissl dataset (line 123). This claim needs much more explanation. Brain atlases are often based on combinations of Nissl staining and Myelin staining because fiber tracts are difficult to see in Nissl staining. It is possible to infer large fiber tracts from the cell body staining. Smaller fiber tracts could be visible if the contrast were not set so high. As it stands, leaving readers with the impression that an atlas exists that reveals micron-resolution fiber tracts is misleading.

Reply:

Thanks for your suggestion. There are several aspects to address regarding this concern.

First, regarding the quality of our Nissl-stained image data, single cells can indeed be observed in our Nissl-stained image volume, as demonstrated in Figure. 2a. Additionally, Figure 2c and 2e illustrate variations in cell shape, size, and arrangement. Specifically, Figure 2c highlights the rapid transitions in neuronal stratification patterns across neighboring cortical areas, while Figure 2e shows differences in cell density, size, and shape, which were key features used for brain structure delineation. To further support this, additional examples can be found in Figure S1 and S2. For instance, Figure S1e provides an example where differences in soma size, shape, and arrangement were used to determine brain structure boundaries, as shown below:

And thank you for bringing up brainmaps.org, developed by Edward G. Jones and Shawn Mikula et al. This platform has been maintained for a long time and provides high-quality histological slice images. Compared to brainmaps.org, our Nissl-staining method and imaging approach were

specifically designed to acquire a 3D Nissl-stained image volume of the entire brain at micron resolution. The primary objective was to capture continuous cytoarchitectural changes in any orientation rather than to maximize subcellular detail in individual 2D sections.

We appreciate the contributions of brainmaps.org to neuroanatomy and explored the possibility of integrating their resources with STAM. To this end, **we selected three datasets from brainmaps.org—one for each of the coronal, sagittal, and horizontal plane, and compared them with arbitrary-angle slices generated from our 3D Nissl-stained image volume.** The results are compiled into a reference table, which is available at <http://atlas.brainsmatics.org/STAM/reference/corresponding.html> . **We cited the work of brainmaps.org [1], and supplemented a brief introduction for the connection between STAM and it in the ‘Innovation and contribution’ part of the Discussion section, in lines 412-415.** We hope this resource will serve as a useful supplement, providing additional cytoarchitectural details alongside our Nissl-stained images.

The second issue is about the image contrast. The original Nissl-stained images were acquired in grayscale and later converted into the pseudocolor images currently displayed on our website. To address this concern, we have enhanced the 2D viewer on the STAM website by incorporating the original grayscale coronal Nissl-stained images alongside the pseudocolor versions.

With this update, users can now switch between pseudocolor and grayscale modes for better visualization. To access the grayscale images, users can navigate to the "Settings" button at the bottom of the page and uncheck the "Pseudocolor" option. The grayscale images may reveal additional details that are less discernible in the pseudocolor representation.

[FIGURE REDACTED]

And thanks for your comment regarding fiber tracts. We acknowledge that the Nissl-staining method does not directly stain fiber tracts, nor does it allow for the visualization of continuous single-neuron fibers at micron resolution. Our statement in the manuscript was intended to convey that cells located along fiber pathways can be observed in the Nissl-stained images, which enables us to infer the orientation and location of major fiber structures.

To avoid ambiguity, we have revised our terminology and now use "fiber bundles" instead of "fiber tracts," specifically referring to large fiber structures such as the anterior commissure and corpus callosum (cc), in line 121 with bold orange text. To illustrate this, below we provide a randomly selected snapshot from our website, where black arrows indicate the fiber bundle cc. Noting that in Video. S1 we can also observe the fiber bundle cc and its continuous change from the 11st second.

[FIGURE REDACTED]

(Link: **[TEXT REDACTED]**)

Reference

[1] Mikula, S., Trotts, I., Stone, J. M. & Jones, E. G. Internet-enabled high-resolution brain mapping and virtual microscopy. *NeuroImage* 35, 9 – 15 (2007).

Website performance:

My experience with the performance of the website was not great. Data loading was slow, making it difficult to navigate through tissue. There are many wonderful tools in STAM. Isotropic reslice is great. The injection planner is great. Unfortunately, data loading was so slow that I would rather go to brainmaps.org to see annotated Nissl stains. Using Chrome, the website appears to be consuming large amounts of memory and CPU. Also, the website is also tagged by Chrome as “Not Secure”. If there are settings that will improve performance, please make those clear.

Reply:

Thank you for bringing up the performance issues with the website, especially as we anticipate a significant increase in global access. To achieve a better performance, we have deployed a secondary instance of STAM services on an Amazon Web Services (AWS) node in the United States while retaining the original server in China. Users from Europe and North America are now automatically directed to the AWS node to improve access speed. Our tests show that while the initial connection may experience a few seconds of latency, performance significantly improves once the connection is established. Regarding the "Not Secure" warning, this issue is related to the SSL certificate settings on our site. We have applied for the certificate, and it is currently in the process, which may

take some time to take effect. In the near future, we will deploy a new instance of STAM based on a Europe-located server to further improve the performance.

We have optimized CPU usage across various web services on the STAM platform, reducing it by 30~40% to enhance performance. However, for the arbitrary-angle reslicing service, each new slice displayed on the web page is generated in real time from a 3D image volume containing over **1436 billion** voxels. This process requires approximately **300 million** Flops for image interpolation and incurs significant time for data transferring. To inform users about this, we have added a prompt to explain this on the page, along with a progress bar to display the data loading status and help manage user expectations.

Comprehensive connectivity map:

In line 255, the authors claim that they were able to make a comprehensive connectivity map. The authors mapped the projection of some number of neurons into their atlas. I could not find the number of neurons reported, but most of my connectivity queries turned up “no available neurons”. The map is not comprehensive and it is, arguably, not connectivity. Please revise the language and be explicit about how many cells are currently in the system.

Reply:

Thanks for your comment. We have initially mapped morphological data from 1,644 neurons onto STAM, and now made all of these neurons fully accessible. The somas of these neurons are mainly located in motor cortex (MO), hippocampus (HIP) and cerebral nuclei (CNU). The purpose of constructing the connectivity map is to demonstrate the STAM can serve as a publicly accessible resource facilitating various neuroscience researches. Determining soma locations and projection pathways at the single-neuron level is one of its potential future applications.

Users can explore the "connectivity diagram" derived from these neurons by opening the ‘neuronal circuits’ panel and clicking the ‘whole-brain connectivity diagram’ button on the right side of the page. By right-clicking the connecting line between any two structures of interest, users can view the specific neurons contributing to that connection. Further instructions can be found on the help page (http://atlas.brainsmatics.org/STAM/help.html?node=Neuronal_Circuits&lang=en_us). Additionally, we have revised the wording in line 258 to explicitly include the neuron count.

While the current dataset includes 1,644 neurons, we are actively working to integrate additional neuronal morphological data from our group’s published studies, as well as data from other public researches. Given the extensive workload required for large-scale image registration, this remains an ongoing effort. We believe these continuous expansions will contribute to the development of a more comprehensive connectivity map.

Accuracy of registration:

139 The authors state “The accuracy of all the registration calculations has been evaluated.” Please provide the evaluations. Please include, at least, a summary statement of these evaluations in the non-supplemental text.

Reply:

As the auxiliary images come from various sources, we evaluated their registration separately, and provided the evaluation results in Tab. S8, Tab. S9 and Tab. S10 respectively. We added the summary statement in lines 138-139, with orange bold text.

Minor issue:

For limitations: If you cite 10 years of annotation, it would be helpful to translate that into work hours/years. One annotator, a hundred annotators?

Reply:

The estimated 10-year timespan encompasses not only the annotation process but also the development of the whole-brain sample preparation method, imaging techniques, and the image processing pipeline, and more. Therefore, this is a long-term research project involving substantial work and contributions from many individuals. We noted this in lines 447–449 of the ‘Limitations of our pipeline’ part in Discussion section, and the participants of this research are listed in the Acknowledgement.

Referees' comments:**Referee #1 (Remarks to the Author):**

I suggest making slight modifications to the paragraph provided by the authors in response to my comments.

Original paragraph:

“Additionally, we intend to integrate more neurological data into the STAM. Besides the multi-omics data and more comprehensive neuronal circuit information, STAM also holds promise for integrating the pathological features, such as amyloid plaques associated with Alzheimer’s disease. Given the small size and scattered distribution of amyloid plaques throughout the brain, the high-resolution spatial localization provided by STAM would enable the study of the spatial distribution of disease-related changes, thereby advancing our understanding of neurodegenerative disorders and brain alterations in mouse models”.

Revised paragraph:

“Beyond multi-omics data and more comprehensive neuronal circuit information, STAM also holds promise for integrating pathological features such as amyloid plaques associated with Alzheimer’s disease. The high-resolution spatial localization provided by STAM would enable the study of the spatial distribution of plaques and their potential impact on neuronal circuits throughout the brain, thereby advancing our understanding of neurodegenerative disorders and brain alterations in mouse models”.

Reply:

Thank you very much for your suggestion. We agree that the revised version provides a clearer and more concise description of the potential application of STAM in localizing pathological features. Accordingly, we have replaced the original paragraph with the revised version in lines 417 – 423 of the manuscript.

Furthermore, an important aspect of this work is that single-neuron morphology data can be visualized, including the locations of all branching points and terminals. However, upon re-reading the manuscript, I believe this aspect should be illustrated in greater detail, as the labeled neurons are shown at low magnification. Including a supplementary figure to provide a more detailed view of labeled neurons, particularly terminal arborizations, could enhance the manuscript.

Reply:

Thank you for your valuable comments. We agree that the ability to spatially localize and visualize single-neuron morphology, including branching points and terminal arborizations, is a key feature of STAM. To address this, we already included Extended Data Fig. 8, which provides a magnified view of representative labeled neurons. In this figure, branching points are marked with highlight dots, and the terminal arborizations are clearly illustrated to better demonstrate the level of morphological detail supported by our platform.

Referee #2 (Remarks to the Author):

Line 1: Delete "The" from the title and insert "A" in its place. The "The" implies that there can be no other "mouse brain stereotaxic...". The suitable pronoun is "A".

Line 68. A cranial landmark (eg Bregma line) or a fiducial mark made by a needle as it passes through the brain is a reference point or line; calling it a "data mark" (rather than a reference point) is not correct.

Line 158. The correct reference to Paxinos and Franklin is: Paxinos G and Franklin KBJ, Paxinos and Franklin's *The Mouse Brain in Stereotaxic Coordinates*, Compact 5th Edition, Academic Press, San Diego, 2019.

Reply:

Thank you very much for your comments.

We have revised the title of the manuscript by replacing 'The' with 'A' to avoid the implication of exclusivity.

The reference to Paxinos and Franklin's *The Mouse Brain in Stereotaxic Coordinates* has been corrected to the full citation of the 5th edition (2019), as suggested.

Regarding the term 'datum mark', our intention was to highlight that these points serve as fixed standards for measuring positions and distances in 3D brain space, analogous to the concept of datum marks in geodesy. While 'reference point' is more widely used, especially in registration tasks, we chose 'datum mark' to emphasize this spatial measurement perspective. Nevertheless, to avoid potential confusion, we have kept the clarification 'reference points, which we called datum marks' in line 74.

Referee #3 (Remarks to the Author):

The authors have addressed the issues I raised in my initial review with additions to the texts, clarifications, and additions to STAM. My straining eyes would still benefit from allowing a couple more steps of magnification (until I can see pixels). Regardless, I am happy to recommend this manuscript for publication.

Reply:

Thank you for your feedback and recommendation. We will consider adding additional magnification levels in future updates, thus further improving the viewing experience in the 2D viewer of 'Canonical plane visualization service' of STAM. In the meantime, the 'Arbitrary plane service' already supports pixel-level magnification, which can be accessed by holding the Ctrl key and scrolling with the mouse wheel during navigation.

1. HERE ARE SUGGESTED IMPROVEMENTS TO EXPRESSION AND CORRECTIONS:

There is need to address these limitations of the traditional 2D reference atlases and facilitate 3D brain mapping
projects. Further, what needs to be met is spatial localization needs arising from extensive 3D image datasets generated in neuronal
circuits and multi-omics studies. Q. Wang et al. constructed a common coordinate
framework (CCF) based on the autofluorescence of the mouse brain tissue ¹¹. Anatomical
delineations of the brain structures in the CCF were based on a computationally derived average
template at relatively low resolution, rather than actual cytoarchitecture. Furthermore, the axial
resolution of the datasets used to construct this template is only 100 μm , which is insufficient for
recognizing cellular-level details. Consequently, delineation of many brain structures became
controversial ¹¹. These limitations are not well-suited for mapping single neuron
resolution morphology and spatial transcriptome data.
To address these challenges, we leveraged an unprecedented 3D Nissl-stained image dataset with
isotropic 1 μm resolution to construct a 3D mouse brain stereotaxic atlas, representing the
topography of all structures while achieving precise single-cell resolution.
We defined a spatial coordinate system for the atlas based on both cranial and intracranial reference
marks. Additionally, we have developed a series of visualization and application services for the
atlas to meet the diverse needs of the scientific community in atlas visualization, anatomical
information retrieval, intelligent stereotaxic surgery planning, and more. Brain atlases have
continuously evolved over the last 100 years. We believe that this newly reconstructed mouse brain

atlas, featuring an unprecedented 1 μm isotropic resolution, will mark another milestone. It

provides a powerful informatics tool for large-scale brain mapping projects and serves as a valuable
"traditional reference atlas" for numerous individual scientists.

3

RESULTS
*The mouse brain stereotaxic topographic atlas*
*We have constructed the stereotaxic topographic atlas of the mouse brain (STAM) with isotropic 1*
*μm resolution based on various types of datasets, including cytoarchitecture,*
*immunohistochemistry and distribution of specific gene-type neurons. The atlas comprises*
*14,000 coronal slices, 11,400 sagittal slices, and 9,000 horizontal slices (Fig. 1a). Following the*
*nomenclatures defined in the original Allen Reference Atlas and Swanson's Brain maps 4.0 ^{7,8}, a*
*total of 916 hierarchically organized brain structures are delineated and reconstructed in 3D,*
*including 185 fine cortical areas, and 445 finest subcortical regions (Fig. 1b, c). We provide a*
*detailed list of discriminative criteria for each brain region in STAM, with most regions relying on*
*two or more types of supporting evidence, including Nissl staining images (Tab. S1).*
*Since the STAM is primarily based on isotropic 1 μm resolution image datasets of the whole brain,*
*we also offer atlas levels generated at arbitrary angles, along with open access to this high-resolution*
*dataset (Fig. 1d, e). The 3D STAM facilitates the localization of various types of neuroinformation,*
*based on which we developed a variety of web services to support neuroscience research (Fig.*
*1f). We also provide tools for conventional needs, such as brain slice registration, multi-⁹²*

modal image fusion, and use of skull-based stereotaxic coordinate system (Fig. 1g, h).

*Open-accessible 3D cytoarchitectural brain image with one-micron resolution*
*Using an improved Nissl staining method and the micro-optical sectioning tomography (MOST)*
*bright-field imaging technique, we obtained one 3D cytoarchitecture image dataset with a resolution*
*of $0.35 \times 0.35 \times 1 \mu\text{m}^3$, achieving micron-level resolution in both horizontal and axial directions ¹².*
*The original data were processed into an isotropic 1 μm resolution in three sectional directions, and*
*then non-linearly mapped to the CCF (Version 3, the same after) to achieve global morphological*
*correction, using the BrainsMapi as the registration tool, and only the outline of the whole brain as*
*the anatomical landmark for registration ¹³. The globally corrected dataset, referred to as the*

MOST101

Nissl dataset, has dimensions of 11400 \AA ~ 9000 \AA ~ 14000 pixels, and it is further used for atlas

construction (Fig. 1a).
The high-quality Nissl staining images obtained encompass neurons and glial cells throughout the
entire brain, providing a clear representation of the shape and size of individual cells of the brain,
which could not be observed on an averaged template created from individual specimens (Fig. 2a).
The obtained high-resolution MOST-Nissl dataset provides rich cytoarchitectonic information,
including cell diversity and distribution patterns in different brain structures, revealing their
boundaries (Fig. S1). For example, by examining the changes in lamination patterns of
cells in the images, we can determine precise boundaries between different cortical areas (Fig. 2b,
c). Also, by observing the discrepancies in density, size, and morphology of neuronal somas, we can
identify distinct subcortical regions (Fig. 2d, e).
Moreover, the isotropic 1 μm resolution of the MOST-Nissl dataset offers an advantage in observing
the continuous changes of any specific anatomical structure on the two-dimensional (2D) planes.
This capability benefits the accurate determination of anatomical locations where the key
features of certain brain structures "appear" or "disappear," thereby obtaining their precise 3D
topography. Using the small triangular nucleus of the septum (TRS) in the cerebral nuclei (CNU)
as an example, we observed its appearance along the Anterior-Posterior (A-P) axis with a 1 μm axial
steps in the MOST-Nissl dataset, starting from the interior side of the fornix and disappearing at the
ventral side of the dorsal fornix (Mov. S1). In contrast, traditional stereotaxic brain atlases, with
larger intervals between coronal sections, cannot precisely reveal the entire TRS on the coronal
plane and risk misinterpreting its remnants as the septofimbrial nucleus (SF) 6,7.
In addition to small nuclei, fiber tracts represent another category of morphologically complex brain
structures. Taking the olfactory limb of the anterior commissure (ac) as an example, the MOST
Nissl images reveal this structure located in the anterior part of the mouse brain with symmetric
branches on both hemispheres. Along the A-P direction, we can observe the precise location of the
intersection point of these branches (Fig. S2), indicating that the spatial reference for brain
positioning is at the single-cell resolution.
Atlas levels on canonical planes
Taking cytoarchitectonic information as the primary foundation and referring to mouse
brain atlases and various other reference data sets 8,14 (including the distribution of neurons with
specific gene types), we initially focused on coronal section images to delineate the boundaries of
different brain structures. Briefly, the 20 μm thickness projected Nissl-stained coronal sections
provided the main templates to identify anatomical structures, with non-linearly aligned auxiliary
coronal images from various other datasets providing additional information (Fig. 2f, Fig. S3-
S6, Tab. S1). The list of used image datasets from our lab is provided in Tab.S2. We also calculated
the cytoarchitectonic profiles along the depth of the cortex and mapped its distribution to uncover
additional features for delineating boundaries of different cortical areas (Fig. 2g) 15. The delineation
of HPF from previously published literature was also introduced through non-linear image
registration (Fig. 2h) 16. The accuracy of all the registration calculations has been evaluated. The
details of registration methods can be found to in the 'Multiple-source auxiliary image
registration' part of the Methods section. By incorporating the information from multiple sources,
we created a comprehensive set of coronal atlas levels with abundant labels for brain structures (Fig.
2i). We ensured seamless adjacency of these labels to eliminate any "terra nullius" between
neighboring structures.
Subsequently, we computed the obtained coronal atlas levels into sagittal and horizontal planes. By
referencing the continuous cytoarchitectural features of the MOST-Nissl dataset on these two
5

planes, we applied smoothing and optimization 147 to the drawn boundaries to address the common
"jigsaw phenomenon" observed when sectional images are re-sliced into other planes in 3D space
17. Since the shapes of most brain structures are irregular, to avoid excessive smoothing that might
lead to the loss of correct anatomical features, we re-sliced the optimized delineation back to the
coronal plane for further examination. The boundaries of each brain structure in the three standard
anatomical planes of STAM underwent multiple iterations of examinations.
Once all traditional canonical anatomical planes were acquired, we developed a canonical plane
visualization platform (<http://atlas.brainmatics.org/STAM/reference/index.html>), comprising 700
coronal levels, 256 sagittal levels, and 367 horizontal levels, each with a projection thickness of 20
156 μm . Additionally, we also visualized the non-linearly mapped specific gene-type neuron distribution
datasets overlaid with the boundaries of STAM on this platform
(<http://atlas.brainsmatics.org/ValidationOfSTAM/list.html>).
The three-dimensional topography of whole-brain anatomical structures
By aligning and resampling the illustrated coronal atlas levels into a 3D image stack with isotropic
10 μm resolution, we generated the 3D topography of each brain structure through surface
reconstruction (Fig. 3a). A detailed description of the reconstruction procedure is provided in the
'Reconstruction and visualization' part of the Methods section. With careful balancing, we preserved
the authentic 3D topographies of structures, minimizing artifacts from the reconstruction
process. The resulting models retain many anatomical details, as seen in Fig. 3a and Fig. 3b. Brain
structures were reconstructed from the finest level of the brain structural ontology, with higher
order structures hierarchically assembled from their constituent parts, creating a complete set of
brain structures across different anatomical levels (Fig. 3c, Mov. S2).
The reconstructed topographies highlight the changing and irregular nature of brain areas. For
example, the thalamus retains fine anatomical features such as fiber-penetrating holes (Fig. 3d).
Benefitting from the details provided by the high-resolution MOST-Nissl dataset, we could
accurately reconstruct fine structures, especially stratifications. One example is the three-layer
structure of the olfactory tubercle (OT), including the islands of Calleja embedded within it, which
are hard to reconstruct on non-cytoarchitectural images (Fig. 3e). Another example is the
somatosensory area, where we not only visualized its 3D morphology and spatial relationship with
surrounding sensory regions, but also subdivided the second somatosensory area into six layers (Fig.
3f).
We developed a 3D visualization platform for STAM that incorporates the 3D atlas label images,
reconstructed 3D models, cerebral vessels from previous studies, and single-neuron morphology
datasets from various sources¹⁸. These data are deeply integrated, allowing us to calculate the brain
regions supplied by specific vascular branches and the branches that pass through particular brain

6

regions. The same procedures apply to the 182 relationships between brain regions and single-neuron

morphology data. We provide an online query service for this integrated information, facilitating
systematic analyses based on multiple types of neuroinformation (Fig. 3g). A detailed description
about the query service is provided in the 'Web service reconstruction' part of the Methods section.
The brain positioning system based on both cranial and intracranial datum marks
To establish intracranial datum marks, we first selected eight anatomical structures, including ac,
nucleus ambiguous (AMB), corpus callosum (cc), dentate gyrus granular layer (DGsg), dorsal raphe
(DR), lateral amygdalar nucleus (LA), medial geniculate complex (MG), and the vestibulocochlear
nerve (VIII_n), from the MOST-Nissl dataset that are easily recognizable in cytoarchitecture images.
Structures such as ac, AMB, cc, and VIII_n are also clearly visible in images of

immunohistochemically stained sections

192, including acetylcholinesterase staining, while DGsg, LA, and MG is identifiable in the

193 images acquired by magnetic resonance imaging (MRI). We defined geometric features such as the

194 center and endpoints of these structures as intracranial datum marks, totaling 18 points (Fig. 4a).

Their specific names and coordinates can be found in Tab. S3. The selection of intracranial datum
marks considers the anterior-posterior, left-right, and superior-inferior directions within the brain.
These datum marks were determined based on their surrounding cytoarchitecture information in
three anatomical orientation images by many neuroanatomists (Fig. 4b, Fig. S7).
To bridge STAM and traditional skull-based stereotaxic coordinate systems, we then utilized
a fluorescent micro-optical sectioning tomography (fMOST) technique to acquire a dataset containing
both skull and brain tissue with propidium iodide (PI) staining, providing a 3D image of the entire
mouse head with a horizontal resolution of 0.325 $\mu\text{m}/\text{pixel}$ and axial resolution of 1 $\mu\text{m}/\text{pixel}$. This
dataset allowed us to obtain the 3D structures of cranial datum marks, namely Bregma and Lambda
(Fig. 4c, d). We extracted the contour of the MOST-Nissl dataset and the cranial cavity from the 3D
image dataset of the entire mouse head, and used non-linear image registration method to align them
together, establishing the spatial correspondence between cranial and intracranial datum marks (Fig.
4e-g, Tab. S3)¹³.
Since STAM's datum marks are distributed throughout the entire brain and some of them can be
identified in MRI and/or immunohistochemistry images, they serve as landmarks for constructing
spatial mapping relationships between non-whole-brain data and certain non-optical imaging
modalities. For example, we established a precise spatial mapping relationship between STAM and
the Waxholm Space (WHS), an MRI-based atlas (Fig. 4h-i). By manually determining the
corresponding datum marks between the two atlases through anatomical analysis, we enabled the
integration of diverse neural data from different imaging modalities^{19,20}.
Additionally, we used these datum marks to develop a micron-resolution brain structural positioning
system. The cranial datum marks in this system are also landmarks traditionally used for

7

constructing the coordinate system in the stereotaxic²¹⁷ atlases of the mouse brain⁶. To assist
neuroanatomists and physiologists, we developed a virtual stereotaxic surgery service
(http://atlas.brainsmatics.org/3DViewer/?index=Virtual_Inject&type=virtual), enabling intelligent
planning of virtual surgeries based on the STAM's brain positioning system. For example, when
performing a virus injection into the small Dopaminergic A13 group (A13) deeply embedded in the
brain, the service can automatically calculate an appropriate injection path and display the brain
structures it traverses after specifying the target and the brain structures to avoid (Fig. 4k). This
function is beneficial in situations where certain cortical structures are integral to the entire neural
circuitry and thus require protection, ensuring they remain undisturbed by the intrusion of injection
equipment. We believe this service can assist in pre-surgical planning for mouse brain surgeries,
reducing the risk of failure and minimizing animal use, thus promoting animal welfare.
Visualization of STAM on arbitrary-angle planes
Through a neuroglancer framework (Google Inc., www.neuroglancer.org) based web platform and
hierarchically organized TB-size MOST-Nissl dataset, we visualized 2D atlas planes at arbitrarily
selected cutting angles, with isotropic 1 μm resolution and any desired projection thickness between
1 and 20 μm (Fig. 5a, b, Mov. S3). This unprecedented capability distinguishes our STAM from
traditional atlases and CCF that could only provide the three canonical planes.
One advantage of arbitrary-angle planes is the ability to observe anatomical features that are not
visible in traditional planes. For example, while the medial habenula (MH) is easily recognized on
coronal, horizontal, and sagittal planes (Fig. 5c), the complete morphology of the fiber bundle
traversing the entire MH is only visible from a resliced plane at a specific angle, running
anteroventral to dorsoposterior (Fig. 5d). The barrel field of primary somatosensory cortex (SSp) is
another feature structure that can only be observed from a deviated angle of view. Therefore, we cut
an oblique slice through layer 4 of SSp, revealing the typical barrel textures in a cytoarchitectural
image with 1 μm resolution (Fig. 5e, f).
Given the complex 3D morphology of many brain structures and the rapid transitions between
anatomical regions, even a slight angle shift can produce varied atlas planes. Fig. 5g and Fig. 5h
compare canonical and non-canonical sagittal planes, showing how a mere 5.6-degree yaw angle
causes the primary motor area (MOp), posterior parietal association areas (PTLp), retrosplenial area,
lateral agranular part (RSPagl), anteromedial visual area (VISam), and posteromedial visual area
(VISpm) to disappear from the oblique plane. This not only indicates the importance of constructing
an accurately delineated atlas, but also demonstrates the necessity of providing non-canonical atlas
planes, which can facilitate brain anatomical and physiological investigations.
Neuronal circuits mapping

8

The 3D space of STAM provides an²⁵¹ ideal foundation for precise localization of subtle
neuroinformation, such as neuronal circuits. We integrated single-neuron morphology datasets
obtained through fMOST imaging technology and public databases, registered them onto STAM,
identified their soma locations and projection targets, and analyzed the connectivity between brain
regions²¹. This enabled us to establish a comprehensive connectivity map
(<http://atlas.brainsmatics.org/3DViewer/?index=Assembling&type=circuits>) of the entire mouse
brain at single-neuron projection level (Fig. 6a, b). This connectivity map depicts possible
connections between any two brain structures, and the queried single-neuron morphology data can
be visualized, with the locations of all the branching points and terminals labeled in the 3D space
(Fig. S12i). Benefiting from the deep integration of multiple neuroinformation, atlas levels on
canonical or non-canonical planes could be visualized alongside neuron morphologies, facilitating
the 3D spatial localization of neuronal circuits (Fig. 6b).
The STAM can also be used to localize afferent and efferent connections for any brain structure of
interest (Fig. 6c). Using the newly annotated hippocampal structure "Field CA1 of hippocampus,
stratum oriens, dorsal domain" (CA1sod) as an example, our 3D visualization platform enables the
observation of afferent neurons from the diagonal band nucleus (NDB) and lateral septal nucleus,
rostral part (LSr) in different colors, along with their somas, branching points, terminals, and the
brain structures of STAM where they are localized. (Fig. 6d, e).
We used the STAM to analyze the connection and projection pattern of the mapped neuron
morphology data. Fig. 6f visually compares the projection pathways from “Field CA3 of
hippocampus, pyramidal layer, intermediate domain” (CA3spi) and “Field CA3 of hippocampus,
stratum oriens, dorsal domain” (CA3sod) in horizontal and sagittal views. We found that one
efferent neuron from CA3spi primarily projects to the ipsilateral and contralateral sides of
hippocampus, with only a few axons reaching the lateral septal nucleus, caudal part (LSc) and LSr
in the CNU region. In contrast, another efferent neuron from CA3sod sends most of its fibers to
deeper brain structures, such as the Medial septal nucleus (MS), Dorsal peduncular area (DP), and
even Taenia tecta (TT). We also observed the preferred projection among sub-regions and sub-
layers, as described in 22. As shown in Fig. 6f, the efferent neuron from CA3spi primarily projects
to the dorsal and intermediate domains of the Field CA3, while the ventral domain is sparsely
projected. The efferent neuron from CA3sod primarily projects to the stratum oriens and stratum
radiatum layers of CA1-CA3, with minimal projections to other layers. These differences are
quantitatively compared in Fig. 6g. The subtle subdivisions of the hippocampus introduced to
STAM, along with the precise localization of projection terminals, serve as important references for
understanding neural information transmission and encoding in the hippocampus.
In addition to the spatial localization of projection terminals, we used STAM to localize the
branching points of neuronal circuits. For the efferent neuron from CA3spi shown in Fig. 6f, we
observed that its fibers primarily branch within fimbria (fi) and SF before further projecting to the
target nuclei LSr and LSc (Fig. S8a, b). A similar branching pattern was found for its contralateral
projections, which first give off branches in alveus before entering CA fields (Fig. S8c). For the
efferent neuron from CA3sod, we observed that its projections to CA fields mainly traverse through
the stratum oriens layers along the hippocampal axes, branching into neighboring layers as they
progress (Fig. S8d, e). By precisely localizing both terminals and branching points of neuronal
circuits, we can depict the complex projection patterns of individual neurons registered to STAM,
supporting computational modeling of signal propagation dynamics at the single-neuron level 23.
New annotations of STAM
We carefully compared the differences and similarities in the delineation between STAM and CCF,
listing a total of 236 newly drawn brain regions and nuclei, as well as the criteria to determine these
structures (Tab. S4). The majority of these newly delineated structures comes from the finer and
continuous cytoarchitecture information provided by the 1 μ m resolution MOST-Nissl dataset.
Some of these structures involve finer layering, primarily distributed in olfactory areas (OLF) and
CNU, while others involve more detailed sub-regions, mainly distributed in the hypothalamus and
midbrain (Fig. S9a-g). Particularly, for structures such as the islands of Calleja and the major island of
Calleja, which exhibit island-like shapes, with individual particles having diameters of only a few
tens of micrometers and scattered among other structure, STAM can still demonstrate their location
and complete 3D topography (Fig. S9h).
Other newly annotated brain structures mainly come from the multiple-source images registered
onto the MOST-Nissl dataset, including those from specific gene-type neuron distributions, in situ
hybridization (ISH), and immunohistochemistry. These images assist in the identification of some
brain regions and nuclei that might be challenging to discern solely based on the Nissl-stained
dataset. Specifically, we refined the anatomical boundaries of the nucleus accumbens (ACB) and
zona incerta (ZI) by identifying their sub-domains using Thy1-cre and Vglut2-Cre neuron
distribution images obtained through fMOST technology (Fig. S10a-f). We also used the ISH
images from online database 24 to define subdivisions of the periaqueductal gray (PAG) (Fig. S10g
i). Additionally, we developed a non-linear image registration algorithm to align the sub-domains
of the hippocampal formation, delineations based on combined connectivity and molecular markers
incorporated in previous studies 16 onto STAM (Fig. S11). The image registrations above are
performed

by the ANTs tool 25. The quantitative evaluation for the registrations can be found in Tab. S5.
Web service for miscellaneous research needs
To facilitate the application of STAM in neuroscience researches, we developed a series of web
services for miscellaneous needs. Besides the visualization of 2D atlas levels and 3D
topographies, we provide open access to this isotropic 1 μ m resolution Nissl-stained dataset,

and have developed an online tool that allows users to 322 select any 3D spatial range of interest and
323 choose various down-sampling rates for downloading. This unprecedented open-accessible 3D
cytoarchitecture brain image with 1 μm resolution reveals abundant and intact anatomical details of
the entire mouse brain.
Leveraging the isotropic 1 μm resolution advantage of STAM, we also managed a cloud service for
the spatial localization of brain slices, offering a more automated approach for using brain atlases
in neuroscience research, reducing manual comparison and registration calculations. Users can
upload brain slice images stained with PI or 4',6-diamidino-2-phenylindole (DAPI), and the service,
through backend computation, will re-slice a section from the 3D space of STAM that is closest to
the location of the user-uploaded brain slice, providing the distribution of brain regions at the same
position and angle (Fig. S12).
We also established a spatial mapping between STAM and CCF. Based on the generated
invertible deformation field, we developed a suite of tools for inter-atlas mapping, enabling the
non335

linearly mapping of single neuron morphology data and 3D whole brain images from CCF to
336 STAM, and vice versa. This tool suite includes an online mapping service, desktop plugins for Fiji
and ImageJ designed for the neuroscientists, and a python-wrapped application program interface
suitable for programmers.
All web services, including previously introduced visualization, data sharing
registration, neuronal circuits mapping, and virtual surgery, have been integrated into a single
entry341

point (<http://atlas.brainsmatics.org/STAM/>). A comparison of these services with existing typical
resources and tools in Tab. S6. By offering open access to all the neuroinformation of our services,
integrating them together with mutual queries, and providing easy download routine for the atlas
data as well as calculated data, we adhere to the FAIR principles—findability, accessibility,
interoperability, and reusability²⁶. We believe that the various developed web services can meet
diverse needs in neuroscience research, and provide a solid spatial localization foundation for
integrating neural information from different sources and modalities.

11

DISCUSSION
Summary of this work This study employed MOST technology to capture a 3D Nissl-stained
cytoarchitectonic image dataset with isotropic 1 μm resolution. By combining
immunohistochemistry, ISH, neural circuit labeling, and distribution patterns of specific gene-type
neurons from various reference datasets, we constructed a stereotaxic topographic atlas. This atlas
enables exploration of cytoarchitecture images of the mouse brain from not only the coronal, sagittal
and horizontal planes, but also at any angle at single-cell resolution. We reconstructed the 3D
topography of 916 brain structures, offering a 3D visualization platform that integrates various
neuronal data, including single-neuron morphology for mutual query. At the core of STAM is
a brain positioning system, serving as the foundation for a suite of informatics tools designed to
meet diverse neuroscience needs. These tools include inter-atlas mapping for 3D imaging and
neuron morphology data, virtual surgery for stereotaxic surgery planning, and brain slice image
registration to a 3D reference atlas.
Comparison with previous work Before our work, there were several published stereotaxic atlases
of the mouse brain^{6,7}. However, inconsistencies in the anatomical planes from different directions,
along with the large and non-uniform axial spacing between adjacent slices, made it challenging to
align slices and reconstruct the authentic topography of brain structures. To address these issues of
non-continuity and anisotropic spatial resolution, we utilized one set of 3D Nissl-stained images
that simultaneously provides single-cell resolution image sequences of coronal, sagittal, horizontal
planes, and even arbitrarily oriented planes from the same brain for the first time. This approach
ensures that the borders of brain structures align seamlessly across different planes. Importantly,
when the STAM is used as a standard template for registration, users can re-slice the STAM at any
angle to match the cutting angles of experimental brains, thereby maximizing anatomical accuracy.
The 3D Nissl-stained image dataset is publicly available for download and holds promise as a
fundamental resource for various neuroscience research.
There also exist 3D mouse brain atlases, represented by the CCF and WHS. However, the images
used to build these atlases lack cytoarchitecture information, making it difficult to clearly depict
fine details necessary for accurate delineation of brain structures, such as hippocampal layers and
different CNU (e.g. OT) or hypothalamic nuclei (e.g. ventromedial hypothalamic nucleus), as shown
in Fig. S9i-k. Additionally, the autofluorescence image datasets lack skull information, preventing
the establishment of the cranial landmarks and skull-based coordinate system. This limitation
hinders their use in typical neuroscience tasks, such as guiding stereotaxic surgeries for physiological
experiments.
In contrast, our atlas, constructed using Nissl-stained isotropic 1 μm resolution 3D image dataset,
provides a clear view of each individual cell and its surrounding cells. This allows for the
observation of the size, shape, and spatial arrangement patterns of individual somas in any
direction

cytoarchitectural texture information aids in
identifying genuine and accurate boundaries of brain structures. The reconstructed 3D topographies
also exhibit richer and more detailed anatomical features. Additionally, by utilizing micron387
resolution images of the skull and brain tissue, we established a spatial positioning system spanning
both the cranial and intracranial regions, making it the highest spatial resolution 3D brain atlas for
any mammalian species to date. With these innovative features, our newly constructed STAM serves
as an ideal atlas for the registration and integration of results from single-cell level brain mapping
projects, including spatial transcriptome and single neuron morphology reconstructions.
Innovation and contribution The micron-resolution 3D continuous MOST image dataset used to
create STAM enables users to visualize individual cells across the entire mouse brain, providing
spatial location and morphological information for all cells. Therefore, the STAM can serve as a
standard reference atlas for brain mapping projects with single-cell resolution, including spatial
transcriptomics and neural circuit studies. To demonstrate its precise spatial localization capability,
we registered single-neuronal circuits datasets from multiple resources onto STAM, localizing
neuron somas, branching points, and terminals to their corresponding brain structures of STAM.
Additionally, we analyzed the distribution of each neuron's terminals, their projection patterns, and
the projection length in different brain regions.
During the process of constructing the atlas, we carefully considered the practical needs of the
neuroscience field. Firstly, to meet the requirements of histology and comparative anatomy research,
we established a web service for the brain atlas. Users are able to browse the distribution of brain
structures on standard anatomical planes as well as the arbitrary-angle planes online, along with the
3D shapes of these brain structures. Furthermore, to address the current demands of research on
projection circuits and cell types, we have established an online registration tool, providing
researchers with a cloud service for spatial localization that is compatible with both standard and
non-standard anatomical orientations of brain slices.
Limitations of our pipeline The MOST-Nissl dataset comprises 14,000 coronal sections with a 1-
410 μm axial interval. Assuming an estimated average drawing speed of one day per section, the total
411 workload would be around 38 years. Therefore, manually delineating brain structural boundaries on
each coronal section is unrealistic and unnecessary. With this consideration, we selected a "typical
coronal plane" at a 20- μm interval for manual annotation of brain structural boundaries.
Subsequently, we constantly switched to other standard anatomical planes for checking and
corrections, ensuring the accuracy of structural delineations on coronal planes between the selected
typical ones. Nevertheless, despite the high-quality requirements for atlas images and the complex
workflow resulting from management and quality control in collaborative brain region annotation
by multiple individuals, it took us approximately 10 years to establish a comprehensive solution for
image acquisition and processing, atlas creation, and the development of miscellaneous services.

13

Considering the significantly greater effort required for 420 mapping non-human primates' or even
human' brain atlases, in the future, we will leverage recent advancements in deep learning
technology to automate and expedite the process of atlas creation by learning the texture information
of different brain structures in cytoarchitecture images¹⁴.
Future work: towards a more comprehensive atlas and more powerful tools The brain atlas
reported in this article currently covers only the central nervous system of the mouse. In the
future, we plan to extend this mouse brain atlas to primates, including the marmoset. Given the
labor-intensive nature of constructing this atlas, we aim to develop deep learning-based
computational methods for high-throughput automated atlas construction, leveraging the anatomical
structure labeling of STAM. Additionally, we intend to integrate more multi-omics data, including
spatial transcriptomics, and incorporate more comprehensive neuronal circuit information into the
STAM. Our goal is to create a user-friendly, all-in-one open-access database *in the near future*.
Acknowledgments
We thank Zhangheng Ding, Zhenyu Wang, Xiaoyun Liu, Zhe Zhao, Xue Peng, Qing Zhou, Yong
Chen, Lei Su, Jing Li, Hang Zhou, Hang Lu, Zhuonan Duan, Xue Li, Pan Luo, Xiaoyu Zhang, Chen
Zhang, Xiaoyan Zhang, Can Zhou, Lei Deng, Tianpeng Luo, Yalan Yu, Lu Tan, Yao Jia, Ling Liu,
Shenghua Cheng, Wenyan Guo, Guanghui Chen, Rui Guo, Yu Shen, Ke Bai, Xiao Hu, Xianlin
Song, Dan Liu, Tian Lei, Shaoyou Qin, Shukang Luo, Rong Xiao, Mei Yao, Wen Sun, Guoqing
Fan, Mingxiang Zhang, Luyao Wei, Yaru Cao, Zhi Wang, Jing Jin, Tao Tang, Zhiwen Xie, Ke
Zhang, Tao Zhang, Wenwei Li, Wu Chen, Yinchuan Luo, Mingwei Liao, Yuegang Wang, Shuijuan
Wang, Jinxing Lu, Yanguo Wang, Yaochen Xiao, Yachao Di, Qing Ye, Zhiguang Wang and Yunyi
Gu for their participation in this work. This work was supported by funding from the STI2030-
Major Projects (2022ZD0205201, 2021ZD0201001, 2021ZD0200203), National Natural Science
Foundation of China Grants (32192412, T2122015, 61890953, 61721092), and
NIHU19MH114821.
Author contributions
Conceptualization, H.G. and Q. L.
Brain sample preparation, X. L., B. L., M. R. and S. C.
Acquisition of Data, T. J., X. J.
Software, G. Y., Y. L., Z. L., H. N., C. T., W. D. and W. S.
Project Administration, A.L., X. L., and H.G.
Anatomical Delineation, Z. F., Y. L., X. L., H. N., L. C., Q. Z., A. T., C. T., Z. X. and H. D.
Data Investigation, X. C., Z. W., M. Z., C. D., S. B. and J. Z.
Web Visualization, X. C., S. Y., Z. Z. and J. L.
Manuscript Preparation, Z. F., X. L., H. D., H. G. and Q. L.
Declaration of interests
The authors declare no competing interests.
Figure legends
**Fig. 1** Overview of STAM and its attached resources and tools. a, The 2D images are the coronal,
sagittal and horizontal planes resliced from the same 3D cytoarchitectural image used to construct
the STAM. The black lines on these planes represents the delineated boundaries of brain regions
14
and nuclei. The 3D model in the middle is the 461 reconstructed brain outline of STAM. The light462
yellow planes cutting the outline represents the generated atlas planes of STAM with any desired
arbitrary angles, which is described in (d) in detail. b, The horizontal, coronal and sagittal atlas
levels in the upper-middle, upper right, and bottom, with the overlaid labels of brain regions and
nuclei with different colors. The text on the upper-left is the basic information of STAM. The brain
model on the upper-right corner displays the navigation widget snapshotted from the visualization
platform, which is used for switching between canonical planes. c, The left part of this sub-figure
displays hierarchically organized anatomical ontologies of STAM. The upper-right is the 3D model
of the checked brain structure isocortex, the lower-right is the expanded view of isocortex,
consisting of its subdivisions. d, A resliced plane from the 3D cytoarchitectural image with arbitrary
slicing angle. The colored labels on the right part represents the delineation of different brain
regions. The local region on this resliced plane enclosed by a black rectangle is magnified to single
cellular resolution at the left-bottom corner. e, The open access to our cytoarchitectural image used
for constructing STAM. The cubes drawn with dashed and solid lines represents different ways to
select the regions of interest for downloading data. The cyan model in the middle is the reconstructed
topography of the brain structure NLL. The cropped 3D cytoarchitectural image data located at the
bounding box of NLL is shown on the right side. f, Localization of numerous neuroinformation
using our STAM. The green dots represent the terminals of the localized neurons, the located brain
regions of which are then calculated. g, The brain slice registration tool provided by the visualization
platform of STAM. The right part presents the corresponding atlas level with the registered slice. h,
A series of inter-atlas mapping using STAM. The first row indicates the bidirectional mapping
between STAM and CCF, which is used for integrating multiple-resource neuron morphology
dataset to STAM. The second row indicates the mapping between STAM and Waxholm Space
(WHS) using intracranial landmarks defined in STAM, which could then be used as the multi485

modal image fusion, as the figure shows. The third row indicates the mapping from the default
position of STAM to the flat-skull position by using the cranial datum marks, based on which the
commonly used skull-based coordinate system is introduced in STAM. The abbreviations for brain
structures correspond to the full names found in the appendix. The same applies hereinafter.
Fig.2 Delineating the brain regions and nuclei of STAM using high-resolution 3D cytoarchitectural
image dataset and other supplementary approaches. a, The main part of this sub-figure showcases
the original coronal, sagittal, and horizontal images of the MOST-Nissl dataset. The tiny cube
locating at the intersection of three orthogonal planes represents a small image volume randomly
selected, which is then magnified at the lower-right. One of this selected volume's corner is further
magnified to show the single neuronal soma located at this corner. b, One sagittal plane from the
cytoarchitectural image dataset of STAM. c, Zoom-in to the cortical region indicated by the
rectangle in (b), where the lamination pattern changes from 5 layers to 6. d, An intact coronal plane
from the same image dataset, locating at the position indicated by the black line in (b). e, Zoom-in
to the region marked by the rectangle in (d). f, The display of multiple source image datasets used
...
The construction of STAM primarily utilized the nomenclature of the ARA and Brain maps 4.0,
with occasional references to *The Mouse Brain in Sterotaxic Coordinates* for the delineation of a
few sub